# Sharpness-Aware Minimization Efficiently Selects Flatter Minima Late In Training

**Zhanpeng Zhou**[1][*][†]**, Mingze Wang**[2][*]**, Yuchen Mao**[1]**, Bingrui Li**[3]**, Junchi Yan**[1,4][†]
[1]Sch. of Computer Science & Sch. of Artificial Intelligence, Shanghai Jiao Tong University
[2]Peking University [3]Tsinghua University [4]Shanghai Artificial Intelligence Laboratory
`{zzp1012,yanjunchi}@sjtu.edu.cn`

## ABSTRACT

Sharpness-Aware Minimization (SAM) has substantially improved the generalization of neural networks under various settings. Despite the success, its effectiveness remains poorly understood. In this work, we discover an intriguing phenomenon in the training dynamics of SAM, shedding light on understanding its implicit bias towards flatter minima over Stochastic Gradient Descent (SGD). Specifically, we find that *SAM efficiently selects flatter minima late in training*[1]. Remarkably, even a few epochs of SAM applied at the end of training yield nearly the same generalization and solution sharpness as full SAM training. Subsequently, we delve deeper into the underlying mechanism behind this phenomenon. Theoretically, we identify two phases in the learning dynamics after applying SAM late in training: i) SAM first escapes the minimum found by SGD exponentially fast; and ii) then rapidly converges to a flatter minimum within the same valley. Furthermore, we empirically investigate the role of SAM during the early training phase. We conjecture that the optimization method chosen in the late phase is more crucial in shaping the final solution's properties. Based on this viewpoint, we extend our findings from SAM to Adversarial Training. We released our source code at `https://github.com/zzp1012/SAM-in-Late-Phase`.

## 1 INTRODUCTION

Understanding the surprising generalization abilities of over-parameterized neural networks yields an important yet open problem in deep learning. Recently, it has been observed that the generalization of neural networks is closely tied to the sharpness of the loss landscape (Keskar et al., 2017; Zhang et al., 2017; Neyshabur et al., 2017; Jiang et al., 2020). This has led to the development of many gradient-based optimization algorithms that explicitly/implicitly regularize the sharpness of solutions. In particular, Foret et al. (2021) proposed Sharpness-Aware Minimization (SAM), which has substantially improved the generalization and robustness (Zhang et al., 2024) of neural networks across many tasks, including computer vision (Foret et al., 2021; Chen et al., 2022; Kaddour et al., 2022) and natural language processing (Bahri et al., 2022).

Despite the empirical success of SAM, its effectiveness is not yet fully understood. Andriushchenko & Flammarion (2022) has shown that existing theoretical justifications based on PAC-Bayes generalization bounds (Foret et al., 2021; Wu et al., 2020a) are incomplete in explaining the superior performance of SAM. Additionally, recent theoretical analyses of SAM's dynamics and properties often rely on unrealistic assumptions, such as a sufficiently small learning rate or perturbation radius, which undermines the validity of their conclusions. Understanding the hidden mechanisms behind SAM remains an active area of research.

Recent works show that the effectiveness of gradient-based optimization methods can be attributed to their *implicit bias* toward solutions with favorable properties (Vardi, 2023). One notable example of such implicit bias is that Stochastic Gradient Descent (SGD) and its variants tend to find flat minima, which often leads to better generalization (Keskar et al., 2017; Zhang et al., 2017). It is known that

---

[*]Equal contribution.
[†]Corresponding author. The work was in part supported by NSFC (62222607) and Shanghai Municipal Science and Technology Major Project (2021SHZDZX0102).

[1]"Late in training" refers to the phase when the optimization process is nearing the late stages of training.

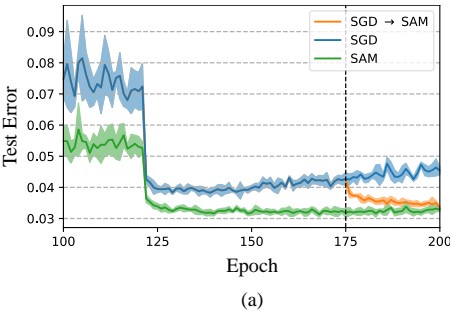 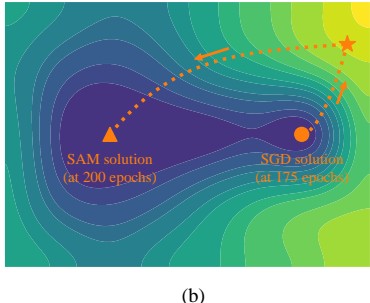

(a)                                                          (b)

Figure 1: **SAM operates efficiently even when applied only during the final few epochs of training**. **(a)** Test error curves for WideResNet-16-8 on CIFAR-10 trained with different strategies. The blue and green baseline curves represent training solely with SGD and SAM, respectively. The orange curve shows the test error of a neural network initially trained with SGD up to epoch $t = 175$, followed by SAM training up to epoch $T = 200$. The first 100 epochs are omitted for clarity. Curves are means and standard deviations over five trials with different random seeds. The detailed settings and hyper-parameters are described in Section 2. **(b)** A schematic picture of the training trajectory after applying SAM late in training. The contour plot represents the loss landscape, with darker regions indicating lower loss. The orange dotted lines depict the path of the SAM iterator.

SAM selects flatter minima over SGD in practice, which represents a form of implicit bias as well. While the design of SAM is inspired by sharpness regularization, its practical implementation (see Equation (3)), which minimizes a first-order approximation of the original objective (see Equations (1) and (2)), does not explicitly achieve this. Understanding SAM's implicit bias towards flatter minima, especially in comparison to SGD, is of paramount importance for explaining its effectiveness.

In this work, we uncover an intriguing phenomenon in the training dynamics of SAM, which sheds light on understanding its implicit bias toward flatter minima. Specifically, we find that SAM operates efficiently even when applied only during the last few epochs of training[2]. As shown in Figure 1 (a), SAM applied in the last few epochs—after initial training with SGD—can achieve nearly the same test performance as training exclusively with SAM. Furthermore, our analysis of the solution sharpness reveals that last-few-epochs SAM identifies flatter minima as effectively as full SAM training. We conclude that *SAM efficiently selects flatter minima over SGD late in training*. Contrary to the conventional view that early training dynamics are critical (Achille et al., 2019; Frankle et al., 2020), our discovery underscores the importance of the late training phase for generalization.

Subsequently, we delve deeper into the mechanism behind the efficiency of SAM in selecting flatter minima over SGD during the late phase of training. Theoretically, we identify a two-phase picture in the learning dynamics after applying SAM late in training: *Phase I.* SAM initially escapes from the relatively sharp minimum found by SGD exponentially fast; *Phase II.* it then rapidly converges to a flatter minimum over SGD within the same valley. See Figure 1 (b) for an illustration. As outlined in Table 1, we characterize the two-phase picture into four key claims **(P1-4)**, each supported by rigorous analyses of SAM's convergence, escape behavior, and bias toward flatter minima. These claims are further validated by experiments in both toy models and real-world scenarios. This two-phase picture explains SAM's ability to efficiently find flatter minima, even when applied only in the last few epochs, yielding novel insights into its implicit bias and effectiveness.

Furthermore, we investigate the necessity of applying SAM during the early phase of training. Our results show that early-phase SAM has a minimal effect on both generalization and sharpness of the final solution. Based on these findings, we conjecture that the choice of the optimization algorithm during the late phase of training plays a more critical role in shaping the final properties of the solution. Extending this viewpoint to Adversarial Training (AT) (Madry et al., 2018) and robustness, we find that similar to SAM, AT efficiently selects robust minima when applied late in training.

In summary, our work focuses on the implicit bias of SAM in the late training phase, which has not yet been carefully studied. Our two-phase picture advances the understanding of its effectiveness.

---

[2]Andriushchenko & Flammarion (2022) observed a similar finding but did not explore it in depth. Surprising, we first show that even few epochs of SAM at the end of training yield generalization comparable to full SAM training. We study this further via extensive experiments under various setup and rigorous theoretical analysis.

## 2 BACKGROUND AND PRELIMINARIES

**Notation and Setup.** Let $\mathcal{D}_{\text{train}} = \{(\boldsymbol{x}_i, \boldsymbol{y}_i)\}_{i=1}^n$ and $\mathcal{D}_{\text{test}} = \{(\boldsymbol{x}_i, \boldsymbol{y}_i)\}_{i=n+1}^{n+m}$ be the training set and the testing set, where $\boldsymbol{x}_i \in \mathbb{R}^d$ is the input and $\boldsymbol{y}_i \in \mathbb{R}^k$ is the corresponding target. We consider a model in the form of $f(\boldsymbol{x}; \boldsymbol{\theta})$, where $\boldsymbol{\theta} \in \mathbb{R}^p$ are the model parameters and $f(\boldsymbol{x}; \boldsymbol{\theta}) \in \mathbb{R}^k$ is the output of the input $\boldsymbol{x}$. The loss of the model at the $i$-th sample $(\boldsymbol{x}_i, \boldsymbol{y}_i)$ is denoted as $\ell(\boldsymbol{y}_i; f(\boldsymbol{x}_i; \boldsymbol{\theta}))$, simplified to $\ell_i(\boldsymbol{\theta})$. The loss over the training set and the testing loss are then given by $\mathcal{L}_{\mathcal{D}_{\text{train}}}(\boldsymbol{\theta}) = \frac{1}{n} \sum_{i=1}^n \ell_i(\boldsymbol{\theta})$ and $\mathcal{L}_{\mathcal{D}_{\text{test}}}(\boldsymbol{\theta}) = \frac{1}{m} \sum_{i=n+1}^{n+m} \ell_i(\boldsymbol{\theta})$, respectively. Unless otherwise specified, we denote $\mathcal{L}(\boldsymbol{\theta}) = \mathcal{L}_{\mathcal{D}_{\text{train}}}(\boldsymbol{\theta})$ for simplicity. For classification tasks, we define the classification error on the test set $\mathcal{D}_{\text{test}}$ as $\text{Err}_{\mathcal{D}_{\text{test}}}(\boldsymbol{\theta})$. For a matrix $A$, let $\|A\|_2$, $\|A\|_F$, and $\text{Tr}(A)$ denote its spectral norm, Frobenius norm and trace, respectively.

For our theoretical analysis in Section 4, we always assume $k = 1$ for simplicity[3] and consider the squared loss, i.e., $\ell_i(\boldsymbol{\theta}) = \frac{1}{2}|f(\boldsymbol{x}_i; \boldsymbol{\theta}) - y_i|^2$. We focus on the over-parameterized case in the sense that $\min_{\boldsymbol{\theta}} \mathcal{L}(\boldsymbol{\theta}) = 0$. To characterize the local geometry of the training loss at point $\boldsymbol{\theta}$, we consider the Fisher matrix $G(\boldsymbol{\theta}) = \frac{1}{n} \sum_{i=1}^n \nabla f(\boldsymbol{x}_i; \boldsymbol{\theta}) \nabla f(\boldsymbol{x}_i; \boldsymbol{\theta})^\top$ and the Hessian matrix $H(\boldsymbol{\theta}) = G(\boldsymbol{\theta}) + \frac{1}{n} \sum_{i=1}^n (f(\boldsymbol{x}_i; \boldsymbol{\theta}) - y_i) \nabla^2 f(\boldsymbol{x}_i; \boldsymbol{\theta})$, where $G(\boldsymbol{\theta}) \approx H(\boldsymbol{\theta})$ in the low-loss region.

**Sharpness-Aware Minimization.** The central idea behind SAM is to minimize the worst-case loss within a neighborhood of the current parameters. Specifically, SAM problem can be formulated as

$$\min_{\boldsymbol{\theta}} \mathcal{L}^{\text{SAM}}(\boldsymbol{\theta}; \rho), \quad \text{where} \quad \mathcal{L}^{\text{SAM}}(\boldsymbol{\theta}; \rho) = \max_{\|\boldsymbol{\epsilon}\|_2 \leq \rho} \mathcal{L}(\boldsymbol{\theta} + \boldsymbol{\epsilon}). \tag{1}$$

Here, $\rho$ is a given neighborhood radius. However, finding the perturbation in the parameter space that maximizes the loss, i.e. $\boldsymbol{\epsilon}$, can be computationally intractable in practice. Thus Foret et al. (2021) approximated the maximize-loss perturbation using a first order solution:

$$\boldsymbol{\epsilon} \approx \arg\max_{\|\boldsymbol{\epsilon}\|_2 \leq \rho} \left( \mathcal{L}(\boldsymbol{\theta}) + \boldsymbol{\epsilon}^\top \nabla \mathcal{L}(\boldsymbol{\theta}) \right) = \rho \nabla \mathcal{L}(\boldsymbol{\theta}) / \|\nabla \mathcal{L}(\boldsymbol{\theta})\|_2. \tag{2}$$

Consequently, let $\eta$ be the learning rate and $\xi_t = \{\xi_{t,1}, \cdots, \xi_{t,B}\} \subset [n]$ be the batch indices sampled at iteration $t$, where $B$ is the batch size. The mini-batch loss is given by $\mathcal{L}_{\xi_t}(\boldsymbol{\theta}) = \frac{1}{B} \sum_{i \in \xi_t} \ell_i(\boldsymbol{\theta})$. With a small yet key modification to SGD , the update rule of SAM with stochastic gradients is:

$$\boldsymbol{\theta}^{t+1} = \boldsymbol{\theta}^t - \eta \nabla \mathcal{L}_{\xi_t} \left( \boldsymbol{\theta}^t + \rho_t \nabla \mathcal{L}_{\xi_t}(\boldsymbol{\theta}^t) \right), \quad \text{where} \quad \rho_t = \rho / \left\| \nabla \mathcal{L}_{\xi_t}(\boldsymbol{\theta}^t) \right\|_2. \tag{3}$$

Our experiments are conducted using SAM in Equation (3), whereas our theoretical analyses in Section 4 apply the simplified SAM in Equation (4). Specifically, we approximate the step size $\rho_t$ in Equation (3) with a constant $\rho$. We also assume the independence between the randomness in the inner and outer updates, i.e., $\xi_t^1, \xi_t^2$. These simplifications facilitate mathematical tractability while capturing the central behavior of SAM in training dynamics and generalization, which has been supported by empirical evidence (Andriushchenko & Flammarion, 2022). They have also been widely adopted in recent theoretical advances on SAM (Andriushchenko & Flammarion, 2022; Behdin & Mazumder, 2023; Agarwala & Dauphin, 2023; Behdin et al., 2023; Monzio Compagnoni et al., 2023).

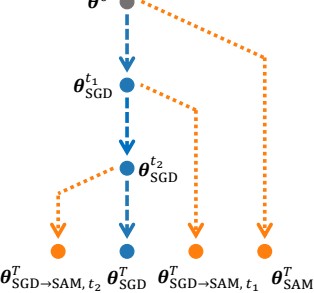

$$\boldsymbol{\theta}^{t+1} = \boldsymbol{\theta}^t - \eta \nabla \mathcal{L}_{\xi_t^1} \left( \boldsymbol{\theta}^t + \rho \nabla \mathcal{L}_{\xi_t^2}(\boldsymbol{\theta}^t) \right). \tag{4}$$

**Switching Method.** In this paper, we demonstrate our main discovery by switching between SGD and SAM at a specific point during training, referred to as *switching method*. Specifically,

Figure 2: **The illustration of the switching method.** Blue dashed lines represent SGD training, while orange dashed lines represent SAM training. $\boldsymbol{\theta}^0$ denotes the random initialization. $t_1, t_2$ denotes two different switching points.

let $T$ be the total number of training epochs and $t$ be the switching epoch, considering the switch from SGD to SAM without loss of generality. We initially train the model from scratch with SGD for $t$ epochs to get $\boldsymbol{\theta}_{\text{SGD}}^t$, and then switch to SAM for the remaining $T - t$ epochs to get $\boldsymbol{\theta}_{\text{SGD} \to \text{SAM}, t}^T$. Notably, when $t = T$ or $t = 0$, the model is trained exclusively with SGD or SAM, simply denoted

---

[3]The extension to the case where $k > 1$ is straightforward.

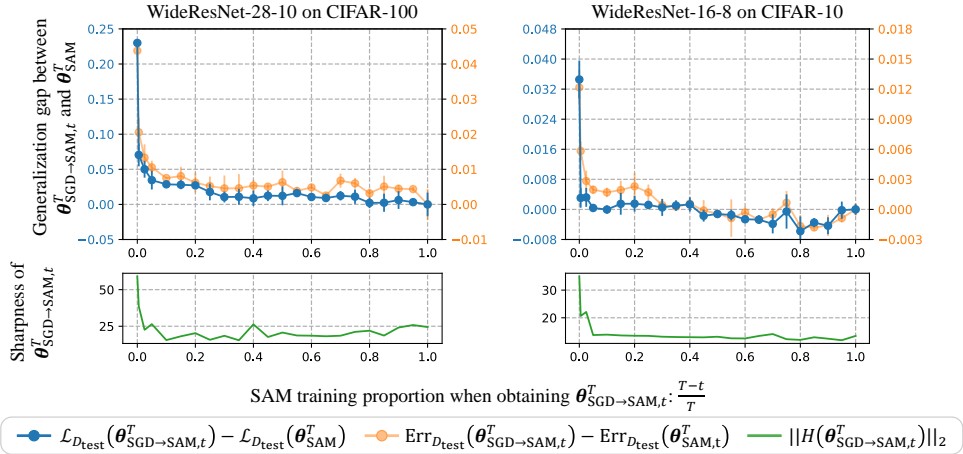

Figure 3: **The impact of SAM training proportion on generalization/sharpness when switching from SGD to SAM.** The generalization gap between the models $\boldsymbol{\theta}^T_{\text{SGD}\to\text{SAM},t}$ and $\boldsymbol{\theta}^T_{\text{SAM}}$ (**top row**) / the sharpness of $\boldsymbol{\theta}^T_{\text{SGD}\to\text{SAM},t}$ (**bottom row**) vs. the SAM training proportion of $\boldsymbol{\theta}^T_{\text{SGD}\to\text{SAM},t}$. Dots represent the mean over three trials with different random seeds, and error bars indicate standard deviations. Results on more datasets and architectures can be found in Appendix B.2.

as $\boldsymbol{\theta}^T_{\text{SGD}}$ and $\boldsymbol{\theta}^T_{\text{SAM}}$, respectively. See Figure 2 for an illustration. We fix the total number of training epochs $T$ and vary the switching point $t$ to study the effect of SAM at different training phases. To ensure a fair comparison, we ensure the same initialization, mini-batch orders, and hyper-parameters across all models, regardless of the switching point. Consequently, the training trajectories of models with different switching points align precisely before the switch.

**Main Experimental Setup.** Following Foret et al. (2021); Kwon et al. (2021), we perform experiments on the commonly used image classification datasets CIFAR-10/100 (Krizhevsky et al., 2009), with standard architectures such as WideResNet (Zagoruyko & Komodakis, 2016), ResNet (He et al., 2016), and VGG (Simonyan & Zisserman, 2015). We use the standard configurations for the basic training settings shared by SAM and SGD (e.g., learning rate, batch size, and data augmentation) as in the original papers , and set the SAM-specific perturbation radius $\rho$ to 0.05, as recommended by Foret et al. (2021). We also extend our findings to AT (Madry et al., 2018) and ASAM (Kwon et al., 2021). Due to space limits, more experimental details and results can be found in Appendix B.

## 3 SAM EFFICIENTLY SELECTS FLATTER MINIMA LATE IN TRAINING

In this section, we provide empirical evidence indicating that SAM efficiently selects flatter minima over SGD late in training. We first show that applying SAM only during the last few epochs of training achieves generalization comparable to training entirely with SAM. Taking a step further, we demonstrate that last-few-epochs SAM finds flatter minima as effectively as full SAM training.

**Applying SAM in the last few epochs achieves generalization comparable to full SAM training.** We conduct experiments on various datasets and architectures to investigate the effect of applying SAM late in training, after initial training with SGD, on generalization. Specifically, we use the switching method (detailed in Section 2), which switches from SGD to SAM at epoch $t$, to obtain the model $\boldsymbol{\theta}^T_{\text{SGD}\to\text{SAM},t}$. For comparison, we train a baseline model entirely with SAM under the same settings, resulting in $\boldsymbol{\theta}^T_{\text{SAM}}$. Both models are trained for a total of $T$ epochs. To assess the generalization ability of $\boldsymbol{\theta}^T_{\text{SGD}\to\text{SAM},t}$ relative to $\boldsymbol{\theta}^T_{\text{SAM}}$, we calculated the test loss and test error gaps between these two models: $\mathcal{L}_{\mathcal{D}_{\text{test}}}\left(\boldsymbol{\theta}^T_{\text{SGD}\to\text{SAM},t}\right) - \mathcal{L}_{\mathcal{D}_{\text{test}}}\left(\boldsymbol{\theta}^T_{\text{SAM}}\right)$ and $\text{Err}_{\mathcal{D}_{\text{test}}}\left(\boldsymbol{\theta}^T_{\text{SGD}\to\text{SAM},t}\right) - \text{Err}_{\mathcal{D}_{\text{test}}}\left(\boldsymbol{\theta}^T_{\text{SAM}}\right)$, where $\mathcal{D}_{\text{test}}$ denotes the unseen test set. If the generalization gap between $\boldsymbol{\theta}^T_{\text{SGD}\to\text{SAM},t}$ and $\boldsymbol{\theta}^T_{\text{SAM}}$ approaches zero, then we say $\boldsymbol{\theta}^T_{\text{SGD}\to\text{SAM},t}$ achieve nearly the same generalization performance as $\boldsymbol{\theta}^T_{\text{SAM}}$.

Surprisingly, we find that applying SAM only in the final few epochs can achieve nearly the same generalization performance as training entirely with SAM. In particular, we fix the number of total training epochs $T$ and vary the switching point $t$ to adjust the SAM training proportion of $\boldsymbol{\theta}^T_{\text{SGD}\to\text{SAM},t}$, which is exactly $(T-t)/T$. We analyze how the generalization gap between $\boldsymbol{\theta}^T_{\text{SGD}\to\text{SAM},t}$ and $\boldsymbol{\theta}^T_{\text{SAM}}$ changes as the SAM training proportion increases. In Figure 3, the generalization gap becomes

negligible once the SAM training proportion exceeds zero, indicating that even a few epochs of SAM applied at the end of training yield generalization performance comparable to full SAM training. This phenomenon is consistently observed across various datasets and architectures.

A similar phenomenon was noted by Andriushchenko & Flammarion (2022), but it was not given significant attention. We isolate and extend this phenomenon by showing that even fewer epochs of SAM (e.g. 1-5 epochs for WideResNet-16-8 on CIFAR-10) applied in the late phase of training yield test error and loss comparable to full SAM training.

**Applying SAM in the last few epochs finds flatter minima as effectively as full SAM training.** Furthermore, we investigate the effect of applying SAM in the late phase of training on the sharpness of the final solution. Similarly as before, we analyze how the sharpness of the solution $\boldsymbol{\theta}^T_{\text{SGD}\to\text{SAM},t}$ changes w.r.t. the SAM training proportion. We measure the sharpness along the most curved direction of the loss surface, i.e., the spectral norm of the Hessian matrix $\|H(\boldsymbol{\theta})\|_2$. In Figure 3, the sharpness of the solution $\boldsymbol{\theta}^T_{\text{SGD}\to\text{SAM},t}$ dramatically drops once the SAM training proportion is greater than zero, reflecting a similar trend to the generalization gap. This indicates that applying SAM in the last few epochs of training is as effective as training entirely with SAM in achieving flatter minima.

In summary, we see that SAM efficiently selects flatter minima over SGD when applied late in training. This suggests a fine-tuning scheme for SAM: start from a pretrained checkpoint and continue training with SAM for several epochs, reducing computational costs while maintaining similar performance. In Section 4, we will dig deeper into the underlying mechanism behind this phenomenon and gain new insights into SAM's implicit bias towards flatter minima.

## 4 HOW SAM EFFICIENTLY SELECTS FLATTER MINIMA LATE IN TRAINING

We have seen that SAM efficiently selects flatter minima over SGD when applied late in training. In this section, we aim to explore the precise mechanism behind this phenomenon. Theoretically, we identify a two-phase picture in training dynamics after switching to SAM in the late phase, which is characterized by four key claims **(P1-4)**, as outlined in Table 1. This picture demonstrates that SAM provably escapes the minimum found by SGD and converges to a flatter minimum within the same valley in just a few steps, explaining its efficiency in selecting flatter minima late in training.

| Phase I | **(P1).** *SAM rapidly escapes from the minimum found by SGD;* | **Corollary 4.2** |
|---------|----------------------------------------------------------------|-------------------|
| (escape) | **(P2).** *However, the iterator remains within the current valley.* | **Proposition 4.1** |
| Phase II | **(P3).** *SAM selects a flatter minimum compared to SGD;* | **Theorem 4.1** |
| (converge) | **(P4).** *The convergence rate of SAM is extremely fast.* | **Theorem 4.3** |

Table 1: Overview of the two-phase picture and corresponding theoretical results.

To understand the two-phase picture, let us first consider a toy but representative example.

**Example 4.1.** *Consider using the shallow neural network $f(u, v; x) = \tanh(v\tanh(ux))$ to fit a single data point ($x = 1, y = 0$) under the squared loss $\ell(y; y') = (y - y')^2/2$, then the loss landscape can be written as $\mathcal{L}(\boldsymbol{\theta}) = \frac{1}{2}\tanh^2(v\tanh(u))$, where $\boldsymbol{\theta} = (u, v)$.*

Figure 4 (a) visualizes the training dynamics for Example 4.1. In Figure 4 (a), the dynamics occur within the valley surrounding the set of the minima $\mathcal{M} = \{(u, v) : v = 0\}$. On $\mathcal{M}$, the sharpness is given by $\|H(\boldsymbol{\theta})\|_2 = \text{Tr}(H(\boldsymbol{\theta})) = \|H(\boldsymbol{\theta})\|_F = \tanh^2(u)$, implying that the smaller the $|u|$, the flatter the minimum. Initially, SGD converges to a relatively sharp minimum $\boldsymbol{\theta}^{\text{end}}_{\text{SGD}}$. After switching to SAM, the iterator exhibits the two-phase dynamics:

- **Phase I (escape).** SAM first escapes the sharp minimum $\boldsymbol{\theta}^{\text{end}}_{\text{SGD}}$ found by SGD within a few iterations **(P1)**, but remains within the same valley around $\mathcal{M}$[4] **(P2)**;

- **Phase II (converge).** SAM then rapidly converges to another flatter minimum $\boldsymbol{\theta}^{\text{end}}_{\text{SAM}}$ **(P3-4)**, with a smaller $|u|$ compared to $\boldsymbol{\theta}^{\text{end}}_{\text{SGD}}$.

---

[4]Note that the landscape has another valley around a different set of minima $\mathcal{N} = \{(u, v) : u = 0\}$

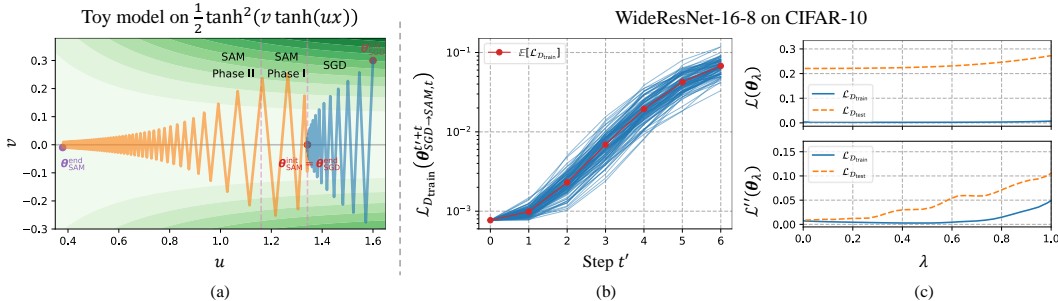

Figure 4: **(a) Visualization of the two-phase dynamics for Example 4.1.** The horizontal gray line represents the set of the global minima $\mathcal{M} = \{(u, v) : v = 0\}$, where smaller values of $|u|$ correspond to flatter minima. The blue lines trace the SGD iterates, leading to $\boldsymbol{\theta}_{\text{SGD}}^{\text{end}}$, while the orange lines show the SAM iterates, which converge to a flatter minimum $\boldsymbol{\theta}_{\text{SAM}}^{\text{end}}$. Notably, $\boldsymbol{\theta}_{\text{SGD}}^{\text{end}}$ and $\boldsymbol{\theta}_{\text{SAM}}^{\text{end}}$ stay in the same valley around $\mathcal{M}$. **(b) The exponentially fast escape from minima found by SGD.** Train loss $\mathcal{L}_{\mathcal{D}_{\text{train}}}(\boldsymbol{\theta}_{\text{SGD}\to\text{SAM},t}^{t+t'})$ v.s. the update step $t'$. Here, $t$ is the switching step, chosen to be sufficiently large for SGD to converge, and $t'$ is the number of updates after switching to SAM. The red line represents the mean over 100 trials with different randomness. **(c) SAM converges to a flatter minimum within the same valley as the one found by SGD.** The loss $\mathcal{L}$ **(top row)** / the second-order finite difference $\mathcal{L}''$ **(bottom row)** of the interpolated model $\boldsymbol{\theta}_\lambda$ v.s. the interpolation coefficient $\lambda$. Here, $\boldsymbol{\theta}_\lambda = (1 - \lambda)\boldsymbol{\theta}_{\text{SGD}\to\text{SAM}}^{\text{end}} + \lambda\boldsymbol{\theta}_{\text{SGD}}^{\text{end}}$, and $\mathcal{L}''(\boldsymbol{\theta}_\lambda) = (\mathcal{L}(\boldsymbol{\theta}_{\lambda+h}) + \mathcal{L}(\boldsymbol{\theta}_{\lambda-h}) - 2\mathcal{L}(\boldsymbol{\theta}_\lambda))/(2h)^2$, where $h$ is fixed to 0.1. This is no barrier in the loss curve, indicating $\boldsymbol{\theta}_{\text{SGD}\to\text{SAM}}^{\text{end}}$ and $\boldsymbol{\theta}_{\text{SGD}}^{\text{end}}$ stay within the same valley. $\mathcal{L}''(\boldsymbol{\theta}_\lambda)$ gradually increases with $\lambda$, implying $\boldsymbol{\theta}_{\text{SGD}\to\text{SAM}}^{\text{end}}$ is flatter than $\boldsymbol{\theta}_{\text{SGD}}^{\text{end}}$.

This toy example successfully illustrates the two-phase picture after switching to SAM in the late training phase, but it offers only a basic intuition. In the following, we will formally present our theoretical results, with each claim rigorously supported by corresponding analysis.

## 4.1 THEORETICAL GUARANTEES FOR **P1** AND **P3**: LINEAR STABILITY ANALYSIS

In this subsection, we provide theoretical support for two key claims in our two-phase picture through a linear stability analysis: **P1**. *SAM escapes from the relatively sharp minima found by SGD exponentially fast*; **P3**. *SAM selects a flatter minimum compared to SGD*.

During the late phase of training, the model remains close to some global minimum $\boldsymbol{\theta}^\star$ and can be locally approximated by its linearization:

$$f_{\text{lin}}(\boldsymbol{x}; \boldsymbol{\theta}) = f(\boldsymbol{x}; \boldsymbol{\theta}^\star) + \langle \nabla f(\boldsymbol{x}; \boldsymbol{\theta}^\star), \boldsymbol{\theta} - \boldsymbol{\theta}^\star \rangle. \tag{5}$$

Consequently, we can characterize the local dynamical stability (Strogatz, 2000) of the SAM iterator near the global minimum on this linearized model.

Similar to the linear stability of SGD (Wu et al., 2022), we define the linear stability of SAM in Definition 4.1. Note that for a specific optimizer, only linearly stable minima can be selected.

**Definition 4.1** (Loss-based linear stability of SAM). A global minimum $\boldsymbol{\theta}^\star$ is said to be linearly stable for SAM if there exists a $C > 0$ such that, when SAM optimizes the loss $\mathcal{L}(\boldsymbol{\theta}) = \frac{1}{2n}\sum_{i=1}^n (f_{\text{lin}}(\boldsymbol{x}_i; \boldsymbol{\theta}) - y_i)^2$ on the linearized model (Equation (5)), the following condition holds: $\mathbb{E}[\mathcal{L}(\boldsymbol{\theta}^t)] \leq C\mathbb{E}[\mathcal{L}(\boldsymbol{\theta}^0)], \forall t \geq 0$. Here, $\boldsymbol{\theta}^0$ could be any point near $\boldsymbol{\theta}^\star$ where the linearization is valid.

Wu et al. (2022) has shown that the linear stability of stochastic algorithms is significantly influenced by the gradient noise $\boldsymbol{\xi}(\boldsymbol{\theta}) = \nabla\ell_\xi(\boldsymbol{\theta}) - \nabla\mathcal{L}(\boldsymbol{\theta})$, where $\xi$ denotes the index of a random sample. Thus, it is crucial to consider the structural properties of the gradient noise in our analysis.

**Gradient noise structure.** A series of studies (Zhu et al., 2018; Feng & Tu, 2021; Wu et al., 2020b; 2022; Mori et al., 2022; Wojtowytsch, 2024) have established two key properties of the gradient noise $\boldsymbol{\xi}(\boldsymbol{\theta})$: (i) its shape is highly anisotropic; (ii) its magnitude is loss dependent. Specifically, by incorporating both the noise shape and magnitude, Mori et al. (2022) has proposed an approximation for the gradient noise covariance $\Sigma(\boldsymbol{\theta}) = \frac{1}{n}\sum_{i=1}^n \boldsymbol{\xi}(\boldsymbol{\theta})\boldsymbol{\xi}(\boldsymbol{\theta})^\top$, suggesting $\Sigma(\boldsymbol{\theta}) \sim 2\mathcal{L}(\boldsymbol{\theta})G(\boldsymbol{\theta})$.

Inspired by this approximation, we assume a weaker alignment property of the noise covariance $\Sigma(\boldsymbol{\theta})$ (see Assumption 4.1) in our analysis, which has been theoretically and empirically validated in recent works (Wu et al., 2022; Wang & Wu, 2023).

**Assumption 4.1** (Gradient noise alignment). *There exists $\gamma > 0$ s.t. $\frac{\text{Tr}(\Sigma(\boldsymbol{\theta})G(\boldsymbol{\theta}))}{2\mathcal{L}(\boldsymbol{\theta})\|G(\boldsymbol{\theta})\|_F^2} \geq \gamma$ for all $\boldsymbol{\theta}$.*

By analyzing the linear stability of SAM under this gradient noise alignment assumption, we establish two theorems supporting the two key claims **P1** and **P3**.

**Theorem 4.1** (**P3**. Proof in Appendix C.1.). *Let $\boldsymbol{\theta}^\star$ be a global minimum that is linearly stable for SAM in Equation (4), and suppose Assumption 4.1 holds. Then we have $\|H(\boldsymbol{\theta}^\star)\|_F^2 \left(1 + \frac{\rho^2\gamma}{B}\|H(\boldsymbol{\theta}^\star)\|_F^2\right) \leq \frac{B}{\eta^2\gamma}$.*

Theorem 4.1 characterizes the sharpness of the linearly stable minima for SAM, which is influenced by the learning rate $\eta$, the batch size $B$, and the perturbation radius $\rho$. Since only linearly stable minima can be selected by optimizer, Theorem 4.1 characterizes the sharpness of global minima selected by SAM. Notably, taking $\rho = 0$ recovers the corresponding result for SGD (see Table 2).

| | SAM (Theorem 4.1) | SGD (Wu et al., 2022) |
|---|---|---|
| Sharpness bound | $\|H(\boldsymbol{\theta}^\star)\|_F^2 \cdot \left(1 + \frac{\rho^2\gamma}{B}\|H(\boldsymbol{\theta}^\star)\|_F^2\right) \leq \frac{B}{\eta^2\gamma}$ | $\|H(\boldsymbol{\theta}^\star)\|_F^2 \leq \frac{B}{\eta^2\gamma}$ |

Table 2: Comparison of the sharpness of global minima selected by SAM and SGD.

**Support for P3.** We compare the bounds on the sharpness of the global minima selected by SAM and SGD in Table 2. The additional term $1 + \rho^2\gamma\|H(\boldsymbol{\theta}^\star)\|_F^2 > 1$ in SAM's bound impose stricter requirement on the sharpness for by SAM compared to SGD, which supports our key claim **P3** that SAM tends to select flatter global minima over SGD. This term arises from SAM's inner update, i.e., $\boldsymbol{\theta}^{t+1/2} = \boldsymbol{\theta}^t + \rho\nabla\ell_{\xi_t}(\boldsymbol{\theta}^t) = \boldsymbol{\theta}^t + \rho\nabla\mathcal{L}(\boldsymbol{\theta}^t) + \rho(\nabla\ell_{\xi_t}(\boldsymbol{\theta}^t) - \nabla\mathcal{L}(\boldsymbol{\theta}^t))$. Intuitively, the ascent update $\rho\nabla\mathcal{L}(\boldsymbol{\theta}^t)$ tends to increase the loss, while the noise $\rho(\nabla\ell_{\xi_t}(\boldsymbol{\theta}^t) - \nabla\mathcal{L}(\boldsymbol{\theta}^t))$ introduces extra stochasticity. These factors together contribute to the flatter-minima selection by SAM.

**Corollary 4.2** (**P1**. Proof in Appendix C.1.). *Let $\boldsymbol{\theta}^\star$ be a global minimum, and suppose Assumption 4.1 holds. If $\|H(\boldsymbol{\theta}^\star)\|_F^2 \left(1 + \frac{\rho^2\gamma}{B}\|H(\boldsymbol{\theta}^\star)\|_F^2\right) > \frac{B}{\eta^2\gamma}$, then $\boldsymbol{\theta}^\star$ is linearly non-stable for SAM in Equation (4). Indeed, $\mathbb{E}[\mathcal{L}(\boldsymbol{\theta}^t)] \geq C^t\mathbb{E}[\mathcal{L}(\boldsymbol{\theta}^0)]$ holds for all $t > 0$ with $C > 1$.*

Corollary 4.2, together with Theorem 4.1, characterizes the necessary condition for a global minimum to be linearly stable for SAM, namely the stability condition of SAM. Furthermore, Corollary 4.2 illustrates that if a global minimum fails to meet the stability condition of SAM, the SAM iterator will escape from the minimum exponentially fast. Notably, such escape occurs within the low-loss regions, where the linear approximation in Equation (5) holds. Complementary analysis of dynamics in the high-loss regions is provided in Proposition 4.1.

**Support for P1.** Although the sharpness bound for SGD in Table 2 is an upper bound, Wu et al. (2022) empirically demonstrates that the bound is nearly tight, with SGD solutions approximately satisfying $\left\|H(\boldsymbol{\theta}_{\text{SGD}}^{\text{end}})\right\|_F^2 \approx \frac{B}{\eta^2\gamma}$[5]. Thus, with the same $\eta$ and $B$, the solution found by SGD cannot satisfy the stability condition of SAM. This leads to our key claim **P1** that SAM escapes from the sharp minima found by SGD exponentially fast, as validated by our experiments in Figure 4 (b).

**Comparison** with other linear stability analyses of SAM (Shin et al., 2023; Behdin et al., 2023). These works have focused on the norm-based linear stability, characterized by $\mathbb{E}[\|\boldsymbol{\theta}^t - \boldsymbol{\theta}^\star\|^2] \leq C\|\boldsymbol{\theta}^0 - \boldsymbol{\theta}^\star\|^2, \forall t \geq 0$, which is stricter than our loss-based stability (see Definition 4.1) due to the presence of degenerate directions in over-parameterized models (see Lemma C.2). Another key difference is the novel use of the alignment property of gradient noise (see Assumption 4.1) in our analysis, enabling us to derive a concrete bound on the sharpness of SAM's solutions.

## 4.2 THEORETICAL GUARANTEE FOR **P2**: BEYOND LOCAL ANALYSIS

In this subsection, we provide theoretical support for another key claim in our two-phase picture: **P2**. *SAM remains within the current valley during escape.*

---

[5]$\approx$ is not used in any formal proofs. Here, it indicates empirical finding that has not been rigorously proven.

**Sub-quadratic landscape.** In high-loss regions, the local linearization in Equation (5) no longer provides a valid approximation for neural networks, and consequently, the *quadratic approximation* of the neural network loss landscape *fails to* capture its true structure. Beyond the quadratic landscape, Ma et al. (2022) has introduced the concept of the *sub-quadratic landscape*: *the loss grows slower than a quadratic function, and the landscape in high-loss regions is significantly flatter than in low-loss regions*. The sub-quadratic landscape has been empirically verified in high-loss regions, and is able to explain the Edge of Stability phenomenon (Cohen et al., 2021) (also observed in Wu et al. (2018); Jastrzebski et al. (2020)), which the quadratic landscape cannot account for.

While Section 4.1 focuses on a local analysis near global minima, the analysis here is necessarily non-local, as SAM might iterate into high-loss regions during escape. In this subsection, we need to show that SAM remains within the current valley during escape under a sub-quadratic landscape.

**Additional setup.** For simplicity, we assume a model size of $p = 1$ and consider the full-batch SAM, i.e., $\theta^{t+1} = \theta^t - \eta \nabla \mathcal{L}(\theta^t + \rho \nabla \mathcal{L}(\theta^t))$. This setup is sufficient to illustrate that SAM will remain within the current valley under a sub-quadratic landscape.

**Definition 4.2** (Sub-quadratic landscape for $p = 1$). Let $\theta^\star = 0$ be a global minimum of $\mathcal{L}$ with sharpness $\mathcal{L}''(0) = a > 0$. The valley $V = [-c, c]$ is termed *sub-quadratic* if for all $z, z_1, z_2 \in V$, it holds that $z\mathcal{L}'(z) \geq 0$ and $\mathcal{L}''(|z_1|) \geq \mathcal{L}''(|z_2|)$ if $|z_1| \leq |z_2|$.

Following Theorem 1 in (Ma et al., 2022), we define the sub-quadratic landscape in Definition 4.2. Specifically, for any $0 < z_1 < z_2 < c$, it holds that $\mathcal{L}''(z_1) > \mathcal{L}''(z_2)$ and $\mathcal{L}(z_1) < \mathcal{L}(z_2)$, indicating that high-loss regions are flatter than low-loss regions. The same applies for $0 > z_1 > z_2 > -c$.

**Proposition 4.1** (**P2**. Proof in Appendix C.2.). *Under Definition 4.2, assume the landscape is sub-quadratic in the valley $V = [-2b, 2b]$. Then, for all initialization $\theta^0 \in (-b, b)$, and $\eta, \rho$ s.t. $\eta < \min_{z \in V} b/|\mathcal{L}'(z)|$, $\rho \leq \min\{1/a, \eta, \eta \min_{0<|z|<b} |\mathcal{L}'(2z)/\mathcal{L}'(z)|\}$, the full-batch SAM (see Equation (10)) will remain within the valley $V$, i.e., $\theta^t \in V$ for all $t \in \mathbb{N}$.*

Proposition 4.1 shows that under a sub-quadratic landscape, even if SAM escapes the minimum $\theta^\star$ when $\eta > 2/a(1 + a\rho)$ and $\theta^0 \approx \theta^\star$, it will still remain within the current valley[6]. This proposition holds when $\eta < \min_{z \in V} b/|\mathcal{L}'(z)|$, where the upper bound $\min_{z \in V} b/|\mathcal{L}'(z)|$ can be sufficiently large under highly sub-quadratic landscapes (see Example C.1). Furthermore, this proposition can be extended to almost all initializations within the valley $V$. See Proposition C.2.

**Support for P2.** Proposition 4.1 supports our key claim **P2** that SAM remains within the current valley during escape. Intuitively, as shown in Section 4.1, the low-loss region around the sharp minimum found by SGD cannot satisfy SAM's stability condition, causing SAM to escape toward higher-loss regions; however, under the sub-quadratic landscape, the high-loss regions are much flatter, which satisfies SAM's stability condition and prevents it from escaping the current valley. The experiment in Figure 4 (c) backs up this theoretical claim.

### 4.3 THEORETICAL GUARANTEE FOR P4: CONVERGENCE ANALYSIS

In this subsection, we provide theoretical support for the final claim in our two-phase picture: **P4.** *The convergence rate of SAM is significantly fast.*

To establish the convergence results for stochastic optimization algorithms, it is often necessary to make assumptions about the loss landscape and the magnitude of gradient noise $\xi(\theta)$.

**Assumption 4.2** (Loss landscape: $L$-smoothness and Polyak–Łojasiewicz). (i) There exists $L > 0$ s.t. $\|H(\theta)\|_2 \leq L$ for all $\theta$; (ii) There exists $\mu > 0$ s.t. $\|\nabla \mathcal{L}(\theta)\|^2 \geq 2\mu \mathcal{L}(\theta)$ for all $\theta$.

Assumption 4.2 are quite standard and is frequently used in the non-convex optimization literature (Karimi et al., 2016). Specifically, for neural networks, if $\mathcal{L}(\cdot)$ is four-times continuously differentiable, this assumption holds at least near the set of global minima of $\mathcal{L}$ (Arora et al., 2022).

**Assumption 4.3** (Gradient noise magnitude). There exists $\sigma > 0$ s.t. $\mathbb{E}[\|\xi(\theta)\|^2] \leq \sigma^2 \mathcal{L}(\theta), \forall \theta$.

The bounded variance assumption $\mathbb{E}[\|\xi(\theta)\|^2] \leq \sigma^2$ is commonly employed in classical optimization theory (Bubeck et al., 2015). However, as discussed in Section 4.1, recent studies (Mori et al., 2022;

---

[6]Multiple valleys exist in the settings of Proposition 4.1 (see Remark C.3 for more details).

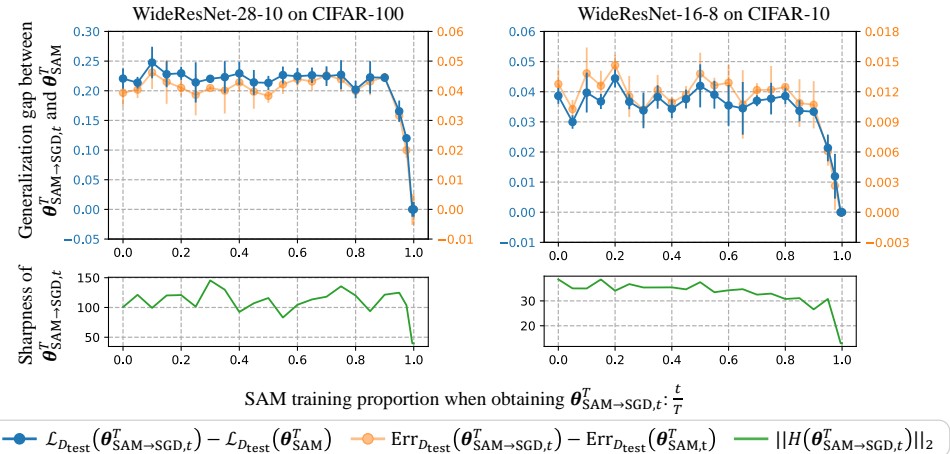

Figure 5: **The impact of SAM training proportion on generalization/sharpness when switching from SAM to SGD.** The generalization gap between the models $\boldsymbol{\theta}_{\text{SAM}\rightarrow\text{SGD},t}^T$ and $\boldsymbol{\theta}_{\text{SAM}}^T$ (**top row**)/the sharpness of $\boldsymbol{\theta}_{\text{SAM}\rightarrow\text{SGD},t}^T$ (**bottom row**) vs. the SAM training proportion of $\boldsymbol{\theta}_{\text{SAM}\rightarrow\text{SGD},t}^T$. Dots represent the mean over three trials with different random seeds, and error bars indicate standard deviations. Results on more datasets and architectures can be found in Appendix B.5.

Feng & Tu, 2021; Liu et al., 2021; Wojtowytsch, 2024) have shown that the magnitude of SGD noise is also bounded by the loss value. Thus, we introduce Assumption 4.3: $\mathbb{E}[\|\boldsymbol{\xi}(\boldsymbol{\theta})\|^2] \leq \sigma^2 \mathcal{L}(\boldsymbol{\theta})$. This indicates that the noise magnitude diminishes to zero at global minima, contrasting sharply with the classical bounded variance assumption and closely aligning with real-world scenarios.

Under these assumptions, we present the convergence result for stochastic SAM in Theorem 4.3.

**Theorem 4.3** (**P4**. Proof in Appendix C.3.). *Under Assumption 4.2 and 4.3, let $\{\boldsymbol{\theta}^t\}_t$ be the parameters trained by SAM in Equation* (4). *If $\eta \leq \min\{1/2L, \mu B/2L\sigma^2\}$ and $\rho \leq \min\left\{1/4L, \mu B/4L\sigma^2, \eta\mu^2/24L^2\right\}$, then we have $\mathbb{E}\left[\mathcal{L}(\boldsymbol{\theta}^t)\right] \leq (1 - \eta\mu/2)^t \mathcal{L}(\boldsymbol{\theta}^0), \forall t \in \mathbb{N}$.*

**Support for P4.** Theorem 4.3 supports our key claim **P4** that the convergence rate of stochastic SAM is significantly fast, and notably faster than the previous result on SAM's convergence rate (Andriushchenko & Flammarion, 2022).

**Comparison** with other convergence results for stochastic SAM (Andriushchenko & Flammarion, 2022; Shin et al., 2023). These works, along with our result, all study the convergence on stochastic SAM under the standard Assumption 4.2. Andriushchenko & Flammarion (2022) has shown that stochastic SAM converges under a polynomially fast rate of $\mathcal{O}(1/t)$, but this is much slower than our exponentially fast rate. The difference lies in the assumptions on gradient noise magnitude: they have used the classical assumption $\mathbb{E}[\|\boldsymbol{\xi}(\boldsymbol{\theta})\|^2] \leq \sigma^2$, whereas we rely on the more realistic Assumption 4.3. Shin et al. (2023) has also proved that stochastic SAM converges exponentially fast. However, their analysis is based on an additional interpolation assumption due to over-parameterization.

## 5  IS SAM NECESSARY IN THE EARLY PHASE OF TRAINING?

Since our main discovery in Section 3, one might question the necessity of using SAM in the early phase. In this section, we demonstrate that applying SAM during the early phase offers only marginal benefits for generalization and sharpness compared to full SGD training. Thus, we conjecture that the optimization algorithm chosen at the end of training is more critical in shaping the final solution's properties. Based on this viewpoint, we extend our findings from SAM to Adversarial Training.

**Early-phase SAM may offer limited improvements over SGD in generalization and sharpness.** We conduct experiments under various settings to investigate the effect of applying SAM during the early phase of training, followed by SGD, on generalization and sharpness. Similarly to Section 3, we use the switching method but switch from SAM to SGD, to obtain the model $\boldsymbol{\theta}_{\text{SAM}\rightarrow\text{SGD},t}^T$. In Figure 5, the generalization gap remains substantial even when trained with SAM for most of the time (e.g., $t/T = 0.8$). A similar trend is observed in the sharpness of $\boldsymbol{\theta}_{\text{SAM}\rightarrow\text{SGD},t}^T$. Only full SAM training (i.e.,

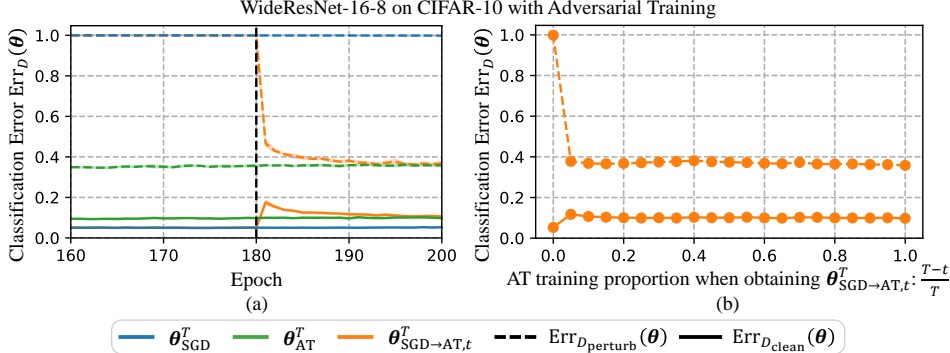

Figure 6: **AT improves robustness efficiently even when applied only during the final few epochs of training**. **(a)** Robust/natural error vs. training epochs for models trained with different strategies. **(b)** Robust/natural error of $\boldsymbol{\theta}^T_{\text{SGD}\to\text{AT}}$ vs. the AT training proportion $(T-t)/T$. Dashed lines denotes the robust error $\text{Err}_{\mathcal{D}_{\text{perturb}}}$, and solid lines denotes the natural error $\text{Err}_{\mathcal{D}_{\text{clean}}}$. Blue and green curves represent models trained solely with SGD and AT, i.e., $\boldsymbol{\theta}^T_{\text{SGD}}$ and $\boldsymbol{\theta}^T_{\text{AT}}$, respectively. Orange curves indicate models which switching from SGD to AT at epoch $t$, i.e., $\boldsymbol{\theta}^T_{\text{SGD}\to\text{AT},t}$. All results are averaged over five trials with different randomness.

$t/T = 1$) consistently finds the flat minima. This suggests that applying SAM before SGD could yield only limited improvements in generalization and sharpness compared to full SGD training.

Together with findings in Section 3, we conjecture that the optimization algorithm chosen in the late training phase is more critical in determining the final solution's properties. However, current findings are based solely on generalization/sharpness—can we extend to other properties?

**Extension: Adversarial Training efficiently finds robust minima late in training.** Adversarial Training (AT) (Madry et al., 2018), in which networks are trained on adversarial examples (Szegedy et al., 2014), significantly improves the robustness of neural networks. Recently, Zhang et al. (2024) highlighted a duality between SAM and AT: while SAM perturbs the model parameters during training (see Equation (1)), AT perturbs input samples, with both methods relying on gradients for perturbations. Here, we pose the question: can our findings on SAM be extended to AT?

Similar to SAM, we discover that applying AT only during the final few epochs yields robustness comparable to training entirely with AT. With the switching method, which switches from SGD to AT at epoch $t$, we obtain the model $\boldsymbol{\theta}^T_{\text{SGD}\to\text{AT},t}$. We evaluate its robustness by classification error on adversarial examples, termed robust error $\text{Err}_{\mathcal{D}_{\text{perturb}}}(\boldsymbol{\theta})$, and its performance on clean examples, referred to as natural error $\text{Err}_{\mathcal{D}_{\text{clean}}}(\boldsymbol{\theta})$. In Figure 6 (a), the robust error of $\boldsymbol{\theta}^T_{\text{SGD}\to\text{AT},t}$ drops sharply after switching to AT at $180$ epoch, matching that of full AT training, while the natural error only slightly spikes. Additionally, we vary the switching point $t$ to examine how the robustness of $\boldsymbol{\theta}^T_{\text{SGD}\to\text{AT},t}$ changes with the increasing proportion of AT epochs, i.e., $(T-t)/T$. In Figure 6 (b), the robust error of $\boldsymbol{\theta}^T_{\text{AT}}$ is comparable to that of models trained entirely with AT (i.e., when $(T-t)/T = 1$) once the proportion of AT epochs exceeds 0. These results together confirm that AT efficiently enhances the robustness of neural networks late in training.

## 6 CONCLUSION AND LIMITATIONS

In conclusion, we discovered an intriguing phenomenon that SAM efficiently selects flatter minima over SGD when applied late in training. We provided a theoretical explanation for this by conducting a two-phase analysis of the learning dynamics after switching to SAM in the late training phase. Our results suggest that late training dynamics may play an important role in understanding the implicit bias of optimization algorithms. Since we extended our findings to AT with the switching method, a natural future direction is to generalize this novel approach to other optimization algorithms.

**Limitations.** We note that even if our theoretical setup closely reflects real settings, it still relies on several simplifications, such as assuming a constant perturbation radius $\rho$ and using squared loss. We also note that our theory only explains the behavior of SAM in the late training phase; it cannot account for the phenomena observed for early-phase SAM.

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

# A  RELATED WORK

**Implicit flat-minima bias.** Extensive studies have been dedicated to understanding the implicit bias in deep learning (Vardi, 2023). One notable example is the *flat-minima bias*: SGD and its variants tend to select flat minima (Keskar et al., 2017; Zhang et al., 2017), which often generalize better (Hochreiter & Schmidhuber, 1997; Jiang et al., 2020). Many works (Ma & Ying, 2021; Mulayoff et al., 2021; Wu & Su, 2023; Gatmiry et al., 2023; Wen et al., 2023b) have theoretically justified the superior generalization of flat minima in neural networks. Wu et al. (2018; 2022); Ma & Ying (2021) have taken a dynamical stability perspective to explain why SGD favors flat minima; in-depth analysis of SGD dynamics near global minima shows that the SGD noise drives SGD towards flatter minima (Blanc et al., 2020; Li et al., 2022; Ma et al., 2022; Damian et al., 2021). Further studies (Jastrzębski et al., 2017; Lyu et al., 2022) have explored how training components, like learning rate and normalization, influence the flat-minima bias of SGD.

**Understandings towards SAM.** Inspired by the implicit flat-minima bias of SGD, Foret et al. (2021) has proposed Sharpness-Aware Minimization (SAM) algorithm. SAM and its variants (Kwon et al., 2021; Zhuang et al., 2022; Du et al., 2022; Liu et al., 2022; Kim et al., 2022; Mueller et al., 2023; Becker et al., 2024; Wang et al., 2024) have achieved superior performance across a wide range tasks (Chen et al., 2022; Kaddour et al., 2022; Bahri et al., 2022). Despite their successes, their effectiveness remains poorly understood. *On the optimization front*, Andriushchenko & Flammarion (2022); Si & Yun (2023); Shin et al. (2023) have studied the convergence properties of stochastic SAM; Monzio Compagnoni et al. (2023); Kim et al. (2023) have demonstrated that SAM requires more time to escape saddle points compared to SGD; Long & Bartlett (2024) have analyzed the edge of stability phenomenon for SAM. *On the implicit bias side*, Shin et al. (2023); Behdin et al. (2023) have examined SAM's flat-minima bias using dynamical stability analysis; Wen et al. (2023a); Monzio Compagnoni et al. (2023); Bartlett et al. (2023) have studied how SAM moves towards flat minima; Andriushchenko & Flammarion (2022) has showed that SAM has the low-rank bias; and Wen et al. (2023b) has revealed that SAM does not only minimize sharpness to achieve better generalization. Dai et al. (2023) has characterized the role of Normalization in SAM. *In comparison, our work* focuses on the dynamics of SAM in the late training phase, a topic largely unattended in the literature, offering both theoretical and empirical analyses of SAM's convergence, escape behavior, and bias toward flatter minima. Additionally, several studies have explored the connection between SAM and other algorithms: Zhang et al. (2024) explored the duality between SAM and AT; Zhu et al. (2023) demonstrated the equivalence between average-direction SAM and decentralized SGD. Another related work is Jia & Su (2020), which introduces a sharpness-regularized training algorithm with a formulation similar to SAM. Notably, its ablation study demonstrates the algorithm's effectiveness when applied in the mid or late training stages. Karakida et al. (2023) has analyzed the implicit bias of Gradient Regularization (GR), which shares a similar form with SAM, and showed that GR tends to select solutions in rich regimes.

## B    MORE EXPERIMENTAL DETAILS AND RESULTS

### B.1    DETAILED EXPERIMENTAL SETTINGS

In this subsection, we introduce the detailed experimental settings.

In this work, we mainly compare the training dynamics of SAM and SGD. In fact, SAM and SGD share almost identical training settings, with the only difference lying in that SAM introduces an extra hyper-parameter, perturbation radius $\rho$. Therefore, we first outline the basic training settings for training with SGD across different network architectures and datasets.

**VGG-16 and ResNet-20 on the CIFAR-10 dataset.** We train the VGG-16 (Simonyan & Zisserman, 2015) and the ResNet-20 architecture (He et al., 2016) on the CIFAR-10 dataset (Krizhevsky et al., 2009), using standard data augmentation and hyperparameters as in the original papers (Simonyan & Zisserman, 2015; He et al., 2016). Data augmentation techniques include random horizontal flips and random $32 \times 32$ pixel crops. A weight decay of $5 \times 10^{-4}$ is applied, and the momentum for gradient update is set to 0.9. The learning rate is initialized at 0.1 and is dropped by 10 times at epoch 80[7]. The total number of training epochs is 160.

**WideResNet-16-8 on the CIFAR-10 dataset, and WideResNet-28-10 on the CIFAR-100 dataset.** We train the WideResNet-16-8 architecture (Zagoruyko & Komodakis, 2016) on the CIFAR-10 dataset and the WideResNet-28-10 architecture on the CIFAR-100 dataset (Krizhevsky et al., 2009), following the standard settings as in the original papers (Zagoruyko & Komodakis, 2016). Data augmentation techniques include random horizontal flips, random $32 \times 32$ pixel crop, and cutout regularization (DeVries, 2017). A weight decay of $5 \times 10^{-4}$ is applied, and the momentum for gradient update is set to 0.9. The learning rate is initialized at 0.1 and is dropped by 5 times at 60 and 120 epochs. The total number of training epochs is 200.

Then, we specify the extra hyper-parameter for SAM, i.e., the perturbation radius $\rho$.

**Additional setup for SAM, ASAM and USAM.** For SAM, the perturbation radius is set to 0.05 across all architectures and datasets, as recommended by Foret et al. (2021). In Appendix B.3, our findings on SAM generalize to another SAM variant, ASAM (Kwon et al., 2021). For ASAM, the perturbation radius is set to 2.0, as recommended by Kwon et al. (2021). In Appendix B.4, we further generalize our findings to USAM (Andriushchenko & Flammarion, 2022) to verify the simplified SAM update rule used in our theoretical analyses. For USAM, the perturbation radius is set to 0.1.

In Section 5, we also extend our findings to Adversarial Training (AT). AT shares almost the same settings as SGD as well, except that AT trains neural networks on adversarial examples generated from the original images. Here, we specify the additional settings for creating adversarial examples.

**Additional setup for AT.** We considered untargeted PGD attack (Madry et al., 2018) to generate adversarial examples $\boldsymbol{x}^{\text{Adv}}$ from the original image $\boldsymbol{x}$, with the constraint that $\left\|\boldsymbol{x}^{\text{Adv}} - \boldsymbol{x}\right\|_{\infty} \leq \epsilon$, where $\|\cdot\|_{\infty}$ denotes the $L^{\infty}$ norm. Following the standard configurations of Madry et al. (2018), we set the perturbation limit $\epsilon$ to $4/255$ and the step size for PGD attack to $2/255$.

**Additional setup for experiments on ViTs.** In Figure 8, we train the ViT-T/S architecture (Dosovitskiy et al., 2021) on the CIFAR-100 dataset, following the settings in Mueller et al. (2023). Data augmentation techniques include random horizontal flips, random $32 \times 32$ pixel crops and AutoAugment. The early phase optimization is done with AdamW Kingma (2014), with a constant learning rate $1 \times 10^{-4}$. A weight decay of $5 \times 10^{-4}$ is applied, and the batch size is set to 64. The late phase optimization is done with SAM, with an extra hyperparameter, the perturbation radius, set to 0.1. The total number of training epochs is 200.

### B.2    MORE EXPERIMENTS ON LATE-PHASE SAM

In this subsection, we provide more experimental results under various settings to demonstrate the effect of SAM on generalization and sharpness when applied during the late phase of training.

---

[7]Note that in the original paper, the learning rate is reduced by a factor of 10 at epoch 120 as well. However, based on our theory in Section 4.1, the learning rate affects the solution sharpness (see Table 2). To fairly compare the sharpness of solutions found by the switching method, which switches at different epochs, we remove the learning rate decay during the late phase of training. The same applies for other architectures and datasets.

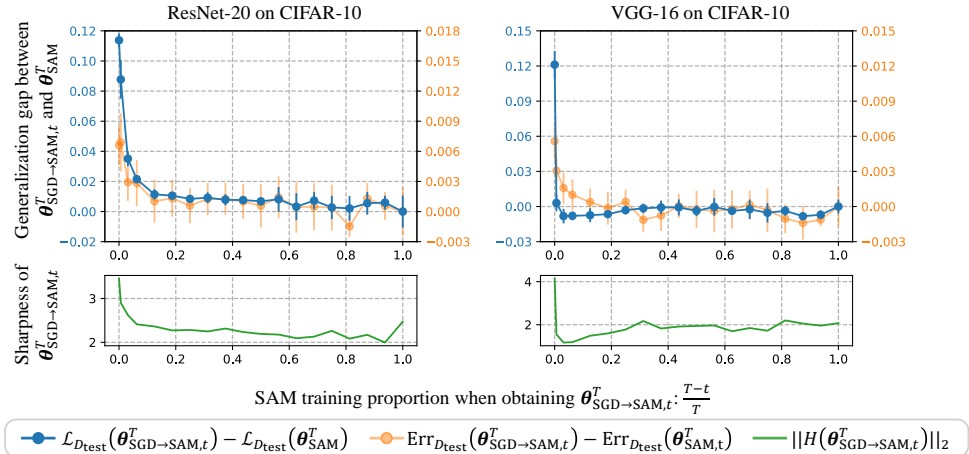

Figure 7: **Additional experiments for VGG-16 and ResNet-20 on CIFAR-10: the impact of SAM training proportion on generalization/sharpness when switching from SGD to SAM.** The generalization gap between the models $\theta_{\text{SGD}\to\text{SAM},t}^T$ and $\theta_{\text{SAM}}^T$ **(top row)** / the sharpness of $\theta_{\text{SGD}\to\text{SAM},t}^T$ **(bottom row)** vs. the SAM training proportion of $\theta_{\text{SGD}\to\text{SAM},t}^T$. Dots represent the mean over five trials with different random seeds, and error bars indicate standard deviations.

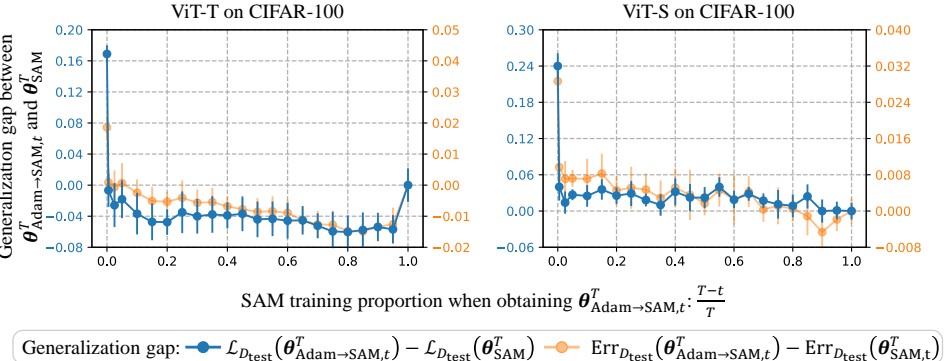

Figure 8: **Additional experiments for ViTs on CIFAR-100: the impact of SAM training proportion on generalization when switching from Adam to SAM.** The generalization gap between the models $\theta_{\text{Adam}\to\text{SAM},t}^T$ and $\theta_{\text{SAM}}^T$ vs. the SAM training proportion of $\theta_{\text{Adam}\to\text{SAM},t}^T$. Dots represent the mean over five trials with different random seeds, and error bars indicate standard deviations.

In Figure 7, we include more experimental results for ResNet-20 and VGG-16 on the CIFAR-10 datasets.

In Figure 8, we include experiment results for ViT-T/S on the CIFAR-100 datasets. Notice that due to an implementation limitation in PyTorch, computing the gradient twice consecutively for ViTs is not feasible. Thus, only generalization results are presented. Additionally, as Adam is a more common practice for training ViTs, we replace SGD with Adam before the switch.

In Figure 9, we include further experimental results using the squared loss. In Figure 9, both the generalization gap and sharpness follow trends consistent with our previous findings in Figure 3, thereby bridging the gap between our theoretical and experimental settings.

## B.3 EXTENDED EXPERIMENTS ON ASAM

In this subsection, we generalize our findings on SAM to another SAM variant, ASAM (Kwon et al., 2021). Similar to SAM, we find that applying ASAM during the late training phase, after initial training with SGD, achieves generalization ability and solution sharpness comparable to full ASAM training. Still, we apply the switching method to investigate the effect of the late-phase ASAM

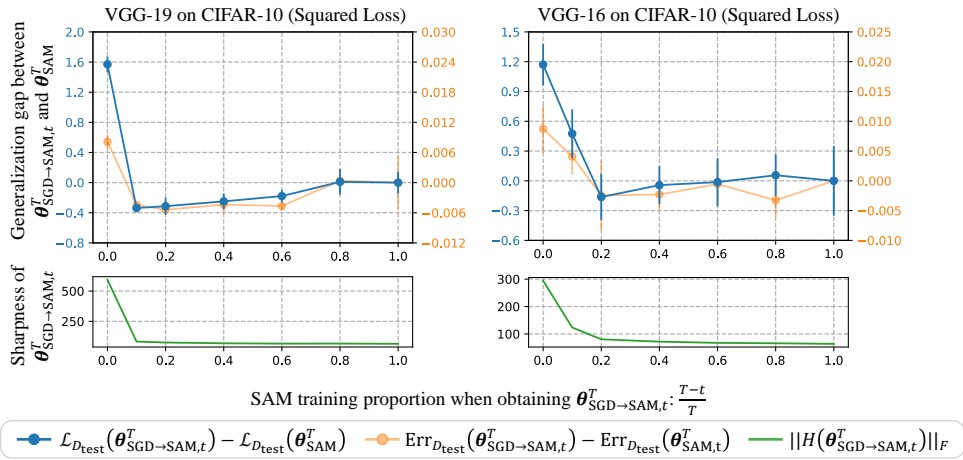

Figure 9: **Additional experiments using squared loss: the impact of SAM training proportion on generalization when switching from SGD to SAM.** The generalization gap between the models $\theta^T_{\text{SGD}\to\text{SAM},t}$ and $\theta^T_{\text{SAM}}$ vs. the USAM training proportion of $\theta^T_{\text{SGD}\to\text{SAM},t}$. Dots represent the mean over three trials with different random seeds, and error bars indicate standard deviations.

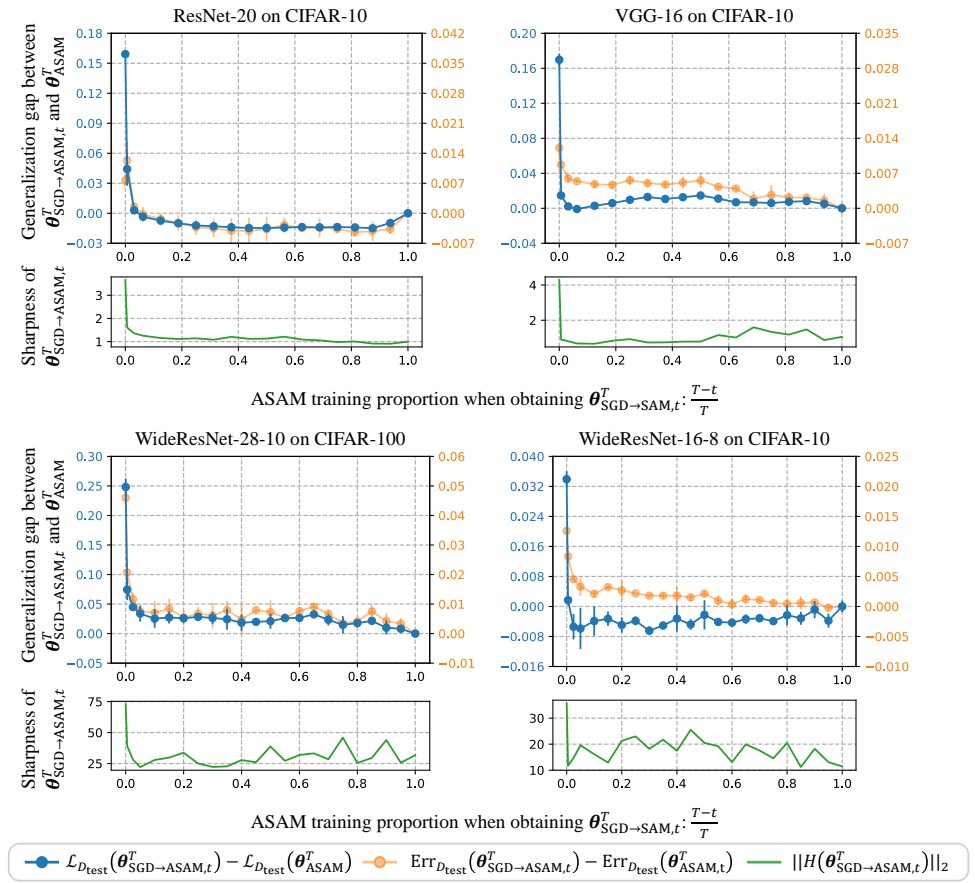

Figure 10: **The impact of ASAM training proportion on generalization/sharpness when switching from SGD to ASAM.** The generalization gap between the models $\theta^T_{\text{SGD}\to\text{ASAM},t}$ and $\theta^T_{\text{ASAM}}$ **(top row)** / the sharpness of $\theta^T_{\text{SGD}\to\text{ASAM},t}$ **(bottom row)** vs. the ASAM training proportion of $\theta^T_{\text{SGD}\to\text{ASAM},t}$. Dots represent the mean over three trials with different random seeds, and error bars indicate standard deviations.

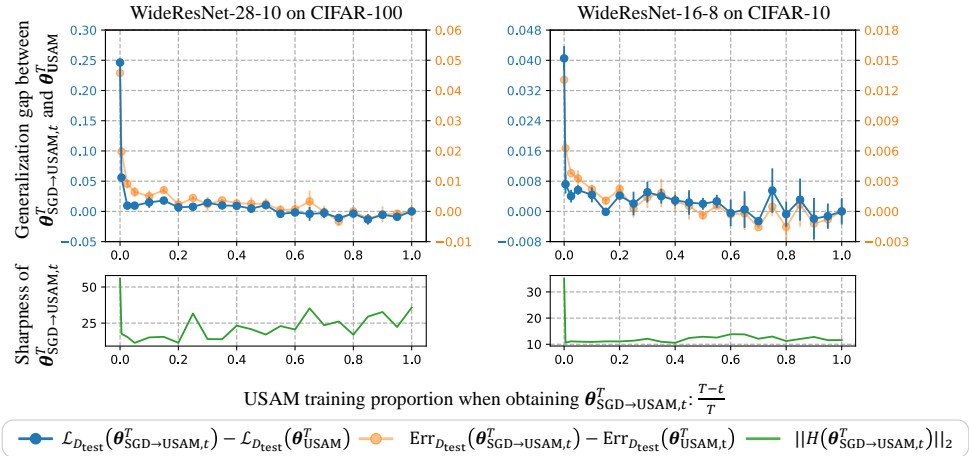

Figure 11: **The impact of USAM training proportion on generalization when switching from SGD to USAM.** The generalization gap between the models $\boldsymbol{\theta}^T_{\text{SGD}\to\text{USAM},t}$ and $\boldsymbol{\theta}^T_{\text{USAM}}$ vs. the USAM training proportion of $\boldsymbol{\theta}^T_{\text{SGD}\to\text{USAM},t}$. Dots represent the mean over three trials with different random seeds, and error bars indicate standard deviations.

across various architectures and datasets. We use the switching method, which switches from SGD to ASAM, to obtain the model $\boldsymbol{\theta}^T_{\text{SGD}\to\text{ASAM},t}$. We also train a baseline model, i.e., $\boldsymbol{\theta}^T_{\text{SGD}\to\text{SAM},t}$. We vary the switching point $t$ while keeping $T$ fixed to adjust the proportion of ASAM training and study its impact on the generalization gap between $\boldsymbol{\theta}^T_{\text{ASAM}}$ and $\boldsymbol{\theta}^T_{\text{SGD}\to\text{ASAM},t}$, as well as the sharpness of $\boldsymbol{\theta}^T_{\text{SGD}\to\text{ASAM},t}$. As shown in Figure 10, both the generalization gap and the sharpness drop once the ASAM training proportion exceeds zero, implying that ASAM, even when applied only during the last few epochs, achieves comparable generalization performance and solution sharpness as training entirely with ASAM. These results further highlight the importance of studying the implicit bias of different optimization algorithms towards the end of training.

## B.4  EXTENDED EXPERIMENTS ON USAM

In this subsection, we generalize our findings on SAM to another SAM variant, USAM (Andriushchenko & Flammarion, 2022), to further justify the simplification made on the SAM update rule in our theoretical analysis. With a small modification to Equation (3) , the update rule of USAM with stochastic gradients is:

$$\boldsymbol{\theta}^{t+1} = \boldsymbol{\theta}^t - \eta \nabla \mathcal{L}_{\xi_t}\left(\boldsymbol{\theta}^t + \rho \nabla \mathcal{L}_{\xi_t}(\boldsymbol{\theta}^t)\right). \tag{6}$$

Consistently with the main paper, we measure how the generalization gap between $\boldsymbol{\theta}^T_{\text{SGD}\to\text{USAM},t}$ and $\boldsymbol{\theta}^T_{\text{USAM}}$, as well as the sharpness of $\boldsymbol{\theta}^T_{\text{SGD}\to\text{USAM},t}$, changes as the USAM training proportion $\frac{T-t}{T}$ increases. In Figure 11, both the generalization gap and sharpness drop sharply once the USAM training proportion exceeds zero. This finding validates the effectiveness of USAM in capturing the main behavior of SAM, further bridging the gap between our theoretical results and empirical findings.

## B.5  MORE EXPERIMENTS ON EARLY-PHASE SAM

In this subsection, we provide more experimental results to demonstrate the effect of SAM on generalization and sharpness when applied during the early phase of training.

**The effect of early-phase SAM on the generalization ability.** In Figure 12, we include more experimental results for ResNet-20 and VGG-16 on the CIFAR-10 datasets. Similar to our results in Figure 5, the generalization gap between $\boldsymbol{\theta}^T_{\text{SAM}\to\text{SGD},t}$ and $\boldsymbol{\theta}^T_{\text{SAM}}$ remains substantial until the SAM training proportion exceeds 0.6. The same applies for the sharpness of $\boldsymbol{\theta}^T_{\text{SAM}\to\text{SGD},t}$. These results further validate that applying SAM only during the early phase offers limited improvements on generalization and sharpness of the final solution compared to full SAM training.

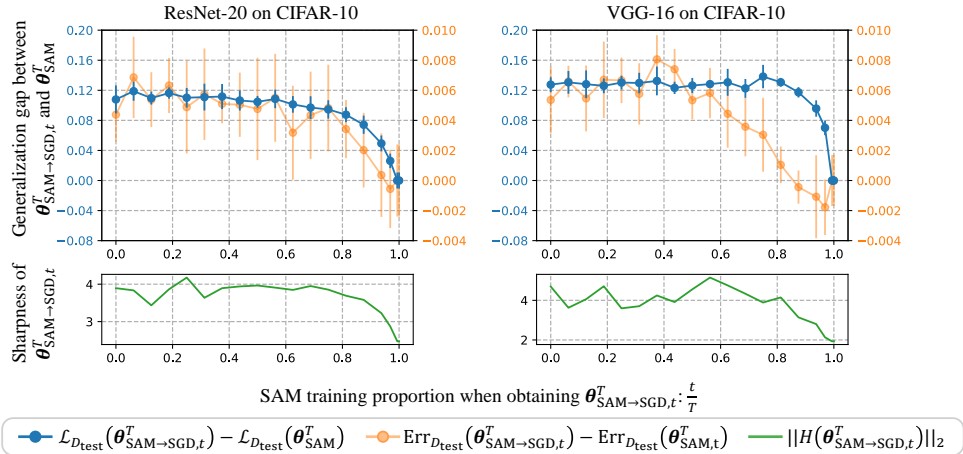

Figure 12: **Additional experiments for VGG-16 and ResNet-20 on CIFAR-10: the impact of SAM training proportion on generalization/sharpness when switching from SAM to SGD.** The generalization gap between the models $\theta^T_{\text{SAM}\to\text{SGD},t}$ and $\theta^T_{\text{SAM}}$ (**top row**)/the sharpness of $\theta^T_{\text{SAM}\to\text{SGD},t}$ (**bottom row**) vs. the SAM training proportion of $\theta^T_{\text{SAM}\to\text{SGD},t}$. Dots represent the mean over five trials with different random seeds, and error bars indicate standard deviations.

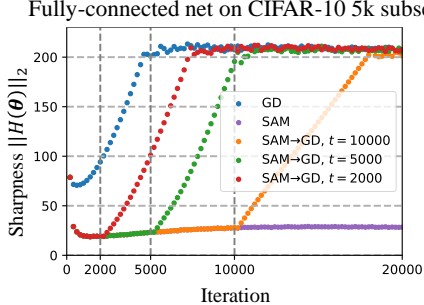

Figure 13: **The impact of (early-phase) SAM on the sharpness evolution.** The sharpness evolution curves for fully-connected nets on CIFAR-10 5k subset trained with different strategies. The blue and purple baseline curves represents training entirely with GD and (full-batch) SAM, respectively. The red, green and orange curves represent the switching method, which switches from GD to (full-batch) SAM at different switching point $t$, with $t \in \{2000, 5000, 10000\}$. A constant learning rate $\eta = 0.01$ ($2/\eta = 200$) is used for both GD and (full-batch) SAM.

**The effect of early-phase SAM on the sharpness evolution.** Other than focusing on the sharpness of the final solution, we are also interested in the effect of early-phase SAM on the evolution process of the sharpness.

Furthermore, we explore how sharpness at the iterator evolves when switching from SAM to (S)GD. Recent works (Cohen et al., 2021; Ahn et al., 2022) have shown that when training neural networks with Gradient Descent (GD), the sharpness $\|H(\theta)\|_2$ tends to increase monotonically until it arrives at $2/\eta$, where $\eta$ is the learning rate. This is commonly known as the *Edge of Stability (EoS)* phenomenon[8]. Specifically, we follow the setup of Cohen et al. (2021), measuring sharpness evolution for models trained with the switching method, which switches from (full-batch) SAM[9] to GD, as well as models trained entirely with GD and SAM. In Figure 13, we observe the EoS phenomenon for models trained entirely with GD. However, for models trained entirely with SAM, sharpness initially decreases, then slightly increases before converging to a value much smaller than $2/\eta$. As for models trained using the switching method, sharpness evolution mirrors that of full SAM training up to the switching point $t$. After $t$, the sharpness rapidly increases and then oscillates around

---

[8]Despite Long & Bartlett (2024) noted an "edge of stability" for SAM, here, the Edge of Stability phenomenon refers specifically to GD.

[9]In this experiment, we adopt the SAM with full-batch gradient for fair comparison with GD.

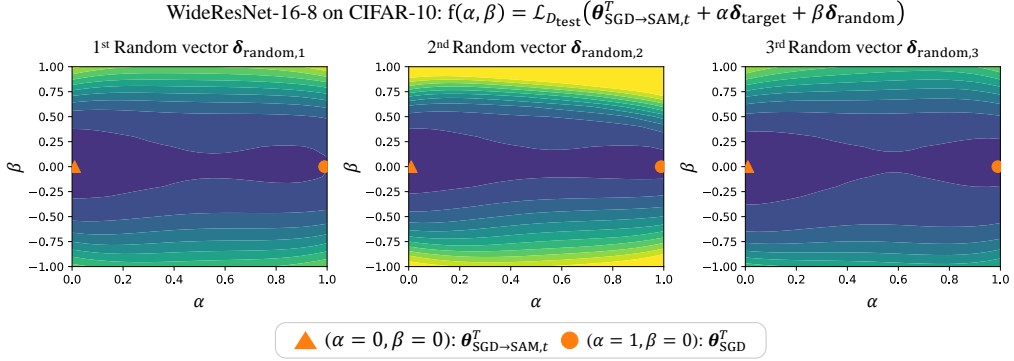

Figure 14: **A 2D visualization of the loss landscape around SGD and SGD-to-SAM solutions.** The orange dot ($\alpha = 1, \beta = 0$) denotes the SGD solution, and the orange triangle denotes ($\alpha = 0, \beta = 0$) denotes the solution obtained by switching from SGD to SAM. The contour plot represents the loss landscape, with darker regions indicating lower loss. The random vectors are sampled three times.

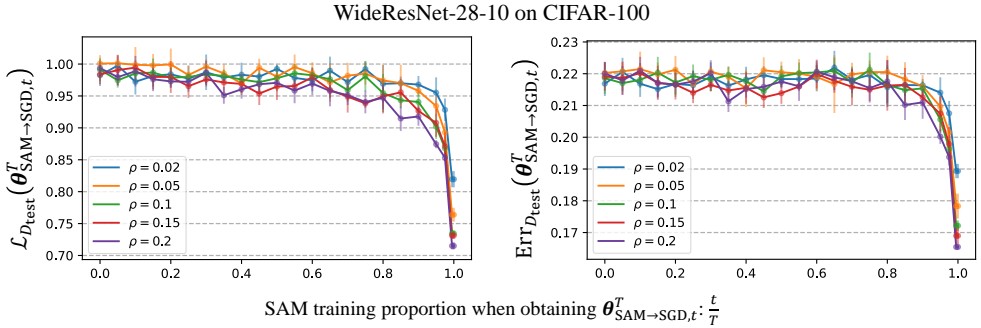

Figure 15: **The impact of perturbation radius on generalization when switching from SAM to SGD.** The test loss **(left)** / test error (right) of the model $\boldsymbol{\theta}_{\text{SAM}\to\text{SGD},t}^T$ vs. the SAM training proportion. Different perturbation radius $\rho \in \{0.02, 0.05, 0.1, 0.15, 0.2\}$ are chosen for SAM. Dots represent the mean over five trials with different random seeds, and error bars indicate standard deviations.

$2/\eta$, regardless of how late the switch occurs. This implies that early-phase SAM cannot prevent the emergence of the EoS phenomenon. Once switching to GD, the iterator immediately escapes the flat minima found by SAM, resulting in relatively sharper minima. Together with our findings in Figures 5 and 12, we firmly say that early-phase SAM has a limited impact on both generalization and sharpness of the final solution.

## B.6 OTHER EXPERIMENTS

**A 2D visualization of loss landscape** Other than the 1D interpolation in Figure 4 (c), we provide 2D visualization of the loss landscape, using the approach proposed by Li et al. (2018). In this setup, we center the visualization at $\boldsymbol{\theta}_{\text{SGD}\to\text{SAM},t}^T$ and select two directions: $\boldsymbol{\delta}_{\text{target}}$ and $\boldsymbol{\delta}_{\text{random}}$. Specifically, $\boldsymbol{\delta}_{\text{target}} = \boldsymbol{\theta}_{\text{SGD}}^T - \boldsymbol{\theta}_{\text{SGD}\to\text{SAM},t}^T$, pointing towards the SGD solution, and $\boldsymbol{\delta}_{\text{random}}$ is a random direction sampled from $\mathcal{N}(0, \boldsymbol{I}_p)$, with layer-wise normalization (Li et al., 2018) applied. We then plot the loss landscape as a function $f(\alpha, \beta) = \mathcal{L}(\boldsymbol{\theta}_{\text{SGD}\to\text{SAM},t}^T + \alpha\boldsymbol{\delta}_{\text{target}} + \beta\boldsymbol{\delta}_{\text{random}})$. In Figure 14, it is clear that $\boldsymbol{\theta}_{\text{SGD}\to\text{SAM},t}^T$ and $\boldsymbol{\theta}_{\text{SGD}}^T$ stay within the same valley. Moreover, the contours around $\boldsymbol{\theta}_{\text{SGD}\to\text{SAM},t}^T$ are wider compared to $\boldsymbol{\theta}_{\text{SGD}}^T$, indicating $\boldsymbol{\theta}_{\text{SGD}\to\text{SAM},t}^T$ corresponds to a flatter minimum. This two-dimensional approach provides additional insights into the structure of the loss landscape and further supports our two key claims **P2** and **P3**.

**The effect of perturbation radius on generalization when switching from SAM to SGD** In our theory (see Table 2), the perturbation radius $\rho$ influences the sharpness of the global minima selected by SAM. Specifically, a larger perturbation radius leads to the selection of flatter global minima. In contrast, SGD's behavior is independent of the perturbation radius. However, as discussed in our

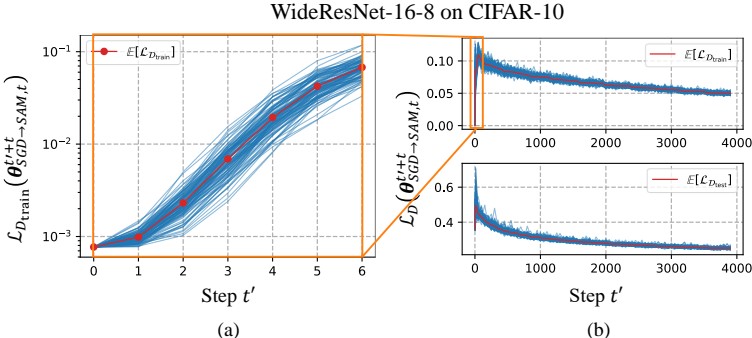

Figure 16: **Extending Figure 4 (c) to include more update steps.** Train/test loss $\mathcal{L}_{\mathcal{D}}(\boldsymbol{\theta}_{\text{SGD}\to\text{SAM},t}^{t+t'})$ v.s. the update step $t'$. Here, $t$ is the switching step, chosen to be sufficiently large for SGD to converge, and $t'$ is the number of updates after switching to SAM. The red line represents the mean over 100 trials with different randomness.

conjecture in Section 5, the properties of the final solutions are primarily shaped by the optimization method chosen in the late training phase. Consequently, when switching from SAM to SGD, the generalization properties of the final solution are expected to be independent of the perturbation radius, as SGD dominate in the final stage of training.

To further explore this, we have conducted experiments to investigate the effect of perturbation radius on generalization when switching from SAM to SGD. Specifically, we varied the perturbation radius $\rho \in \{0.02, 0.05, 0.1, 0.15, 0.2\}$ for SAM and measured how the test loss and error of $\boldsymbol{\theta}_{\text{SAM}\to\text{SGD},t}^{T}$ change as the SAM training proportion $\frac{t}{T}$ increases. In Figure 15, we observe no significant differences in test loss or error across different $\rho$ values, even when training with SAM for most of the time (e.g., $\frac{t}{T} = 0.8$). Notably, only full SAM training (i.e., $\frac{t}{T} = 1$) shows a pronounced dependence on $\rho$, where a larger perturbation radius leads to better generalization. These findings align with our expectation that the perturbation radius has minimal impact on generalization when transitioning from SAM to SGD, further validating our theoretical results in Section 4 and conjecture in Section 5.

**Extending Figure 4 (c) to include more update steps.** In Figure 16, we observe that the training loss initially increases exponentially fast and then gradually decreases, consistent with our theoretical predictions.

## C    PROOFS IN SECTION 4

In our theoretical analysis of Stochastic SAM, defined in Equation (4), we adopt the technical simplification in Appendix C.2 of Andriushchenko & Flammarion (2022), which assumes that the inner noise and outer noise are independent. Specifically, we analyze the following variant of Equation (4):

$$\boldsymbol{\theta}^{t+1} = \boldsymbol{\theta}^t - \eta \mathcal{L}_{\xi_t^2}(\boldsymbol{\theta}^t + \rho \nabla \mathcal{L}_{\xi_t^1}(\boldsymbol{\theta}^t)) \tag{7}$$

where $\xi_t^1$ and $\xi_t^2$ are two independent stochastic mini-batches of data of size $B$.

### C.1    PROOF OF THEOREM 4.1 AND COROLLARY 4.2

For simplicity, we denote the two-step update rules of SAM defined in Equation (7) as:

$$\boldsymbol{\theta}^{t+1/2} = \boldsymbol{\theta}^t + \rho \nabla \mathcal{L}_{\xi_t^1}(\boldsymbol{\theta}^t),$$

$$\boldsymbol{\theta}^{t+1} = \boldsymbol{\theta}^t - \eta \nabla \mathcal{L}_{\xi_t^2}(\boldsymbol{\theta}^{t+1/2}).$$

For $B$-batch gradient noise, denoted by $\boldsymbol{\xi}^B(\boldsymbol{\theta}) := \mathcal{L}_\xi(\boldsymbol{\theta}) - \nabla \mathcal{L}(\boldsymbol{\theta})$, we have:

$$\mathbb{E}[\boldsymbol{\xi}^B(\boldsymbol{\theta})\boldsymbol{\xi}^B(\boldsymbol{\theta})^\top] = \frac{1}{B}\mathbb{E}[\boldsymbol{\xi}(\boldsymbol{\theta})\boldsymbol{\xi}(\boldsymbol{\theta})^\top] = \frac{1}{B}\Sigma(\boldsymbol{\theta}), \tag{8}$$

where $\xi$ are the sample indices in a mini-batch and $\boldsymbol{\xi}(\boldsymbol{\theta})$ is the 1-batch gradient noise, used in Assumption 4.1.

For the linearized model ( Equation (5)), it holds that:

$$\mathcal{L}(\boldsymbol{\theta}) = \frac{1}{2}(\boldsymbol{\theta} - \boldsymbol{\theta}^\star)^\top H(\boldsymbol{\theta}^\star)(\boldsymbol{\theta} - \boldsymbol{\theta}^\star).$$

Without loss of generality, we can let $\boldsymbol{\theta}^\star = 0$. For simplicity, we denote $G := H(\boldsymbol{\theta}^\star)$.

Next, according to the definition of linear stability (Definition 4.1), we consider the dynamics of stochastic SAM on the linearized model.

First, we have the following estimate for $\mathbb{E}\left[\mathcal{L}(\boldsymbol{\theta}^{t+1})\right]$:

$$\begin{aligned}
&\mathbb{E}\left[\mathcal{L}(\boldsymbol{\theta}^{t+1})\right] \\
=& \frac{1}{2}\mathbb{E}\left[(\boldsymbol{\theta}^t - \eta\nabla\mathcal{L}_{\xi_t^2}(\boldsymbol{\theta}^{t+1/2}))^\top G(\boldsymbol{\theta}^t - \eta\nabla\mathcal{L}_{\xi_t^2}(\boldsymbol{\theta}^{t+1/2}))\right] \\
=& \frac{1}{2}\mathbb{E}\left[(\boldsymbol{\theta}^t - \eta\nabla\mathcal{L}(\boldsymbol{\theta}^{t+1/2}) - \eta\boldsymbol{\xi}^B(\boldsymbol{\theta}^{t+1/2}))^\top G(\boldsymbol{\theta}^t - \eta\nabla\mathcal{L}(\boldsymbol{\theta}^{t+1/2}) - \eta\boldsymbol{\xi}^B(\boldsymbol{\theta}^{t+1/2}))\right] \\
=& \frac{1}{2}\mathbb{E}\left[(\boldsymbol{\theta}^t - \eta\nabla\mathcal{L}(\boldsymbol{\theta}^{t+1/2})^\top G(\boldsymbol{\theta}^t - \eta\nabla\mathcal{L}(\boldsymbol{\theta}^{t+1/2})\right] + 0 \\
&+ \frac{1}{2}\eta^2\mathbb{E}\left[\boldsymbol{\xi}^B(\boldsymbol{\theta}^{t+1/2})^\top G\boldsymbol{\xi}^B(\boldsymbol{\theta}^{t+1/2})\right] \\
\overset{(8)}{=}& \frac{1}{2}\mathbb{E}\left[(\boldsymbol{\theta}^t - \eta\nabla\mathcal{L}(\boldsymbol{\theta}^{t+1/2})^\top G(\boldsymbol{\theta}^t - \eta\nabla\mathcal{L}(\boldsymbol{\theta}^{t+1/2})\right] + \frac{\eta^2}{2B}\mathbb{E}\left[\mathrm{Tr}\left(\Sigma(\boldsymbol{\theta}^{t+1/2})G\right)\right] \\
\overset{\text{Assumption 4.1}}{\geq}& 0 + \frac{\eta^2\gamma\|G\|_F^2}{B}\mathbb{E}\left[\mathcal{L}(\boldsymbol{\theta}^{t+1/2})\right] = \frac{\eta^2\gamma\|G\|_F^2}{B}\mathbb{E}\left[\mathcal{L}(\boldsymbol{\theta}^{t+1/2})\right].
\end{aligned}$$

**Remark C.1** (Clarifications on $\mathbb{E}$). The expectation $\mathbb{E}$ is taken with respect to the random variables $\{\xi_t^1, \xi_t^2\}_t$ ($t \geq 0$), which are the indices of independently random examples sampled from $\mathcal{D}_{\mathrm{train}}$.

**Remark C.2** ($\boldsymbol{\xi}^B(\boldsymbol{\theta}^{t+1/2})$ and $\boldsymbol{\theta}^t - \eta\nabla\mathcal{L}(\boldsymbol{\theta}^{t+1/2})$ are uncorrelated). Observing that $\boldsymbol{\theta}^t - \eta\nabla\mathcal{L}(\boldsymbol{\theta}^{t+1/2}) = \phi(\xi_1^1, \xi_1^2, \cdots, \xi_t^1)$ and $\boldsymbol{\theta}^{t+1/2} = \psi(\xi_1^1, \xi_1^2, \cdots, \xi_t^1)$, we can use the law of total expectation to derive

$$\mathbb{E}[\langle\boldsymbol{\xi}^B(\boldsymbol{\theta}^{t+1/2}), \boldsymbol{\theta}^t - \eta\nabla\mathcal{L}(\boldsymbol{\theta}^{t+1/2})\rangle]$$

$$=\mathbb{E}_{\xi_1^1,\xi_1^2,\cdots,\xi_t^1}\left[\mathbb{E}_{\xi_t^2}\left[\left\langle\nabla\mathcal{L}_{\xi_t^2}(\boldsymbol{\theta}^{t+1/2})-\nabla\mathcal{L}(\boldsymbol{\theta}^{t+1/2}),\boldsymbol{\theta}^t-\eta\nabla\mathcal{L}(\boldsymbol{\theta}^{t+1/2})\right\rangle\Big|\xi_1^1,\xi_1^2,\cdots,\xi_t^1\right]\right]$$

$$=\mathbb{E}_{\xi_1^1,\xi_1^2,\cdots,\xi_t^1}\left[\mathbb{E}_{\xi_t^2}\left[\left\langle\nabla\mathcal{L}_{\xi_t^2}(\psi(\xi_1^1,\xi_1^2,\cdots,\xi_t^1))-\nabla\mathcal{L}(\psi(\xi_1^1,\xi_1^2,\cdots,\xi_t^1)),\phi(\xi_1^1,\xi_1^2,\cdots,\xi_t^1)\right\rangle\Big|\xi_1^1,\xi_1^2,\cdots,\xi_t^1\right]\right]$$

$$=\mathbb{E}_{\xi_1^1,\xi_1^2,\cdots,\xi_t^1}[0]=0.$$

Then we estimate the lower bound for $\mathbb{E}\left[\mathcal{L}(\boldsymbol{\theta}^{t+1/2})\right]$:

$$\mathbb{E}\left[\mathcal{L}(\boldsymbol{\theta}^{t+1/2})\right]$$
$$=\frac{1}{2}\mathbb{E}\left[(\boldsymbol{\theta}^t-\rho\nabla\mathcal{L}_{\xi_t^1}(\boldsymbol{\theta}^t))^\top G(\boldsymbol{\theta}^t+\rho\nabla\mathcal{L}_{\xi_t^1}(\boldsymbol{\theta}^t))\right]$$
$$=\frac{1}{2}\mathbb{E}\left[(\boldsymbol{\theta}^t+\rho\nabla\mathcal{L}(\boldsymbol{\theta}^t)+\rho\boldsymbol{\xi}^B(\boldsymbol{\theta}^t))^\top G(\boldsymbol{\theta}^t+\rho\nabla\mathcal{L}(\boldsymbol{\theta}^t)+\rho\boldsymbol{\xi}^B(\boldsymbol{\theta}^t))\right]$$
$$=\frac{1}{2}\mathbb{E}\left[(\boldsymbol{\theta}^t+\rho\nabla\mathcal{L}(\boldsymbol{\theta}^t))^\top G(\boldsymbol{\theta}^t+\rho\nabla\mathcal{L}(\boldsymbol{\theta}^t))\right]+\frac{1}{2}\rho^2\mathbb{E}\left[\boldsymbol{\xi}^B(\boldsymbol{\theta}^t)^\top G\boldsymbol{\xi}^B(\boldsymbol{\theta}^t)\right]$$
$$\overset{(8)}{=}\frac{1}{2}\mathbb{E}\left[(\boldsymbol{\theta}^t+\rho\nabla\mathcal{L}(\boldsymbol{\theta}^t))^\top G(\boldsymbol{\theta}^t+\rho\nabla\mathcal{L}(\boldsymbol{\theta}^t))\right]+\frac{\rho^2}{2B}\mathbb{E}\left[\mathrm{Tr}\left(\Sigma(\boldsymbol{\theta}^t)G\right)\right]$$
$$\overset{\text{Assumption 4.1}}{\geq}\frac{1}{2}\mathbb{E}\left[(\boldsymbol{\theta}^t+\rho\nabla\mathcal{L}(\boldsymbol{\theta}^t))^\top G(\boldsymbol{\theta}^t+\rho\nabla\mathcal{L}(\boldsymbol{\theta}^t))\right]+\frac{\rho^2\gamma\|G\|_F^2}{B}\mathbb{E}\left[\mathcal{L}(\boldsymbol{\theta}^t)\right]$$
$$=\frac{1}{2}\mathbb{E}\left[\boldsymbol{\theta}^{t\top}(I+\rho G)G(I+\rho G)\boldsymbol{\theta}^t\right]+\frac{\rho^2\gamma\|G\|_F^2}{B}\mathbb{E}\left[\mathcal{L}(\boldsymbol{\theta}^t)\right]$$
$$\overset{(\clubsuit)}{\geq}\frac{1}{2}\mathbb{E}\left[\boldsymbol{\theta}^{t\top}G\boldsymbol{\theta}^t\right]+\frac{\rho^2\gamma\|G\|_F^2}{B}\mathbb{E}\left[\mathcal{L}(\boldsymbol{\theta}^t)\right]$$
$$=\left(1+\frac{\rho^2\gamma\|G\|_F^2}{B}\right)\mathbb{E}\left[\mathcal{L}(\boldsymbol{\theta}^t)\right],$$

where ($\clubsuit$) holds due to $(I+\rho G)G(I+\rho G)=G+2\rho G^2+\rho^2 G^3\succeq G$.

By combining these two estimates, we obtain:

$$\mathbb{E}\left[\mathcal{L}(\boldsymbol{\theta}^{t+1})\right]\geq\frac{\eta^2\gamma\|G\|_F^2}{B}\left(1+\frac{\rho^2\gamma\|G\|_F^2}{B}\right)\mathbb{E}\left[\mathcal{L}(\boldsymbol{\theta}^t)\right]$$

Hence, if $\boldsymbol{\theta}^\star$ is linear stable for stochastic SAM, it must hold:

$$\frac{\eta^2\gamma\|G\|_F^2}{B}\left(1+\frac{\rho^2\gamma\|G\|_F^2}{B}\right)\leq 1,\tag{9}$$

which completes the proof of Theorem 4.1.

Furthermore, if the condition (9) can not be satisfied, i.e., $\frac{\eta^2\gamma\|G\|_F^2}{B}\left(1+\frac{\rho^2\gamma\|G\|_F^2}{B}\right)>1$, then the exponentially fast escape holds:

$$\mathbb{E}\left[\mathcal{L}(\boldsymbol{\theta}^t)\right]\geq C^t\mathbb{E}\left[\mathcal{L}(\boldsymbol{\theta}^0)\right],$$

where $C=\frac{\eta^2\gamma\|G\|_F^2}{B}\left(1+\frac{\rho^2\gamma\|G\|_F^2}{B}\right)>1$. This complete the proof of Corollary 4.2.

## C.2 PROOF OF PROPOSITION 4.1

As we mentioned in Section 4.2, this proposition focus on a model size of $p=1$ and consider the full-batch SAM, i.e.,

$$\theta^{t+1}=\theta^t-\eta\nabla\mathcal{L}(\theta^t+\rho\nabla\mathcal{L}(\theta^t)).\tag{10}$$

*Proof of Proposition 4.1.*
First, we define the hitting time:

$$T := \inf\left\{t \in \mathbb{N} : |\theta^{t+1}| \geq b\right\}.$$

We aim to prove that $T$ does not exists, which implies that $\theta^t \in (-b, b) \subset V$ holds for all $t \in \mathbb{N}$.

Assuming $T$ exists, then one of the following three cases must hold:

- Case I: $\theta^T = 0$. In this case, $\mathcal{L}'(\theta^T) = 0$, which implies that $\theta^{T+1} = 0$. This contradicts the definition of $T$.

- Case II: $0 < \theta^T < b$. By the sub-quadratic property and $\rho \leq 1/a$, for the inner update $\theta^{T+1/2} := \theta^T + \rho\mathcal{L}'(\theta^T)$, we have:

$$0 < \theta^T < \theta^{T+1/2} = \theta^T + \rho\int_0^{\theta^T}\mathcal{L}''(z)\mathrm{d}z < \theta^T + \rho a\theta^T \leq 2\theta^T < 2b.$$

  Consequently, we estimate the two-sided bounds for $\theta^{T+1}$:

  – The lower bound for $\theta^{T+1}$:

$$\theta^{T+1} = \theta^T - \eta\mathcal{L}'(\theta^{T+1/2}) > 0 - \eta\sup_{z\in(0,2b)}\mathcal{L}'(z)$$

$$\geq 0 - \eta\max_{z\in(-2b,2b)}\mathcal{L}'(z) > -b.$$

  – The upper bound for $\theta^{T+1}$:

$$\theta^{T+1} = \theta^T - \eta\mathcal{L}'(\theta^{T+1/2}) < \theta^T + \rho\mathcal{L}'(\theta^T) - \eta\mathcal{L}'(\theta^{T+1/2})$$

$$\overset{(\spadesuit)}{<} \theta^T + \eta\mathcal{L}'(\theta^{T+1/2}) - \eta\mathcal{L}'(\theta^{T+1/2}) = \theta^T < b.$$

  To justify ($\spadesuit$), we prove $\rho < \eta\dfrac{\mathcal{L}'(\theta^{T+1/2})}{\mathcal{L}'(\theta^T)}$ when $\mathcal{L}'(\theta^T) > 0$ (noticing $\theta^T\mathcal{L}'(\theta^T) \geq 0$ by sub-quadratic property) Since $\theta^T < \theta^{T+1/2} < 2\theta^T$, we aconsider two scenarios:

  * If $\int_{\theta^{T+1/2}}^{2\theta^T}\mathcal{L}''(z)\mathrm{d}z \leq 0$, then

$$\frac{\mathcal{L}'(\theta^{T+1/2})}{\mathcal{L}'(\theta^T)} = \frac{\mathcal{L}'(2\theta^T) - \int_{\theta^{T+1/2}}^{2\theta^T}\mathcal{L}''(z)\mathrm{d}z}{\mathcal{L}'(\theta^T)} \geq \frac{\mathcal{L}'(2\theta^T)}{\mathcal{L}'(\theta^T)}.$$

    Using the condition $\rho < \eta\min_{0<|z|<b}\left|\dfrac{\mathcal{L}'(2z)}{\mathcal{L}'(z)}\right|$, we obtain $\rho < \eta\dfrac{\mathcal{L}'(\theta^{T+1/2})}{\mathcal{L}'(\theta^T)}$.

  * If $\int_{\theta^{T+1/2}}^{2\theta^T}\mathcal{L}''(z)\mathrm{d}z > 0$, then since the sub-quaratic property $\mathcal{L}''(|z_1|) \geq \mathcal{L}''(|z_2|)$ for $|z_1| \leq |z_2|$, we have $\mathcal{L}''(\theta^{T+1/2}) > 0$. Thus, $\mathcal{L}''(z) \geq \mathcal{L}''(\theta^{T+1/2}) > 0$ for $\theta^T < z < \theta^{T+1/2}$, implying:

$$\mathcal{L}'(\theta^{T+1/2}) = \mathcal{L}'(\theta^T) + \int_{\theta^T}^{\theta^{T+1/2}}\mathcal{L}''(z)\mathrm{d}z > \mathcal{L}'(\theta^T).$$

    Using the condition $\rho < \eta$, we obtain $\rho < \eta < \eta\dfrac{\mathcal{L}'(\theta^{T+1/2})}{\mathcal{L}'(\theta^T)}$.

  Therefore, we have shown $\rho < \eta\dfrac{\mathcal{L}'(\theta^{T+1/2})}{\mathcal{L}'(\theta^T)}$, implying ($\spadesuit$).

  Thus, we obtain $|\theta^{T+1}| < b$, which contradicts the definition of $T$.

- Case III: $-b < \theta^T < 0$. The proof for this case is extremely similar to the proof for Case II. This case is also contradicts the definition of $T$.

Thus, we have proved that $T$ does not exist, which implies i.e., $\theta^t \in (-b, b) \subset V, \forall t \in \mathbb{N}$.

$\square$

The next result provides an example for the highly sub-quadratic landscape. Intuitively, it means that the decay of $\mathcal{L}''(|z|)$ is extremely fast.

**Example C.1** (highly sub-quadratic landscape). *Let $\mathcal{L}(0) = \mathcal{L}'(0) = 0$, $\mathcal{L}''(0) = a$, and*

$$\mathcal{L}''(z) = \begin{cases} a - \frac{|z|}{\epsilon}, & |z| < a\epsilon \\ 0, & \text{else in } [-2b, 2b] \end{cases},$$

*where $0 < \epsilon \ll 1$ is a constant to reflect the sub-quadratic degree.*

*Notice that this example satisfies Definition 4.2. Moreover, it is straightforward that*

$$\mathcal{L}'(z) = \begin{cases} z\left(a - \frac{|z|}{2\epsilon}\right), & |z| < a\epsilon \\ \frac{a\epsilon}{2}, & \text{else in } [-2b, 2b] \end{cases}.$$

*Additionally, we can calculate the upper bounds for $\eta$ and $\rho$ in Proposition 4.1:*

$$\eta \leq \min_{|z| \leq 2b} \frac{b}{|\mathcal{L}'(z)|} = \frac{2b}{\epsilon a},$$

$$\rho \leq \min\left\{\frac{1}{a}, \min_{|z| \leq b} \left|\frac{\mathcal{L}'(2z)}{\mathcal{L}'(z)}\right|\right\} = \min\left\{\frac{1}{a}, 2\right\}.$$

*One can see that in the highly sub-quadratic case, i.e., $0 < \epsilon \ll 1$, the upper bound for $\eta$ goes to $+\infty$.*

In addition, we prove Proposition C.1, which illustrates that SAM will escape the current valley under a quadratic landscape. This result stands in stark contrast to Proposition 4.1 for sub-quadratic landscape, which further confirms our adaptation of the sub-quadratic landscape.

**Proposition C.1** (Non-local escape behaviour under a quadratic landscape.). *Assume the landscape in the valley $V_b = (-2b, 2b)$ be quadratic, i.e., $\mathcal{L}(z) = az^2/2$. Then, for all initialization $\theta^0 \in V_b \backslash \{0\}$, and $\eta, \rho$ s.t. $\eta > \frac{2}{a(1+a\rho)}$, full-batch SAM will escape from the valley $V_b$, i.e., there exists $T > 0$ s.t. $\theta^T \notin V_b$.*

*Proof of Proposition C.1.* By a straightforward calculation, we have

$$\theta^{t+1} = \theta^t - \eta a\left(\theta^t + \rho \cdot a\theta^t\right) = (1 - \eta a(1 + a\rho))\theta^t.$$

If $\eta > \frac{2}{a(1+a\rho)}$, then $|\theta^t| \geq |1 - \eta a(1 + a\rho)|^t |\theta^0|$, where $|1 - \eta a(1 + a\rho)| > 1$. This means that there exists $T > 0$ s.t. $\theta^T \notin V_b$. $\qquad\square$

**Remark C.3** (multiple valleys in Proposition 4.1). Proposition 4.1 assumes the landscape is sub-quadratic within the valley $V = [-2b, 2b]$, but it does not impose any assumptions about the landscape outside $V$. Therefore, additional valleys can exist in $(-\infty, -2b) \cup (2b, +\infty)$. For example, consider

$$\mathcal{L}(\theta) = 1 - \cos\left(\frac{\pi\theta}{2b}\right),$$

which satisfies the sub-quadratic property within $V = [-2b, 2b]$. This loss function has infinitely many valleys:

$$[2(2k-1)b, 2(2k+1)b], k \in \mathbb{Z}.$$

By Proposition 4.1, SAM remains in the current valley $V = [-2b, 2b]$ and cannot enter into other valleys.

**Proposition C.2** (Extension of Proposition 4.1). *Under Definition 4.2, assume the landscape is sub-quadratic in the valley $V = [-2b, 2b]$. Then, for any $\epsilon \in (0, 1)$ and all initialization $\theta^0 \in (-(2-\epsilon)b, (2-\epsilon)b)$, and $\eta, \rho$ s.t. $\eta < \min_{z \in V}(2-\epsilon)b/|\mathcal{L}'(z)|$, $\rho \leq \min\left\{\frac{\epsilon}{a(2-\epsilon)}, \eta, \eta \min_{0<|z|<(2-\epsilon)b} \left|\frac{\mathcal{L}'(\frac{2}{2-\epsilon}z)}{\mathcal{L}'(z)}\right|\right\}$, the full-batch SAM will remain within the valley $V$, i.e., $\theta^t \in V$ for all $t \in \mathbb{N}$.*

Notice that Proposition C.2 generalizes Proposition 4.1 to almost all initializations within the valley $(-2b, 2b)$. By choosing sufficiently small $\epsilon$, the result effectively applies to all initializations within $(-2b, 2b)$.

*Proof of Proposition C.2.* This proof is highly similar to that of Proposition 4.1. First, we define the hitting time:

$$T := \inf \left\{ t \in \mathbb{N} : |\theta^{t+1}| \geq (2 - \epsilon)b \right\}.$$

We aim to prove that $T$ does not exists, which implies that $\theta^t \in (-(2 - \epsilon)b, (2 - \epsilon)b) \subset V$ holds for all $t \in \mathbb{N}$.

Assuming $T$ exists, then one of the following three cases must hold:

- Case I: $\theta^T = 0$. In this case, $\mathcal{L}'(\theta^T) = 0$, which implies that $\theta^{T+1} = 0$. This contradicts the definition of $T$.

- Case II: $0 < \theta^T < (2 - \epsilon)b$. By the sub-quadratic property and $\rho \leq \frac{\epsilon}{a(2-\epsilon)}$, for the inner update $\theta^{T+1/2} := \theta^T + \rho \mathcal{L}'(\theta^T)$, we have:

$$0 < \theta^T < \theta^{T+1/2} = \theta^T + \rho \int_0^{\theta^T} \mathcal{L}''(z) \mathrm{d}z < \theta^T + \rho a \theta^T \leq \frac{2}{2 - \epsilon} \theta^T < 2b.$$

  Consequently, we estimate the two-sided bounds for $\theta^{T+1}$:

  - The lower bound for $\theta^{T+1}$:
  $$\theta^{T+1} = \theta^T - \eta \mathcal{L}'(\theta^{T+1/2}) > 0 - \eta \sup_{z \in (0, 2b)} \mathcal{L}'(z)$$
  $$\geq 0 - \eta \max_{z \in (-2b, 2b)} \mathcal{L}'(z) \geq -(2 - \epsilon)b.$$

  - The upper bound for $\theta^{T+1}$:
  $$\theta^{T+1} = \theta^T - \eta \mathcal{L}'(\theta^{T+1/2}) < \theta^T + \rho \mathcal{L}'(\theta^T) - \eta \mathcal{L}'(\theta^{T+1/2})$$
  $$\overset{(\sharp)}{<} \theta^T + \eta \mathcal{L}'(\theta^{T+1/2}) - \eta \mathcal{L}'(\theta^{T+1/2}) = \theta^T < (2 - \epsilon)b.$$

  To justify $(\sharp)$, we prove $\rho < \eta \frac{\mathcal{L}'(\theta^{T+1/2})}{\mathcal{L}'(\theta^T)}$ when $\mathcal{L}'(\theta^T) > 0$ (noticing $\theta^T \mathcal{L}'(\theta^T) \geq 0$ by sub-quadratic property) Since $\theta^T < \theta^{T+1/2} < \frac{2}{2-\epsilon} \theta^T$, we consider two scenarios:

    * If $\int_{\theta^{T+1/2}}^{\frac{2}{2-\epsilon}\theta^T} \mathcal{L}''(z) \mathrm{d}z \leq 0$, then

    $$\frac{\mathcal{L}'(\theta^{T+1/2})}{\mathcal{L}'(\theta^T)} = \frac{\mathcal{L}'(\frac{2}{2-\epsilon}\theta^T) - \int_{\theta^{T+1/2}}^{\frac{2}{2-\epsilon}\theta^T} \mathcal{L}''(z)\mathrm{d}z}{\mathcal{L}'(\theta^T)} \geq \frac{\mathcal{L}'(\frac{2}{2-\epsilon}\theta^T)}{\mathcal{L}'(\theta^T)}.$$

    Using the condition $\rho < \eta \min_{0 < |z| < (2-\epsilon)b} \left| \frac{\mathcal{L}'(\frac{2}{2-\epsilon}z)}{\mathcal{L}'(z)} \right|$, we obtain $\rho < \eta \frac{\mathcal{L}'(\theta^{T+1/2})}{\mathcal{L}'(\theta^T)}$.

    * If $\int_{\theta^{T+1/2}}^{\frac{2}{2-\epsilon}\theta^T} \mathcal{L}''(z) \mathrm{d}z > 0$, then since the sub-quaratic property $\mathcal{L}''(|z_1|) \geq \mathcal{L}''(|z_2|)$ for $|z_1| \leq |z_2|$, we have $\mathcal{L}''(\theta^{T+1/2}) > 0$. Thus, $\mathcal{L}''(z) \geq \mathcal{L}''(\theta^{T+1/2}) > 0$ for $\theta^T < z < \theta^{T+1/2}$, implying:

    $$\mathcal{L}'(\theta^{T+1/2}) = \mathcal{L}'(\theta^T) + \int_{\theta^T}^{\theta^{T+1/2}} \mathcal{L}''(z)\mathrm{d}z > \mathcal{L}'(\theta^T).$$

    Using the condition $\rho < \eta$, we obtain $\rho < \eta < \eta \frac{\mathcal{L}'(\theta^{T+1/2})}{\mathcal{L}'(\theta^T)}$.

    Therefore, we have shown $\rho < \eta \frac{\mathcal{L}'(\theta^{T+1/2})}{\mathcal{L}'(\theta^T)}$, implying $(\sharp)$.

  Thus, we obtain $|\theta^{T+1}| < b$, which contradicts the definition of $T$.

- Case III: $-b < \theta^T < 0$. The proof for this case is extremely similar to the proof for Case II. This case is also contradicts the definition of $T$.

Thus, we have proved that $T$ does not exist, which implies i.e., $\theta^t \in (-b, b) \subset V, \forall t \in \mathbb{N}$.

$\square$

## C.3 PROOF OF THEOREM 4.3

For simplicity, we denote the two-step update rules of SAM defined in Equation (7) as:

$$\boldsymbol{\theta}^{t+1/2} = \boldsymbol{\theta}^t + \rho\nabla\mathcal{L}_{\xi_t^1}(\boldsymbol{\theta}^t),$$

$$\boldsymbol{\theta}^{t+1} = \boldsymbol{\theta}^t - \eta\nabla\mathcal{L}_{\xi_t^2}(\boldsymbol{\theta}^{t+1/2}).$$

Then we list a few useful properties under the condition of Theorem 4.3.

**Lemma C.1.** *Suppose Assumption 4.2. Then we have:*

$$2\mu\mathcal{L}(\boldsymbol{\theta}) \le \|\nabla\mathcal{L}(\boldsymbol{\theta})\|^2 \le \frac{2L^2}{\mu}\mathcal{L}(\boldsymbol{\theta}), \ \forall\boldsymbol{\theta}. \tag{11}$$

*Proof of Lemma C.1.* The upper bound is a direct corollary of PL and smoothness. Please refer to Lemma B.6 in Arora et al. (2022) and Lemma 5 in Ahn et al. (2023) for proof details. □

The smoothness in Assumption 4.2 also implies the classical quadratic upper bound for the loss (Bubeck et al., 2015):

$$\mathcal{L}(\boldsymbol{\theta} + \boldsymbol{v}) \le \mathcal{L}(\boldsymbol{\theta}) + \langle\nabla\mathcal{L}(\boldsymbol{\theta}), \boldsymbol{v}\rangle + \frac{L}{2}\|\boldsymbol{v}\|^2, \ \forall\boldsymbol{\theta}, \boldsymbol{v}. \tag{12}$$

Additionally, Assumption 4.3 holds for the gradient noise $\boldsymbol{\xi}(\boldsymbol{\theta})$ with batch size 1. Thus, for $B$-batch gradient noise, denoted by $\boldsymbol{\xi}^B(\boldsymbol{\theta})$, we have $\mathbb{E}[\boldsymbol{\xi}^B(\boldsymbol{\theta})\boldsymbol{\xi}^B(\boldsymbol{\theta})^\top] = \frac{1}{B}\mathbb{E}[\boldsymbol{\xi}(\boldsymbol{\theta})\boldsymbol{\xi}(\boldsymbol{\theta})^\top]$, implying:

$$\mathbb{E}\left[\|\boldsymbol{\xi}^B(\boldsymbol{\theta})\|^2\right] \le \frac{\sigma^2}{B}\mathcal{L}(\boldsymbol{\theta}), \ \forall\boldsymbol{\theta}. \tag{13}$$

For the learning rate $\eta$ and the permutation radius $\rho$, we choose then as follows:

$$\eta \le \min\left\{\frac{1}{2L}, \frac{\mu B}{2L\sigma^2}\right\}, \quad \rho \le \min\left\{\frac{1}{4L}, \frac{\mu B}{4L\sigma^2}, \frac{\eta\mu^2}{24L^2}\right\}. \tag{14}$$

### C.3.1 PROOF OF THEOREM 4.3

For $\mathcal{L}(\boldsymbol{\theta}^{t+1/2})$, it has the quadratic upper bound:

$$\mathcal{L}(\boldsymbol{\theta}^{t+1/2}) = \mathcal{L}\left(\boldsymbol{\theta}^t + \rho\nabla\mathcal{L}_{\xi_t^1}(\boldsymbol{\theta}^t)\right) = \mathcal{L}\left(\boldsymbol{\theta}^t + \rho\nabla\mathcal{L}(\boldsymbol{\theta}^t) + \rho\boldsymbol{\xi}^B(\boldsymbol{\theta}^t)\right)$$

$$\overset{(12)}{\le} \mathcal{L}\left(\boldsymbol{\theta}^t + \rho\nabla\mathcal{L}(\boldsymbol{\theta}^t)\right) + \left\langle\nabla\mathcal{L}\left(\boldsymbol{\theta}^t + \rho\nabla\mathcal{L}(\boldsymbol{\theta}^t)\right), \rho\boldsymbol{\xi}(\boldsymbol{\theta}^t)\right\rangle + \frac{\rho^2 L}{2}\|\boldsymbol{\xi}^B(\boldsymbol{\theta}^t)\|_2^2.$$

Taking the expectation, we obtain the upper bound of $\mathbb{E}[\mathcal{L}(\boldsymbol{\theta}^{t+1/2})]$:

$$\mathbb{E}\left[\mathcal{L}(\boldsymbol{\theta}^{t+1/2})\right] \le \mathbb{E}\left[\mathcal{L}\left(\boldsymbol{\theta}^t + \rho\nabla\mathcal{L}(\boldsymbol{\theta}^t)\right)\right] + \frac{\rho^2 L}{2}\mathbb{E}\left[\|\boldsymbol{\xi}^B(\boldsymbol{\theta}^t)\|^2\right]$$

$$\overset{(13)}{\le} \mathbb{E}\left[\mathcal{L}\left(\boldsymbol{\theta}^t + \rho\nabla\mathcal{L}(\boldsymbol{\theta}^t)\right)\right] + \frac{\rho^2 L}{2}\frac{\sigma^2}{B}\mathbb{E}\left[\mathcal{L}(\boldsymbol{\theta}^t)\right]$$

$$\overset{(12)}{\le} \mathbb{E}\left[\mathcal{L}\left(\boldsymbol{\theta}^t\right) + \rho\|\nabla\mathcal{L}(\boldsymbol{\theta}^t)\|^2 + \frac{\rho^2 L}{2}\|\nabla\mathcal{L}(\boldsymbol{\theta}^t)\|^2\right] + \frac{\rho^2 L}{2}\frac{\sigma^2}{B}\mathbb{E}\left[\mathcal{L}(\boldsymbol{\theta}^t)\right]$$

$$\overset{(11)}{\le} \left(1 + \frac{\rho(2 + \rho L)L^2}{\mu} + \frac{\rho^2 L\sigma^2}{2B}\right)\mathbb{E}\left[\mathcal{L}(\boldsymbol{\theta}^t)\right]$$

$$\overset{(14)}{\le} \left(1 + \frac{6\rho L^2}{\mu}\right)\mathbb{E}\left[\mathcal{L}(\boldsymbol{\theta}^t)\right] \le \left(1 + \frac{\eta\mu}{4}\right)\mathbb{E}\left[\mathcal{L}(\boldsymbol{\theta}^t)\right] \le \frac{5}{4}\mathbb{E}\left[\mathcal{L}(\boldsymbol{\theta}^t)\right].$$

In the similar way, for $\mathcal{L}(\boldsymbol{\theta}^{t+1})$, it has the quadratic upper bound:

$$\mathcal{L}(\boldsymbol{\theta}^{t+1}) = \mathcal{L}\left(\boldsymbol{\theta}^t - \eta\nabla\mathcal{L}_{\xi_t^2}(\boldsymbol{\theta}^{t+1/2})\right) = \mathcal{L}\left(\boldsymbol{\theta}^t - \eta\nabla\mathcal{L}(\boldsymbol{\theta}^{t+1/2}) - \eta\boldsymbol{\xi}^B(\boldsymbol{\theta}^{t+1/2})\right)$$

$$\overset{(12)}{\le} \mathcal{L}\left(\boldsymbol{\theta}^t - \eta\nabla\mathcal{L}(\boldsymbol{\theta}^{t+1/2})\right) - \left\langle\nabla\mathcal{L}\left(\boldsymbol{\theta}^t - \eta\nabla\mathcal{L}(\boldsymbol{\theta}^{t+1/2})\right), \eta\boldsymbol{\xi}(\boldsymbol{\theta}^{t+1/2})\right\rangle$$
$$+ \frac{\eta^2 L}{2}\left\|\boldsymbol{\xi}^B(\boldsymbol{\theta}^{t+1/2})\right\|_2^2.$$

Taking the expectation, we obtain the upper bound of $\mathbb{E}[\mathcal{L}(\boldsymbol{\theta}^{t+1})]$:

$$\mathbb{E}\left[\mathcal{L}(\boldsymbol{\theta}^{t+1})\right] \le \mathbb{E}\left[\mathcal{L}\left(\boldsymbol{\theta}^t - \eta\nabla\mathcal{L}(\boldsymbol{\theta}^{t+1/2})\right)\right] + \frac{\eta^2 L}{2}\mathbb{E}\left[\left\|\boldsymbol{\xi}^B(\boldsymbol{\theta}^{t+1/2})\right\|^2\right]$$

$$\overset{(13)}{\le} \mathbb{E}\left[\mathcal{L}\left(\boldsymbol{\theta}^t - \eta\nabla\mathcal{L}(\boldsymbol{\theta}^{t+1/2})\right)\right] + \frac{\eta^2 L}{2}\frac{\sigma^2}{B}\mathbb{E}\left[\mathcal{L}(\boldsymbol{\theta}^{t+1/2})\right]$$

$$\overset{(12)}{\le} \mathbb{E}\left[\mathcal{L}\left(\boldsymbol{\theta}^t\right) - \eta\left\langle\nabla\mathcal{L}(\boldsymbol{\theta}^t), \nabla\mathcal{L}(\boldsymbol{\theta}^{t+1/2})\right\rangle + \frac{\eta^2 L}{2}\left\|\nabla\mathcal{L}(\boldsymbol{\theta}^{t+1/2})\right\|^2\right] + \frac{\eta^2 L}{2}\frac{\sigma^2}{B}\mathbb{E}\left[\mathcal{L}(\boldsymbol{\theta}^{t+1/2})\right]$$

$$\overset{(11)}{\le} \mathbb{E}\left[\mathcal{L}\left(\boldsymbol{\theta}^t\right) - \eta\left\langle\nabla\mathcal{L}(\boldsymbol{\theta}^t), \nabla\mathcal{L}(\boldsymbol{\theta}^{t+1/2})\right\rangle\right] + \left(\eta^2 L\mu + \frac{\eta^2 L\sigma^2}{2B}\right)\mathbb{E}\left[\mathcal{L}(\boldsymbol{\theta}^{t+1/2})\right]$$

$$\overset{(14)}{\le} \mathbb{E}\left[\mathcal{L}\left(\boldsymbol{\theta}^t\right) - \eta\left\langle\nabla\mathcal{L}(\boldsymbol{\theta}^t), \nabla\mathcal{L}(\boldsymbol{\theta}^{t+1/2})\right\rangle\right] + \frac{3\eta\mu}{4}\mathbb{E}\left[\mathcal{L}(\boldsymbol{\theta}^{t+1/2})\right]$$

$$\le \mathbb{E}\left[\mathcal{L}\left(\boldsymbol{\theta}^t\right) - \eta\left\|\nabla\mathcal{L}(\boldsymbol{\theta}^t)\right\|^2 + \eta\left\|\nabla\mathcal{L}(\boldsymbol{\theta}^t)\right\|\left\|\nabla\mathcal{L}(\boldsymbol{\theta}^{t+1/2}) - \nabla\mathcal{L}(\boldsymbol{\theta}^t)\right\|\right] + \frac{3\eta\mu}{4}\mathbb{E}\left[\mathcal{L}(\boldsymbol{\theta}^{t+1/2})\right]$$

$$\overset{(11)}{\le} \mathbb{E}\left[\mathcal{L}\left(\boldsymbol{\theta}^t\right) - \eta\left\|\nabla\mathcal{L}(\boldsymbol{\theta}^t)\right\|^2 + \eta\rho L\left\|\nabla\mathcal{L}(\boldsymbol{\theta}^t)\right\|^2\right] + \frac{3\eta\mu}{4}\mathbb{E}\left[\mathcal{L}(\boldsymbol{\theta}^{t+1/2})\right]$$

$$\overset{(14)}{\le} \mathbb{E}\left[\mathcal{L}\left(\boldsymbol{\theta}^t\right) - \frac{3\eta}{4}\left\|\nabla\mathcal{L}(\boldsymbol{\theta}^t)\right\|^2\right] + \frac{3\eta\mu}{4}\mathbb{E}\left[\mathcal{L}(\boldsymbol{\theta}^{t+1/2})\right]$$

$$\overset{(14)}{\le} \left(1 - \frac{3\eta\mu}{2}\right)\mathbb{E}\left[\mathcal{L}(\boldsymbol{\theta}^t)\right] + \frac{3\eta\mu}{4}\mathbb{E}\left[\mathcal{L}(\boldsymbol{\theta}^{t+1/2})\right].$$

Finally, by combining our estimates for $\mathbb{E}\left[\mathcal{L}(\boldsymbol{\theta}^{t+1})\right]$ and $\mathbb{E}\left[\mathcal{L}(\boldsymbol{\theta}^{t+1/2})\right]$, we obtain:

$$\mathbb{E}\left[\mathcal{L}(\boldsymbol{\theta}^{t+1})\right] \le \left(1 - \frac{3\eta\mu}{2}\right)\mathbb{E}\left[\mathcal{L}(\boldsymbol{\theta}^t)\right] + \frac{3\eta\mu}{4}\mathbb{E}\left[\mathcal{L}(\boldsymbol{\theta}^{t+1/2})\right].$$
$$\le \left(1 - \frac{3\eta\mu}{2}\right)\mathbb{E}\left[\mathcal{L}(\boldsymbol{\theta}^t)\right] + \frac{3\eta\mu}{4}\cdot\frac{5}{4}\mathbb{E}\left[\mathcal{L}(\boldsymbol{\theta}^t)\right] \le \left(1 - \frac{\eta\mu}{2}\right)\mathbb{E}\left[\mathcal{L}(\boldsymbol{\theta}^t)\right],$$

which implies the convergence rate:

$$\mathbb{E}\left[\mathcal{L}(\boldsymbol{\theta}^t)\right] \le \left(1 - \frac{\eta\mu}{2}\right)^t\mathbb{E}\left[\mathcal{L}(\boldsymbol{\theta}^0)\right], \ \forall t \in \mathbb{N}.$$

### C.4 COMPARISON BETWEEN TWO DEFINITIONS ON LINEAR STABILITY

**Definition C.1** (Norm/loss-based linear stability). For an update rule $\boldsymbol{\theta}^{t+1} = \boldsymbol{\theta}^t + F(\boldsymbol{\theta}^t; \boldsymbol{\xi}_t)$,

- (Norm-based linear stability) A global minimum $\boldsymbol{\theta}^\star$ is said to be linearly stable if there exists a $C > 0$ such that it holds for SAM on the linearized model (Equation (5)) that $\mathbb{E}[\|\boldsymbol{\theta}^t - \boldsymbol{\theta}^\star\|^2] \le C\mathbb{E}[\|\boldsymbol{\theta}^0 - \boldsymbol{\theta}^\star\|^2], \forall t \ge 0$.

- (Loss-based linear stability) A global minimum $\boldsymbol{\theta}^\star$ is said to be linearly stable if there exists a $C > 0$ such that it holds for SAM on the linearized model (Equation (5)) that $\mathbb{E}[\mathcal{L}(\boldsymbol{\theta}^t)] \le C\mathbb{E}[\mathcal{L}(\boldsymbol{\theta}^0)], \forall t \ge 0$.

**Lemma C.2.** *The following two-fold results holds:*

- *(L1). If $\boldsymbol{\theta}^\star$ is norm-based linearly stable, then it must be loss-based linearly stable.*

- *(L2). There exists an update rule and a $G(\boldsymbol{\theta}^\star)$, such that the converse of (L1) does not hold.*

*Proof of Lemma C.2.*
For (L1), it is straightforward by

$$\mathcal{L}(\boldsymbol{\theta}^t) = \frac{1}{2}(\boldsymbol{\theta} - \boldsymbol{\theta}^\star)^\top G(\boldsymbol{\theta}^\star)(\boldsymbol{\theta} - \boldsymbol{\theta}^\star) \le \|G(\boldsymbol{\theta}^\star)\| \|\boldsymbol{\theta}^t - \boldsymbol{\theta}^\star\|^2.$$

Thus, the norm-based linear stability can imply the loss-based linear stability.

For (L2), we only need to consider an example containing degenerated directions. Specifically, let

$$L(\boldsymbol{\theta}) = \frac{1}{2}\theta_1^2 + 0 \cdot \theta_2^2,$$

with $\boldsymbol{\theta}^\star = \mathbf{0}$.

Consider an update rule only occurs on $\theta_2$:

$$\theta_1^{t+1} = \theta_1^t; \quad \theta_2^{t+1} = \theta_2^t + \zeta_t, \ \zeta_t \sim \mathcal{N}(0, 1),$$

starting from $\boldsymbol{\theta}^0 = (1, 1)^\top$. Then it is loss-base linearly stable due to

$$\mathcal{L}(\boldsymbol{\theta}^t) = \frac{1}{2}(\theta_1^t)^2 + 0 = \frac{1}{2}(\theta_1^t)^2 \equiv \mathcal{L}(\boldsymbol{\theta}^0).$$

However, it is norm-base linearly nonstable:

$$\mathbb{E}\left[\|\boldsymbol{\theta}^t\|^2\right] = 1 + \mathbb{E}\left[|\theta_2^t|^2\right] = 1 + \mathbb{E}\left[\left|1 + \sum_{s=0}^{t-1} \zeta_s\right|^2\right] = 2 + t \to +\infty \text{ when } t \to +\infty.$$

$\square$

