# OpenReview forum: "Sharpness-Aware Minimization Efficiently Selects Flatter Minima Late In Training"
_ICLR.cc/2025/Conference — ICLR 2025 Spotlight_

### Official Review · Reviewer_S3eC · 2024-10-29

**Soundness:** 3
**Presentation:** 2
**Contribution:** 3
**Rating:** 8
**Confidence:** 3

**Summary:**

This paper studies Sharpness Aware Minimization (SAM), an optimization method that modifies the Stochastic Gradient Descent (SGD) in a way that the solution will prefer a flatter region (Eq. (1)).
In practice, its approximation (Eq. (4)) is implemented for computationally efficiency.
It is not well understood whether this approximated method has the intended effect of finding a flatter solution, and this is what this paper studies.

First, the paper empirically shows that when we switch from the SGD to the SAM during the updates, it increasingly improves the solution as we make the switching time earlier, both in terms of the loss and the sharpness (Figure 3). However, using SAM for the last few updates already accounts for the majority of the improvement, and the performance increment becomes very small after that. This suggests that applying SAM in a final few epochs may be nearly as good as doing so for the whole training.

Next, the authors theoretically study several properties of SAM. Theorems 4.1 and 4.2 show that SAM escapes from a solution with large sharpness exponentially fast. Furthermore, with a few additional assumptions, Theorem 4.3 shows SAM converges to the global minimum exponentially fast.

**Strengths:**

- The theorems are interesting and nicely presented with interpretations. They successfully explain how SAM escapes a sharp region and converges to a flat region.
- The paper cites relevant references and clearly discuss the differences.
- The paper is well written and easy to follow, except for a few points (please see the Weaknesses section).
- The convergence result in Theorem 4.3 does not rely on convexity.

**Weaknesses:**

- Eq. (4) and the update rule that the authors use in theory (the two-step update rule defined at the beginning of C.1) are not equivalent after eliminating $\theta_{t+1/2}$.

- The experiments and the theory part use different methods (Eqs. (3) and (4)).

- I could not understand
  - how to obtain the inequality in lines 1137-1138, and
  - the phrase "which implies that $\theta_{T+1} = 0$" in lines 1143-1144.
  - the equality in lines 1085-1086 and the inequality in lines 1087-1088.

- This paper only studies the squared loss.

---
### Relatively minor comments
- Theorems 4.1 and 4.2 rely on the linear approximation (5), so it does not explain the whole dynamics after escaping from a sharp minimum and before getting close to a flat minimum. Furthermore, combined with the squared loss, the analysis is in a classical strongly convex setup.

- The single-data setup of Example 4.1 is unrealistic, and I do not know how relevant it is. Moreover, Proposition 4.2 considers the case in which there is only one parameter and has almost no practical usefullness.

- Theorem 4.1 and 4.2 seem to be almost equivalent (contrapositive) to each other. Theorem 4.2 is slightly more detailed in that it provides the explicit inequality showing the exponential bound. Theorem 4.2 directly implies Theorem 4.1, so the latter should be a corollary.

- I am not totally convinced that Assumptions 4.1-4.3 will be actually satisfied in practical situations. To be fair, many supporting references are provided in this paper.

**Questions:**

### Major comments:
- Please address the first three comments in the Weaknesses section.

- I expected to see the loss going down in later steps in Figure 4 (b). Does the figure not show the entire steps?

- "(P3). SAM converges to a flatter minimum compared to SGD; Theorem 4.1" The result of Theorem 4.1 seems to be a little overstated here. If I understand correctly, Theorem 4.1 does not show SAM converges although Theorem 4.3 does. Note that the latter assumes additional conditions (Assumptions 4.2 and 4.3). Could the authors revise the sentence to a more accurate one?

### Minor comments:
- Are there any experiments using Eq. (4)?

- The paper uses the symbol $\approx$, but its mathematical definition is missing. Could the authors add the precise meaning?

- Proof of Proposition 4.1 could be improved by adding comments on which part the authors used the assumptions.

- Does this refer to Eq. (3)? If so, it should be clearly stated: "Our experiments are conducted using the original implementation of SAM"

---
**Edit**: During the discussion period, the authors addressed all of my concerns. I updated the Rating score from 6 to 8. (I was about to set it to 7, but there wasn't this option.)

---

> ### Author Response · Authors · 2024-11-21
> **Response to Reviewer S3eC (1/3)**
>
> Thank you for your great efforts on the review of this paper and for appreciating our recognizing the value of our theoretical analyses. We will try our best to address your questions.
>
> **If you still have further concerns, or if you are not satisfied by the current responses, please let us know, so that we can update the response ASAP.**
>
> **Q1: “Eq. (4) and the update rule that the authors use in theory (the two-step update rule defined at the beginning of C.1) are not equivalent.”**
>
> **A1**: Thank you for your careful review. We acknowledge that there is indeed a difference between these two formulations. Specifically, the update rule used in Appendix C.1 assumes independence between the randomness in the inner and outer updates of SAM. This simplification is a common technical approach, as decoupling the randomness in SAM's updates is challenging. Notably, this technique has also been employed in the classic work on understanding SAM (see Appendix C.2 of Andriushchenko & Flammarion (2022)). We agree that analyzing the original formulation of SAM is an intriguing direction for future research.
>
> **Q2: Concerns about our theoretical setup regarding the update rule. “The experiments and the theory part use different methods (Eqs. (3) and (4)).” and “Are there any experiments using Eq. (4)?”**
>
> **A2**: We are glad to discuss more on the discrepancy between our experimental and theoretical settings. First, we’d like to emphasize that the simplification of approximating the step size $\rho_t$ with a constant $\rho$ has been empirically justified by Andriushchenko & Flammarion (2022), and has been widely adopted in recent theoretical analysis of SAM (see more references in Sec. 2). This simplification maintains mathematical tractability without significantly affecting the general behavior of SAM, which has been supported by empirical evidence.
>
> **Additional Experiments**: Second, we have conducted new experiments using Eq. (4). Here, we refer to the update rule in Eq. (4) as Unnormalized-SAM (USAM). Specifically, we measured how the generalization gap between $\boldsymbol{\theta}\_{{\rm SGD \to USAM}, t}^T$ and $\boldsymbol{\theta}\_{{\rm USAM}}^T$, as well as the sharpness of $\boldsymbol{\theta}\_{{\rm SGD \to USAM}, t}^T$, changes as the USAM training proportion $\frac{T-t}{T}$ increases. In [Figure R.5](https://anonymous.4open.science/api/repo/SAM-Stability-ICLR2025-rebuttal-6B51/file/reb_fig5.pdf?v=6f25d852) (please click the link to check the figure), both the generalization gap and sharpness drop sharply once the USAM training proportion exceeds zero, consistent with our main discovery in Fig. 3. This finding validates the effectiveness of USAM in capturing the main behavior of SAM, further bridging the gap between our theoretical results and empirical findings.
>
> **Q3: Concerns about our theoretical setup regarding the loss function. “This paper only studies the squared loss.”**
>
> **A3**: Thank you for your comment. The use of the squared loss function is a common practice in theoretical analyses of neural networks, and it facilitates the study of various aspects including implicit bias [1], training dynamics [2], and convergence [3]. While this choice simplifies our theoretical framework, our theoretical results remain representative for understanding the training dynamics and generalization behavior of neural networks.
>
> **Additional Experiments**: To further address your concerns, we have conducted additional experiments to reproduce our main discovery in Sec. 3 using the squared loss. Consistently with the main paper, we measured how the generalization gap between $\boldsymbol{\theta}\_{{\rm SGD \to SAM}, t}^T$ and $\boldsymbol{\theta}\_{{\rm SAM}}^T$, as well as the sharpness of $\boldsymbol{\theta}\_{{\rm SGD \to SAM}, t}^T$, changes as the SAM training proportion $\frac{T-t}{T}$ increases. In [Figure R.6](https://anonymous.4open.science/api/repo/SAM-Stability-ICLR2025-rebuttal-6B51/file/reb_fig6.pdf?v=448aef37), both the generalization gap and sharpness follow trends consistent with our previous findings in Fig. 3, thereby bridging the gap between our theoretical and experimental results.

---

> ### Author Response · Authors · 2024-11-21
> **Response to Reviewer S3eC (2/3)**
>
> **Q4: Technical problems. “how to obtain the inequality in lines 1137-1138; the phrase "which implies that $\boldsymbol{\theta}\_{T+1}=0$" in lines 1143-1144; the equality in lines 1085-1086; the inequality in lines 1087-1088.”**
>
> **A4**: Thank you for your careful review. We have addressed these concerns and correct the mistakes in the revised paper.
>
> - Lines 1137-1138: The inequality can be derived by separately analyzing $\int\_{\theta\_{T+1/2}}^{2\theta\_T}\mathcal{L}''(z)\mathrm{d}z$ in two cases ($>0$ or $<0$) and applying two upper bounds of $\rho$ in each case. The detailed derivation has been included in our revised paper for clarity. Please kindly refer to our revised paper.
> - Lines 1143-1144: We thank the reviewer for pointing this out. The phrase "which implies that $\boldsymbol{\theta}\_{T+1}=0$" has been deleted in the revised version.
> - Lines 1085-1086: This equality follows from $\boldsymbol{\theta}+\rho\nabla\mathcal{L}(\boldsymbol{\theta})=(I+\rho G) \boldsymbol{\theta}$, given $\mathcal{L}(\boldsymbol{\theta})=\frac{1}{2}\boldsymbol{\theta}^\top G \boldsymbol{\theta}$).
> - Lines 1087-1088: The inequality holds because $G=\frac{1}{n}\sum\_{i=1}^n\nabla f(\boldsymbol{x}_i;\boldsymbol{\theta})\nabla f(\boldsymbol{x}_i;\boldsymbol{\theta})^\top\succeq0$. Thus, $(I+\rho G)G(I+\rho G)=G+2\rho G^2+\rho^2 G^3 \succeq G$.
>
> **Q5: “Theorems 4.1 and 4.2 rely on the linear approximation (5), so it does not explain the whole dynamics after escaping from a sharp minimum and before getting close to a flat minimum. Furthermore, combined with the squared loss, the analysis is in a classical strongly convex setup.”**
>
> **A5**: Thank you for your thoughtful comments.
>
> - **Beyond the dynamics near the minima**: Theorems 4.1 and 4.2 are indeed based on linear stability and apply to the dynamics near the minima. However, it is worth noting that *Proposition 4.1 and Theorem 4.3 are not based on linear approximation*. In particular, Proposition 4.1 addresses the high-loss region where the linear approximation does not hold (as detailed in the beginning of Section 4.2). Together, these four results provide a more comprehensive view of the training dynamics.
> - **Inquiry for “whole” training dynamics**: Characterizing the "whole" training dynamics is inherently challenging under general settings. Most existing analyses of the "whole" dynamics, even for (S)GD, are limited to extremely simplified setups, such as single-index models [4] or feature learning regime [5].
> - **Convex setup, not strongly convex.** Finally, we clarify that our analysis in Theorems 4.1 and 4.2 assumes a convex setup rather than a strongly convex one. This distinction is crucial because over-parameterized models typically exhibit many degenerate directions, which we discuss in more detail in Lemma C.2.
>
> **Q6: Concerns about Example 4.1. “The single-data setup of Example 4.1 is unrealistic, and I do not know how relevant it is.”**
>
> **A6**: We would like to re-clarify that Example 4.1 is primarily intended to clearly illustrate our two-phase picture. While this example is simple, it is representative in several key ways: 1) it uses a two-layer neural network; 2) addresses a fitting problem; 3) includes a landscape with multiple valleys; 4) satisfies the sub-quadratic property; 5) exhibits differences in flatness between minima; 6) and highlights SAM’s two-phase training behavior.
>
> In Sec. 4.1-4.3, we present our formal theoretical results in general case with corresponding experimental evidence to support our two-phase picture.
>
> **Q7: Concerns about Proposition 4.1. “Proposition 4.2 considers the case in which there is only one parameter and has almost no practical usefulness.”**
>
> **A7**: Thank you for this valuable comment. In fact, generalizing Proposition 4.1 beyond $p=1$ presents significant challenges. Extending Proposition 4.1 to higher dimensions requires a more comprehensive definition of sub-quadratic properties in high-dimensional settings. This necessitates a deeper exploration of the loss landscape, which lies beyond the scope of our primary focus on SAM's dynamics and is left for future work.
>
> **Q8: “Theorem 4.2 directly implies Theorem 4.1, so the latter should be a corollary.”**
>
> **A8**: Thank you for pointing out this problem. In fact, the proof of Theorem 4.2 builds upon the results of Theorem 4.1. To reflect this dependency, we have revised Theorem 4.2 into Corollary 4.2.

---

> ### Author Response · Authors · 2024-11-21
> **Response to Reviewer S3eC (3/3)**
>
> **Q9: Concerns about Assumptions 4.1-4.3.“I am not totally convinced that Assumptions 4.1-4.3 will be actually satisfied in practical situations. To be fair, many supporting references are provided in this paper.”**
>
> **A9**: Thank you for this fair comment. As noted, these assumptions are widely adopted in previous theoretical studies, and we have also cited many supporting references to demonstrate their relevance. Verifying whether these assumptions hold for deep neural networks is an interesting direction for future research and may have independent significance beyond our current focus.
>
> **Q10: Request for loss curves in more update steps. “I expected to see the loss going down in later steps in Figure 4 (b). Does the figure not show the entire steps?”**
>
> **A10**: Thank you for your question. Following your suggestion, we have extended the loss curve to include more update steps. In [Figure R.4](https://anonymous.4open.science/api/repo/SAM-Stability-ICLR2025-rebuttal-6B51/file/reb_fig4.pdf?v=9ca10dc6), we observe that the training loss initially increases exponentially fast and then gradually decreases, consistent with our theoretical predictions.
>
> **Q11: Overstated claims. “The result of Theorem 4.1 seems to be a little overstated … Theorem 4.1 does not show SAM converges although Theorem 4.3 does … Could the authors revise the sentence to a more accurate one?”**
>
> **A11**: Thank you for pointing out this problem. We have revised our claim P3 to “SAM selects a flatter minimum compared to SGD.”
>
> **Q12: “The paper uses the symbol $\approx$, but its mathematical definition is missing. Could the authors add the precise meaning?”**
>
> **A12**: Thank you for this question. First, we would like to emphasize that the symbol $\approx$ is not used in any formal theorems or proofs. In the main text, the symbol $\approx$ is used for two primary reasons:
>
> - **Ease of Understanding**: The symbol $\approx$ simplifies the reading flow. For example, $\theta_0\approx\theta^\star$ in line 402 indicates that $\theta_0$ is close to $\theta^\star$, which is formally defined in Proposition 4.1.
> - **Approximate Observations**: The symbol $\approx$ is also used to convey results that are not yet strictly established in the literature. For example, in line 362, the $\approx$​ is used to denote that the experimental phenomenon observed is approximate, as discussed in Wu et al. (2022), and has not yet been rigorously proven.
>
> Following your suggestion, we have added additional commentary in the revised paper to explicitly explain the context in which $\approx$ is used.
>
> **Q13: Suggestion on the proof of Proposition 4.1. “Proof of Proposition 4.1 could be improved by adding comments on which part the authors used the assumptions.”**
>
> **A13**: Thank you for your suggestion. We have revised the proof of Proposition 4.1 to explicitly highlight where each assumption is applied. We believe this improvement will help readers follow the reasoning more easily.
>
> **Q14: “Does this refer to Eq. (3)? If so, it should be clearly stated: "Our experiments are conducted using the original implementation of SAM"”**
>
> **A14**: We appreciate your question. However, we are uncertain about the specific context that “this” refers to in your comment. Could you kindly provide further clarification or specify the exact lines where this concern arises? This will help us address your question accurately.
>
> ---------------------------------------------
> ### Reference
>
> [1] Implicit Regularization in Matrix Factorization. In NeurIPS, 2017.
>
> [2] Neural Tangent Kernel: Convergence and Generalization in Neural Networks. In NeurIPS, 2018.
>
> [3] Gradient Descent Provably Optimizes Over-parameterized Neural Networks. In ICLR, 2019.
>
> [4] Computational-Statistical Gaps in Gaussian Single-Index Models. In COLT, 2024.
>
> [5] Gradient Descent on Two-layer Nets: Margin Maximization and Simplicity Bias. In NeurIPS, 2021.

---

> > ### Comment · Reviewer_S3eC · 2024-11-21
> >
> > I thank the authors for their replies to my comments. I'd like to give a quick response to the following question from the authors:
> >
> > > Q14: “Does this refer to Eq. (3)? If so, it should be clearly stated: "Our experiments are conducted using the original implementation of SAM"”
> > >
> > > A14: (...) we are uncertain about the specific context that “this” refers to in your comment. Could you kindly provide further clarification or specify the exact lines where this concern arises?
> >
> > "This" refers to "the original implementation of SAM" from line 138.

---

> > > ### Comment · Reviewer_S3eC · 2024-11-21
> > >
> > > > Q1: “Eq. (4) and the update rule that the authors use in theory (the two-step update rule defined at the beginning of C.1) are not equivalent.”
> > >
> > > > A1: (...) We acknowledge that there is indeed a difference between these two formulations. Specifically, the update rule used in Appendix C.1 assumes independence between the randomness in the inner and outer updates of SAM. This simplification is a common technical approach, as decoupling the randomness in SAM's updates is challenging. Notably, this technique has also been employed in the classic work on understanding SAM (see Appendix C.2 of Andriushchenko & Flammarion (2022)). We agree that analyzing the original formulation of SAM is an intriguing direction for future research.
> > >
> > > If the results proven in this paper are for something different from Eq. (4), it should be clearly stated. Otherwise, the results would be incorrect. I request the authors to revise the paper so that the reader will be able to easily understand which method each theorem is talking about. Saying "SAM" or "the original SAM" is not clear enough to me.

---

> > > > ### Author Response · Authors · 2024-11-21
> > > >
> > > > Thank you for the suggestion. We will revise our preliminary section to explicitly state the actual SAM update rule used in our theoretical analyses and the assumption of the independence between the randomness in the inner and outer updates.

---

> > > > > ### Comment · Reviewer_S3eC · 2024-11-24
> > > > >
> > > > > > our preliminary section
> > > > >
> > > > > Assuming the authors refers to lines 141--150 by that, I believe my comment was misunderstood.
> > > > >
> > > > > The proofs study the update rule given by the first two equations of Appendix C.1
> > > > > $$
> > > > > \text{(i)\quad} \boldsymbol{\theta}^{t+1 / 2} =\boldsymbol{\theta}^t+\rho \nabla \mathcal{L}\_{\xi\_t^1}\left(\boldsymbol{\theta}^t\right)
> > > > > $$ and
> > > > > $$
> > > > > \text{(ii)\quad} \boldsymbol{\theta}^{t+1} =\boldsymbol{\theta}^{t+1 / 2}-\eta \nabla \mathcal{L}\_{\xi_t^2}\left(\boldsymbol{\theta}^{t+1 / 2}\right).
> > > > > $$
> > > > > Plugging (i) into (ii) gives me
> > > > > $$
> > > > > \boldsymbol{\theta}^{t+1}
> > > > > = \boldsymbol{\theta}^t+\rho \nabla \mathcal{L}\_{\xi\_t^1}\left(\boldsymbol{\theta}^t\right)
> > > > >  -\eta \nabla \mathcal{L}\_{\xi_t^2}\left( \boldsymbol{\theta}^t
> > > > > +\rho \nabla \mathcal{L}\_{\xi\_t^1}\left(\boldsymbol{\theta}^t\right) \right),
> > > > > $$
> > > > > which is different from Eq. (7):
> > > > > $$
> > > > > \boldsymbol{\theta}^{t+1}
> > > > > =\boldsymbol{\theta}^t-\eta \mathcal{L}\_{\xi\_t^2}\left(\boldsymbol{\theta}^t+\rho \nabla \mathcal{L}_{\xi\_t^1}\left(\boldsymbol{\theta}^t\right)\right).
> > > > > $$
> > > > >
> > > > > For this reason, I consider the current form of the theorems incorrect, and I would have to lower my score if the authors do not address this issue. Furthermore, it is not easy to understand what update rule each theorem concerns, if the reader misses or forgets lines 141--150. The presentation is not very nice either in this sense.
> > > > >
> > > > > Also, Karakida et al. (2023) seems like an important reference for this work but is missing.
> > > > >
> > > > > **Reference**
> > > > > Ryo Karakida, Tomoumi Takase, Tomohiro Hayase, and Kazuki Osawa. Understanding Gradient Regularization in Deep Learning: Efficient Finite-Difference Computation and Implicit Bias. In Proceedings of the 40 th International Conference on Machine Learning, 2023.

---

> > > > > > ### Author Response · Authors · 2024-11-24
> > > > > >
> > > > > > Thank you for your quick response!
> > > > > >
> > > > > > Sorry for our misunderstanding! We have just noticed this incorrect formulation in the first two equations of Appendix C.1. This mistake indeed came from a typo. It is just because we previously ignored the typo, we misunderstood your concerns. Now, we have fixed this issue, along with other two problems you raised, in our revised paper. We have highlighted the additional revisions in **red**. Here, we provide a summary of our revisions:
> > > > > >
> > > > > > - **Theorem 4.1, Corollary 4.2**: The typos have been fixed. The correctness of Theorem 4.1 and Corollary 4.2 are not influenced by this typo.
> > > > > > - **Proposition 4.1**: The proof of Proposition 4.1 are indeed based on the correct formulation.
> > > > > > - **Theorem 4.3**: Some minor updates have been made to the intermediate steps of the proof. The final conclusion are still the same. See line 1575-1583.
> > > > > > - **Direct reference in each theorems.** For better presentation, we have explicitly added the reference to the exact update rule of SAM that our theoretical analyses consider in each theorems.
> > > > > > - **Missing reference**: Thank you for pointing this out. We reviewed the paper you mentioned, which studied the implicit bias of Gradient Regularization towards favorable solutions in rich regimes. We have discussed this paper in the related work section in our revised paper.
> > > > > >
> > > > > > Again, we sincerely appreciate your careful review and valuable feedback. We hope our response could adequately address your concerns. If you have any further questions, please us know, we will fix them ASAP.

---

> > > > > > > ### Comment · Reviewer_S3eC · 2024-11-24
> > > > > > >
> > > > > > > Thank you for the update. The new version indeed looks much better.
> > > > > > >
> > > > > > > An additional question: do you assume $\xi^1_1 \cup \xi^2_1, \dots, \xi^1_T \cup \xi^2_T$ are disjoint? Because the proof seems to use the independence between $\theta_{t}$ and $\xi_{t}^1 \cup \xi_{t}^2$.
> > > > > > >
> > > > > > > This is important because it would mean we can only run the iterations for 1 epoch (no reuse of a data point is allowed).

---

> > > > > > > > ### Author Response · Authors · 2024-11-24
> > > > > > > >
> > > > > > > > Thank you for your quick response! We are glad to answer your additional question.
> > > > > > > >
> > > > > > > > > An additional question: do you assume $\xi^1_1 \cup \xi^2_1, \dots, \xi^1_T \cup \xi^2_T$ are disjoint? Because the proof seems to use the independence between $\theta_{t}$ and $\xi_{t}^1 \cup \xi_{t}^2$.
> > > > > > > >
> > > > > > > > No, we do not assume $\xi^1_1 \cup \xi^2_1, \dots, \xi^1_T \cup \xi^2_T$ are disjoint. To clarify, we assume independence between $\xi_t^{1}$ and $\xi_t^{2}$, where both are batch indices sampled from the training set. This independence means that the mini-batches are sampled independently from the same distribution, but it does not imply that the sampled sets are disjoint. This assumption is a common technical approach (see Appendix C.2 of Andriushchenko & Flammarion (2022)), as decoupling the randomness in SAM's updates is challenging.
> > > > > > > >
> > > > > > > > Again, If you have any further questions, please us know, we will fix them ASAP.

---

> > > > > > > > > ### Comment · Reviewer_S3eC · 2024-11-24
> > > > > > > > >
> > > > > > > > > Then, how did you conclude the cross term is zero in line 1285 without the independence between $\theta_t$ and $\xi^B(\theta^{t+1/2})$? Could you write down the calculation?

---

> > > > > > > > > > ### Author Response · Authors · 2024-11-24
> > > > > > > > > >
> > > > > > > > > > We thank the reviewer for the interest in proof details. We are glad to answer your additional question.
> > > > > > > > > >
> > > > > > > > > > We would like to further clarify that we assume the independence of all $\xi_1^1,\xi_1^2,\cdots,\xi_t^1,\xi_t^2$ during the iteration. Recalling the update rule of SAM, we can express $\theta^t-\nabla L(\theta^{t+1/2})=\phi(\theta^0;\xi_1^1,\xi_1^2,\cdots,\xi_t^1)$. Additionally, $\xi^B(\theta^{t+1/2})=\xi_t^2$. Thus, $\theta^t-\nabla L(\theta^{t+1/2})$ and $\xi^B(\theta^{t+1/2})$ are independent.
> > > > > > > > > > Such properties are widely utilized in the literature on analysis of stochastic algorithms [1].
> > > > > > > > > >
> > > > > > > > > > If you have any further questions, please us know, we will fix them ASAP.
> > > > > > > > > >
> > > > > > > > > > [1] Elad Hazan et al. Introduction to online convex optimization. Foundations and Trends® in Optimization, 2(3-4):157–325, 2016.

---

> > > > > > > > > > > ### Comment · Reviewer_S3eC · 2024-11-24
> > > > > > > > > > >
> > > > > > > > > > > Is $\mathbb{E}$ taken conditionally on $\mathcal{D}\_\text{train}$? I only agree that the two vectors you mention are independent _conditionally_ on $\mathcal{D}\_\text{train}$.
> > > > > > > > > > >
> > > > > > > > > > > I don't know if the paper is clear about this point. The notation could be improved too.
> > > > > > > > > > >
> > > > > > > > > > > (By the way, I don't think $\boldsymbol{\xi}^B\left(\theta^{t+1 / 2}\right)=\xi_t^2$. The former is the "gradient noise" depending on $\mathcal{D}\_\text{train}$ and the latter is a set of indices.)

---

> ### Author Response · Authors · 2024-11-21
>
> Thank you for the quick response! Yes, it refers to Eq. (3). We will revise line 138 into "Our experiments are conducted using the original implementation of SAM (see Eq. (3))."

---

> ### Author Response · Authors · 2024-11-25
>
> Yes, we clarify that $\mathbb{E}$ is taken with respect to the random variables $\xi_t^1$, $\xi_t^2$ ($t\geq0$), which are the indices of independently random examples sampled from $\mathcal{D}_{\mathrm{train}}$.
>
> We apologize for the potential confusion caused by the notation and re-clarify it:  $\boldsymbol{\xi}\^B(\boldsymbol{\theta}\^{t+1/2}):=\mathcal{L}\_{\xi\_t\^2}(\boldsymbol{\theta}\^{t+1/2})-\nabla\mathcal{L}(\boldsymbol{\theta}\^{t+1/2})$.
> Observing that $\boldsymbol{\theta}^t-\eta \nabla \mathcal{L}(\boldsymbol{\theta}^{t+1/2})=\phi(\xi\_1\^1,\xi\_1\^2,\cdots,\xi_t\^1)$ and $\boldsymbol{\theta}^{t+1/2}=\psi(\xi\_1\^1,\xi\_1\^2,\cdots,\xi_t\^1)$,  we can use the law of total expectation to derive
> $$
> \begin{align}
> \mathbb{E}[\left<\boldsymbol{\xi}\^B(\boldsymbol{\theta}\^{t+1/2}),\boldsymbol{\theta}^t-\eta \nabla \mathcal{L}(\boldsymbol{\theta}^{t+1/2})\right>]
> &=\mathbb{E}\_{\xi\_1\^1,\xi\_1\^2,\cdots,\xi\_t\^1}\Big[\mathbb{E}\_{\xi\_t\^2}\big[\left<\nabla\mathcal{L}\_{\xi\_t\^2}(\boldsymbol{\theta}\^{t+1/2})-\nabla\mathcal{L}(\boldsymbol{\theta}\^{t+1/2}), \boldsymbol{\theta}^t-\eta \nabla \mathcal{L}(\boldsymbol{\theta}^{t+1/2})\right>\big| \xi\_1\^1,\xi\_1\^2,\cdots,\xi\_t\^1 \big]\Big]\\\\
> &=\mathbb{E}\_{\xi\_1\^1,\xi\_1\^2,\cdots,\xi\_t\^1}\Big[\mathbb{E}\_{\xi\_t\^2}\big[\left<\nabla\mathcal{L}\_{\xi\_t\^2}(\psi(\xi\_1\^1,\xi\_1\^2,\cdots,\xi_t\^1))-\nabla\mathcal{L}(\psi(\xi\_1\^1,\xi\_1\^2,\cdots,\xi_t\^1)), \phi(\xi\_1\^1,\xi\_1\^2,\cdots,\xi_t\^1)\right>\big| \xi\_1\^1,\xi\_1\^2,\cdots,\xi\_t\^1 \big]\Big]\\\\
> &=\mathbb{E}\_{\xi\_1\^1,\xi\_1\^2,\cdots,\xi\_t\^1}[0]=0.\\\\
> \end{align}
> $$
>
> PS: If no further concerns remain, we will really appreciate it if you could update the confidence score to reflect those resolutions in our responses. Thank you!

---

> > ### Comment · Reviewer_S3eC · 2024-11-25
> >
> > Thank you. It is clear now. I encourage you to add a remark or change the notation when you revise the paper next time.
> >
> > I think all of my major concerns have been addressed, and I have no more questions. I thank the authors for the careful answers to my questions. I would like to update my Rating score from 6 to 7.

---

> > > ### Author Response · Authors · 2024-11-26
> > >
> > > Thank you for your constructive feedback and for taking the time to engage with our work so thoroughly. We greatly appreciate your thoughtful questions and your updated evaluation.
> > >
> > > We will add a remark in the proof of Theorem 4.1 in our next revision.

---

### Official Review · Reviewer_bzn5 · 2024-11-03

**Soundness:** 3
**Presentation:** 4
**Contribution:** 3
**Rating:** 8
**Confidence:** 3

**Summary:**

The paper studies effects of SAM at the end of training upon an SGD baseline, and claims that using SAM for the final stages of training is sufficient to improve generalization. Considering a trajectory where SGD is switched to SAM at epoch $t$, the paper's theoretical analyses show that SAM rapidly escapes the SGD minima, while staying in the same valley. SAM then proceeds to converge rapidly to a new minimum that is flatter. These claims are supported by empirical evidence, and the authors posit a general conjecture that these loss objectives are only necessary toward the end of training.

**Strengths:**

While the empirical observations originate elsewhere, the theoretical analysis appears to be novel and interesting.
While the analyses are limited to the interplay between SGD and SAM during the later phases of training, it is sound and, this line of reasoning can conceivably generalize to other learning algorithms of interest, allowing the potential significance to be large.
The paper is clearly structured and easy to follow. The authors introduce a well-defined scope with thorough justification, both theoretically and empirically, and clearly discuss the motivation and role of assumptions. They then propose and support empirically a few general impacts of these insights to broader aspects of optimization, such as the role of SAM early-in-training and adversarial training late-in-training.

**Weaknesses:**

1. For some experiments (Fig 9) using SGD toward the end of training outperformed SAM in terms of accuracy, which is mildly contradicting toward an implicit claim that at the end-of-training, SAM outperforms SGD. Some commentary in the main section on these results would be appreciated.
2. The size of the valley in Proposition 4.1 for P2, $(-2b,2b)$, appear extremely wide when the parameter is initialized between $(-b, b)$, which weakens the significance of this proof. The empirical evidence (Fig 4c) for this claim also appears limited. Can a more general visualization of the loss landscape be considered?

**Questions:**

1. In 'Support for P3', can you support why increasing the loss might lead to flatter minima and better generalization? The empirical evidence suggests that SAM decreases the training loss, in addition to test loss.

**Details Of Ethics Concerns:**

No ethics review needed - this paper studies an optimization algorithm.

---

> ### Author Response · Authors · 2024-11-21
> **Response to Reviewer bzn5 (1/2)**
>
> Thank you for your great efforts on the review of this paper and for appreciating our novelty and contributions! We will try our best to address your questions.
>
> **If you still have further concerns, or if you are not satisfied by the current responses, please let us know, so that we can update the response ASAP.**
>
> **Q1: “For some experiments (Fig 9) using SGD toward the end of training outperformed SAM in terms of accuracy, which is mildly contradicting toward an implicit claim that at the end-of-training, SAM outperforms SGD. Some commentary in the main section on these results would be appreciated.”**
>
> **A1**: Thank you for your insightful suggestion. In our revised paper, we have included additional commentary in the main section to address the mild contradiction you mentioned. We acknowledge that in Fig. 9, when SAM training proportion approaches one, switching from SAM to SGD may slightly outperform full SAM training in terms of accuracy.
>
> - However, we would like to point out that accuracy is a discrete variable and can exhibit small fluctuations, making it a less stable measure. For example, in Fig. 9 for ResNet-20 on CIFAR-10, the switching method only outperforms full SAM training by approximately $10^{-4}$ in accuracy, which we consider to be a minor fluctuation.
> - In contrast, when examining test loss, we observe consistent behavior, with full SAM training continuing to outperform using SGD toward towards the end of training.
>
> **Q2: Concerns about Proposition 4.1. “The size of the valley in Proposition 4.1 for P2, (−2b,2b), appear extremely wide when the parameter is initialized between (−b,b), which weakens the significance of this proof.”**
>
> **A2**: Thank you for this constructive comment. In fact, the result can be extended to *almost all initializations within the valley $(-2b,2b)$*. Specifically, we can prove that a similar result to Proposition 4.1 holds for all initializations within $(-(2-\epsilon)b,(2-\epsilon)b), \forall \epsilon\in(0,1)$. By choosing $\epsilon$ sufficiently small, the result effectively applies to nearly all initializations within $(-2b,2b)$. We have included this generalized result for Proposition 4.1 in the revised paper.
>
> **Q3: Suggestions for more general loss landscape visualization. “The empirical evidence (Fig 4c) for this claim also appears limited. Can a more general visualization of the loss landscape be considered?”**
>
> **A3**: Thank you for your suggestion. We’d like to clarify that the one-dimensional linear interpolation presented in Fig. 4 (c) is widely adopted for visualizing the complex loss landscape of neural networks [1-3], and it is sufficient to support our key claim P2 that the SAM remains within the current valley during escape.
>
> **Additional Experiments**: To provide a more general visualization, we have conducted additional experiments using a two-dimensional visualization approach, as proposed by [4]. In this setup, we center the visualization at $\boldsymbol{\theta}\_{{\rm SGD \to SAM}, t}^T$ and select two directions: $\boldsymbol{\delta}\_{\rm target}$ and $\boldsymbol{\delta}\_{\rm random}$. Specifically, $\boldsymbol{\delta}\_{\rm target} = \boldsymbol{\theta}\_{{\rm SGD}}^T - \boldsymbol{\theta}\_{{\rm SGD \to SAM}, t}^T$, pointing towards the SGD solution, and $\boldsymbol{\delta}\_{\rm random}$ is a random direction sampled from $\mathcal{N}(0, \boldsymbol{I}_p)$ , with layer-wise normalization [4] applied. We then plot the loss landscape as a function $f(\alpha, \beta) = \mathcal{L}(\boldsymbol{\theta}\_{{\rm SGD \to SAM}, t}^T + \alpha \boldsymbol{\delta}\_{\rm target} + \beta \boldsymbol{\delta}\_{\rm random})$. In [Figure R.3](https://anonymous.4open.science/api/repo/SAM-Stability-ICLR2025-rebuttal-6B51/file/reb_fig3.pdf?v=a190c5f9) (please click this link to check the figure), it is clear that $\boldsymbol{\theta}\_{{\rm SGD \to SAM}, t}^T$ and $\boldsymbol{\theta}\_{{\rm SGD}}^T$ stay within the same valley. Moreover, the contours around $\boldsymbol{\theta}\_{{\rm SGD \to SAM}, t}^T$ is wider compared to $\boldsymbol{\theta}\_{{\rm SGD}}^T$, indicating $\boldsymbol{\theta}\_{{\rm SGD \to SAM}, t}^T$ corresponds to a flatter minimum. This two-dimensional approach provides additional insights into the structure of the loss landscape and further supports our two key claims P2 and P3.

---

> ### Author Response · Authors · 2024-11-21
> **Response to Reviewer bzn5 (2/2)**
>
> **Q4: “In 'Support for P3', can you support why increasing the loss might lead to flatter minima and better generalization? The empirical evidence suggests that SAM decreases the training loss, in addition to test loss.”**
>
> **A4**: We suspect you are asking about "the ascent update $\rho\nabla\mathcal{L}(\boldsymbol{\theta}_t)$ tends to increase the loss" in L348-349 in our "Support for P3".
>
> - **Intuitive Explanation**: Intuitively, the inner update $\boldsymbol{\theta}\_{t+1/2}=\boldsymbol{\theta}_t+\rho\nabla\mathcal{L}(\boldsymbol{\theta}_t)$ tends to increase the loss, making SAM more unstable (and more prone to divergence) than SGD. As a result, SAM must identify flatter minima to satisfy its stricter stability condition. Please kindly refer to Appendix C.3 for the formal proof.
> - **Inner vs. Outer Updates**: Notice that this claim about loss increase only pertains to the *inner update* of SAM, i.e., the comparison between $\mathcal{L}(\boldsymbol{\theta}_t)$ and $\mathcal{L}(\boldsymbol{\theta}\_{t+1/2})$. This is not inconsistent with observed outer loss decrease, i.e., the comparison between $\mathcal{L}(\boldsymbol{\theta}\_{t})$ and $\mathcal{L}(\boldsymbol{\theta}\_{t+1})$. In fact, our theory (see Theorem 4.3) and experiments demonstrate the convergence and loss descent of SAM.
>
> -------------------------------------------------
> ### Reference
>
> [1] Qualitatively Characterizing Neural Network Optimization Problems. In ICLR, 2015.
>
> [2] On Large-Batch Training for Deep Learning: Generalization Gap and Sharp Minima. In ICLR, 2017.
>
> [3] Linear Mode Connectivity and the Lottery Ticket Hypothesis. In ICML, 2020.
>
> [4] Visualizing the Loss Landscape of Neural Nets. In NeurIPS, 2018.

---

> > ### Comment · Reviewer_bzn5 · 2024-11-25
> >
> > I thank the authors for considering my comments carefully and addressing my concerns.
> > I maintain my original score on this paper which reflects the clarity and impact of this work.

---

### Official Review · Reviewer_cw28 · 2024-11-04

**Soundness:** 3
**Presentation:** 3
**Contribution:** 3
**Rating:** 6
**Confidence:** 3

**Summary:**

The authors explore how algorithm SAM converges to flat local minima and find a phase transition when switching SGD to SAM. By identifying the quick convergence property of SAM, the authors propose to use SAM in the later few epochs, which significantly reduces the computation of using SAM for flat local optima. The experimental results further verify the effectiveness of the proposed algorithm

**Strengths:**

1. The authors propose to use SAM in the last few epochs instead of starting from the beginning. The new method can reduce almost half of the computation of SAM while retaining the generalization ability of SAM.

2. The authors identify 2 phases of switching from SGD to SAM. With the theoretical analysis, the authors give 4 properties of SAM.

**Weaknesses:**

1. Some definitions are not clear.
     (i) Definition 4.1:  Does it mean that SAM is used to optimize linearized function but is evaluated on the real loss or that SAM both trains and tests on the linearized loss?

    (ii)  What is $a$ in Proposition 4.1, which is used to give an upper bound of $\rho$?

2. Is the first claim in Theorem 4.2 the contrapositive of Theorem 4.1? If not, can you explain the difference between these two claims?

3. After Theorem 4.1 and Theorem 4.2, the authors claim that SAM escapes from the sharp minima exponentially fast, which sounds like a good result. However, in Theorem 4.2, the result says that the final loss is $C^t$ larger than the initial one, which seems to be a bad result.  What is the relationship of these two results or the exponentially fast claim should be derived from some other theorem?

4. When the authors want to prove that SAM can stay in the same valley as SGD, there is only one valley in Proposition 4.2. Thus, to show SAM stays in the same valley, the function should have at least two valleys.

**Questions:**

1. Can the authors give a clear definition of linear stability? It seems to be a very important definition in the paper.

2. Can the authors provide proof when the function has two valleys, SAM will not go from one valley to another? In my view, Proposition 4.1 can not support iterates generated by SAM stay in the same valley.

3. What are the advantages of SAM shown in Theorem 4.3? SGD converges at the same rate under the Assumptions 4.2 and 4.3.

---

> ### Author Response · Authors · 2024-11-21
> **Response to Reviewer cw28 (1/2)**
>
> Thank you for your great efforts on the review of this paper and for your constructive comments on our work. We will try our best to address your questions.
>
> **If you still have further concerns, or if you are not satisfied by the current responses, please let us know, so that we can update the response ASAP.**
>
> **Q1: Suggestions for clarifications of Definition 4.1 (Linear Stability). “Does it mean that SAM is used to optimize linearized function but is evaluated on the real loss or that SAM both trains and tests on the linearized loss?” and “Can the authors give a clear definition of linear stability?”**
>
> **A1**: Thank you for this valuable suggestion.
>
> - **Clarification on the use of linearized loss**: First, we’d like to clarify that the linear stability theory, which involves the linearized loss, is used solely for theoretical analysis. All of our experiments are conducted on the actual training and testing loss, not the linearized version.
>
> - **Definition of linear stability**: Second, linear stability theory studies the training dynamics of SAM *near the global minima of training loss*. Specifically, near the global minimum $\boldsymbol{\theta}^\star$, the model $f$ can be approximated by its linearization $f\_{\rm lin}$ (see Eq. (5)). Consequently, the training loss can be expressed as $\mathcal{L}(\boldsymbol{\theta})=\frac{1}{2n}\sum\_{i=1}^n(f\_{\rm lin}(\boldsymbol{x}_i;\boldsymbol{\theta})-y_i)^2=\frac{1}{2}(\boldsymbol{\theta}-\boldsymbol{\theta}^\star)^\top \Big(\frac{1}{n}\sum\_{i=1}^n \nabla f(\boldsymbol{x}_i;\boldsymbol{\theta}^\star)\nabla f(\boldsymbol{x}_i;\boldsymbol{\theta}^\star)^\top\Big) (\boldsymbol{\theta}-\boldsymbol{\theta}^\star)$.
>   The linear stability analysis focuses on SAM's dynamics (Eq. (4)) when optimizing this linearized loss $\mathcal{L}(\boldsymbol{\theta})$​. This theoretical analysis demonstrates the flatness of the minima selected by SAM, which correlates with its generalization capability.
>
> In addition, following your suggestion, we have further clarified Definition 4.1 in our revised paper.
>
> **Q2: “What is $a$ in Proposition 4.1, which is used to give an upper bound of $\rho$?”**
>
> **A2**: Thanks for this inquiry. $a$ is defined in Definition 4.2 as the *sharpness* $\mathcal{L}''(\theta^\star)$ of the global minimum $\theta^\star$.
>
> **Q3: Concerns on Theorem 4.2. “Is the first claim in Theorem 4.2 the contrapositive of Theorem 4.1? If not, can you explain the difference between these two claims?”**
>
> **A3**: Thank you for this insightful question. Yes, Theorem 4.2 is indeed the contrapositive of Theorem 4.1. To clarify this relationship, we have revised Theorem 4.2 and rephrased it as Corollary 4.2 in the revised paper.
>
> **Q4: “After Theorem 4.1 and Theorem 4.2, the authors claim that SAM escapes from the sharp minima exponentially fast, which sounds like a good result. However, in Theorem 4.2, the result says that the final loss is $C^t$ larger than the initial one, which seems to be a bad result. What is the relationship of these two results or the exponentially fast claim should be derived from some other theorem?”**
>
> **A4**: Thank you for raising this concern.
>
> - **Definition of Exponentially Fast Escape**: We clarify that the definition of exponentially fast escape is precisely expressed as $\mathbb{E}[\mathcal{L}(\boldsymbol{\theta}_t)]\geq C^t \mathbb{E}[\mathcal{L}(\boldsymbol{\theta}_0)]$, where $C>1$. This reflects the rapid increase in loss during the escape phase, as captured in Theorem 4.2 (now Corollary 4.2).
> - **Escape then converge**: The exponential escape in Theorem 4.2 applies only to Phase I. Once SAM transitions into Phase II, the loss begins to descend, and SAM converges to a flatter minimum.
>
> **Q5: Concerns about Proposition 4.1. “When the authors want to prove that SAM can stay in the same valley as SGD, there is only one valley in Proposition 4.1. Thus, to show SAM stays in the same valley, the function should have at least two valleys.”  and “Can the authors provide proof when the function has two valleys, SAM will not go from one valley to another?”**
>
> **A5**: Thank you for this insightful question. We’d like to clarify that *multiple valleys can indeed exist* in the setting of Proposition 4.1:
>
> - Proposition 4.1 assumes the landscape is sub-quadratic within the valley $V=[-2b,2b]$, *but it does not impose any assumptions about the landscape outside $V$*. Therefore, additional valleys can exist in $(-\infty,-2b)\cup(2b,+\infty)$.
> - For example, consider $\mathcal{L}(\theta)=1-\cos(\frac{\pi\theta}{2b})$, which satisfies the sub-quadratic property within $V=[-2b,2b]$. This loss function has infinitely many valleys: $[2(2k-1)b,2(2k+1)b],k\in\mathbb{Z}$. By Proposition 4.1, SAM remains in the current valley $V=[-2b,2b]$ and cannot enter into other valleys.
>
> We have added this clarification to the revised paper to address potential misunderstandings.

---

> ### Author Response · Authors · 2024-11-21
> **Response to Reviewer cw28 (2/2)**
>
> **Q6: “What are the advantages of SAM shown in Theorem 4.3? SGD converges at the same rate under the Assumptions 4.2 and 4.3.”**
>
> **A6**: Thank you for this insightful question.
>
> - **Clarification on Theorem 4.3's purpose**: First, we would like to clarify that Theorem 4.3 is not intended to demonstrate the advantages of SAM over SGD. Instead, the advantage of SAM is primarily established in Theorem 4.1, which shows that SAM selects flatter minima than SGD, leading to better generalization. Instead, Theorem 4.3 aims to support "P4: The convergence rate of SAM is extremely fast" in Phase II (see Tab. 1). Notably, Theorem 4.3 establishes a faster convergence rate of SAM compared to the classic result from Andriushchenko & Flammarion, (2022).
>
> - **Comparison to SGD**: As noted by the reviewer, under the same conditions, SGD and SAM exhibit the same convergence rate. Indeed, there is currently no consistent empirical evidence suggesting that SAM outperforms SGD in terms of convergence rate.

---

> > ### Comment · Reviewer_cw28 · 2024-11-26
> >
> > Thank the authors for the detailed explanation of the questions. I have no further questions and raise my score.

---

> ### Author Response · Authors · 2024-11-24
> **Looking forward your feedback**
>
> Dear Reviewer cw28,
>
> We hope that our responses could adequately address your concerns. As the deadline of this discussion phase is approaching, we warmly welcome further discussion regarding any additional concerns that you may have, and we sincerely hope you can reconsider the rating accordingly.
>
> Thank you for the time and appreciation that you have dedicated to our work.
>
> Best regards,
>
> Authors of submission 1483

---

### Official Review · Reviewer_WFBm · 2024-11-05

**Soundness:** 4
**Presentation:** 3
**Contribution:** 4
**Rating:** 8
**Confidence:** 3

**Summary:**

The authors demonstrate that effects of sharpness-aware minimization (SAM) are independent of whether it is applied in early training. They show that when applied late, SAM-driven iterates first escape the narrow minimum found by SGD, and then find a wider minimum. The authors demonstrate that the earlier application of SAM has no discernible effect on the solution. Inspired by these results, the authors find a similar pattern in adversarial training (AT) as well.

**Strengths:**

- The paper's findings are novel, important, while also counterintuitive given previous research's emphasis on early training. The findings are likely to be of interest to research communities that focus on SAM in specific, and training dynamics and generalization in general.
- The authors make a clear and convincing case regarding the importance of adopting SAM in late (vs. early training) using empirical and theoretical findings.
- The paper is written well and is easy to follow, with the authors' visualizations being especially instructive. Theoretical results are promptly compared with previous results in the literature.
- The extension of the authors' results to adversarial training both supports their original results and also is an important contribution in and of itself.

**Weaknesses:**

- The divergence between paper's experimental and theoretical setting is notable especially in the choice of the loss function. This is acknowledged in the paper and is acceptable given the novelty of their results.
- Although the paper is written well overall, certain unclearly defined concepts, choice of notation, and design choices make it hard to follow at times. I provide more detailed feedback regarding these in the section below.

**Questions:**

- L000: The concepts of early vs. late training are used throughout the training, but is not given clear definitions. Please commit to an explicit definition (in accordance with L306), or discuss why this is not necessary/possible. Doing so earlier in the paper is important.
- L081: Efficient sounds like a technical ter
- L110: The notation for dataset and distribution are the same, $\mathcal{S}$. Please differentiate, and differentiate their use as subscripts to $\mathcal{L}$ as well (as this is almost exclusively used with distributions.)
- L115: Please define classification error explicitly.
- L177: "when switches from SGD to SAM"
- L177: The ordering of the x-axes in Figure 3 is confusing. I understand that the authors aimed for consistency between Figures 3, 5, and 6; but I think their choice makes the visualization less legible. I believe having x axes stand for changing time will make the graph easier to read. I ultimately leave this choice to authors' discretion.
- L286: Why change $T$ and $t$'s usage?
- L291: Typo after $\mathcal{L}''$
- L318: Is there a reason for usage of $\ell$ here?
- L352: After changing the use of $T$ and $t$ at L286, Theorems 4.1 and 4.2 changes this use one more time (from $T$ to $T + t$ to $0$ to $t$). Can these be made more consistent?
- L354: Please alert the reader that you will introduce Proposition 4.1 to resolve the conflict between the exponential escape and low-loss assumption
- L388: Do these results generalize to cases where $p>1$ and minibatch SAM?
- L406: Can we assume any valley around $\theta^*$ found by SGD to be sub-quadratic?
- L524: Are you making a claim about robustness-accuracy trade-off? I.e. do you expect robustness to increase without sacrificing vanilla accuracy if the AT is started late?

---

> ### Author Response · Authors · 2024-11-21
> **Response to Reviewer WFBm (1/2)**
>
> Thank you for your great efforts on the review of this paper and for appreciating our novelty and contributions! We will try our best to address your questions.
>
> **If you still have further concerns, or if you are not satisfied by the current responses, please let us know, so that we can update the response ASAP.**
>
> **Q1: Concerns about the limitation of our theory regarding the choice of loss function. “The divergence between paper's experimental and theoretical setting is notable especially in the choice of the loss function. This is acknowledged in the paper and is acceptable given the novelty of their results.”**
>
> **A1**: Thank you for appreciating the novelty of our work. The use of the squared loss function is a common practice in theoretical analyses of neural networks, and it facilitates the study of various aspects including implicit bias [1], training dynamics [2], and convergence [3]. While this choice simplifies our theoretical framework,  our theoretical results remain representative for understanding the training dynamics and generalization behavior of neural networks.
>
> **Additional Experiments:** To further close the gap between our experimental and theoretical settings, we have conducted additional experiments to reproduce our main discovery in Sec. 3 using the squared loss. Consistently with the main paper, we measured how the generalization gap between $\boldsymbol{\theta}\_{{\rm SGD \to SAM}, t}^T$ and $\boldsymbol{\theta}\_{{\rm SAM}}^T$, as well as the sharpness of $\boldsymbol{\theta}\_{{\rm SGD \to SAM}, t}^T$, changes as the SAM training proportion $\frac{T-t}{T}$ increases. In [Figure R.6](https://anonymous.4open.science/api/repo/SAM-Stability-ICLR2025-rebuttal-6B51/file/reb_fig6.pdf?v=448aef37) (please click the link to check the figure), both the generalization gap and sharpness follow trends consistent with our previous findings in Fig. 3, thereby bridging the gap between our theoretical and experimental results.
>
> **Q2: Writing problems. “Although the paper is written well overall, certain unclearly defined concepts, choice of notation, and design choices make it hard to follow at times. I provide more detailed feedback regarding these in the section below.”**
>
> **A2**: Thank you for your careful review. Your feedback is invaluable in helping us improve our paper. We have carefully revised the paper to address the writing problems you highlighted. Below, we outline the key revisions made:
>
> - **Clarification of Concepts** (L000, L115): Additional explanations and definitions to ensure clarity and accessibility.
> - **Improved Notation** (L110, L286, L352): Streamlined notations for better consistency and readability.
> - **Corrections** (L177, L291, L354): Fixed typos and other minor issues.
>
> Regarding the other specific points you raised:
>
> - L081: We kindly request further elaboration on this issue to ensure we fully understand your concern and address it appropriately.
> - L177 (X-axes): To maintain consistency across Fig. 3, 5 and 6-9 and facilitate comparison, we have decided to retain the current x-axis formatting.
> - L318: Following the notation $\ell\_i(\boldsymbol{\theta})$, which denotes the loss on the $i$-th sample (see Line 113), $\nabla \ell\_{\xi}(\boldsymbol{\theta})$ represents the 1-batch loss gradient (at $\boldsymbol{\theta}$), where $\xi$ denotes the index of a random sample.
>
> We hope our revision and explanations can address your concerns. Thank you again for your constructive feedback.
>
> **Q3: Questions about Proposition 4.1. “L388: Do these results generalize to cases where $p>1$ and minibatch SAM?”**
>
> **A3**: Thank you for the insightful question. In fact, generalizing Proposition 4.1 beyond $p=1$ and full-batch SAM presents significant challenges:
>
> - **Extending to $p>1$**: Generalizing Proposition 4.1 to higher dimensions would require a more comprehensive definition of sub-quadratic properties in high-dimensional settings. This necessitates a deeper exploration of the loss landscape, which lies beyond the scope of our primary focus on SAM's dynamics and is left for future work.
> - **Mini-batch SAM**: The sub-quadratic property pertains to the overall loss $\mathcal{L}(\boldsymbol{\theta})$ across the entire training set and does not impose assumptions on the mini-batch loss $\ell_i(\boldsymbol{\theta})$. Therefore, extending our analysis to mini-batch SAM would require characterizing the landscape of $\ell_i(\boldsymbol{\theta})$, which introduces additional complexities and is not directly addressed in our current framework.
>
> **Q4: Questions about the sub-quadratic property. “L406: Can we assume any valley around $\boldsymbol{\theta}^\*$ found by SGD to be sub-quadratic?”**
>
> **A4**: Thank you for your question. As mentioned above, research on the sub-quadratic property is still in its early stages, and its definition in high-dimensional settings remains unclear. Answering this question will require a deeper investigation into the structure of the loss landscape.

---

> ### Author Response · Authors · 2024-11-21
> **Response to Reviewer WFBm (2/2)**
>
> **Q5: Questions about the robustness-accuracy trade-off. “L524: Are you making a claim about robustness-accuracy trade-off? I.e. do you expect robustness to increase without sacrificing vanilla accuracy if the AT is started late?”**
>
> **A5**: Thank you for your question.
>
> - We are not making a claim about the robustness-accuracy trade-off in this paper, as it is not the primary focus of our work. Our experiments in Sec. 5 aim to show that Adversarial Training (AT) can efficiently improve robustness even when applied only in the final few epochs.
> - Regarding the second question, we do observe a slight drop in accuracy on clean samples when starting AT late in training. This trade-off is expected and aligns with findings reported in the literature [4, 5].
>
> ---------------------------------
> ### Reference
>
> [1] Implicit Regularization in Matrix Factorization. In NeurIPS, 2017.
>
> [2] Neural Tangent Kernel: Convergence and Generalization in Neural Networks. In NeurIPS, 2018.
>
> [3] Gradient Descent Provably Optimizes Over-parameterized Neural Networks. In ICLR, 2019.
>
> [4] Robustness May Be at Odds With Accuracy. In ICLR, 2019.
>
> [5] Theoretically Principled Trade-off between Robustness and Accuracy. In ICML, 2019

---

> > ### Comment · Reviewer_WFBm · 2024-11-24
> >
> > I thank the authors for their response and improvements to the paper. I will be maintaining my score and recommend acceptance. P.s. I had meant to remove L081 from my review, please safely ignore.

---

### Official Review · Reviewer_6K1k · 2024-11-12

**Soundness:** 3
**Presentation:** 2
**Contribution:** 3
**Rating:** 6
**Confidence:** 3

**Summary:**

The paper delves into the behavior of Sharpness-Aware Minimization (SAM) in the later stages of neural network training, demonstrating that SAM can effectively evade the sharp minima typically identified by Stochastic Gradient Descent (SGD) and instead settle into flatter minima. It outlines a two-stage learning process in which SAM initially hastens the departure from sharp minima and subsequently swiftly reaches flatter minima, a pattern observed even when SAM is employed only during the final training epochs. The research indicates that the selection of the optimization algorithm during the late training phase has a substantial impact on the ultimate attributes of the solution, underscoring the significance of scrutinizing optimization tactics at this critical juncture.

**Strengths:**

This paper offers an in-depth examination of Sharpness-Aware Minimization (SAM) throughout the training process, with a particular emphasis on its late-phase behavior, an area that has been understudied in prior research.
The paper elucidates a distinct two-phase learning pattern of SAM, illustrating its ability to swiftly exit sharp minima and subsequently settle into flatter minima, thereby enriching the theoretical grasp of SAM's optimization trajectory.

The study underscores the pivotal impact of the optimization algorithm selected for the late training phase, proposing a potential optimization strategy that could reduce computational demands while sustaining optimal performance, thus offering practical relevance for deep learning applications in real-world contexts.

**Weaknesses:**

1.The theoretical results have limited applicability to real training scenarios. They do not sufficiently explain why transitioning the training method from SGD to SAM results in a large generalization gap compared to using the full SAM training method. The experiments in the paper demonstrate that switching from SAM to SGD often leads to poorer results.

2.In theory, the switch from SGD to SAM is expected to yield certain benefits; however, no experimental evidence is provided to support this.

3.The models discussed are primarily convolution-based, so it would be beneficial to include some transformer-based models as well.

**Questions:**

Could the authors explain how the neighborhood size affects the results when switching from SAM to SGD?

---

> ### Author Response · Authors · 2024-11-21
> **Response to Reviewer 6K1k (1/2)**
>
> Thank you for your great efforts on the review of this paper and for recognizing the value of our contributions. We will try our best to address your questions.
>
> **If you still have further concerns, or if you are not satisfied by the current responses, please let us know, so that we can update the response ASAP.**
>
> **Q1: Concerns about practical applicability of our theoretical analysis. “They do not sufficiently explain why transitioning the training method from SGD to SAM results in a large generalization gap compared to using the full SAM training method. The experiments in the paper demonstrate that switching from SAM to SGD often leads to poorer results.”**
>
> **A1**: Thank you for your question. In Fig. 3, switching from SGD to SAM does not result in a large generalization gap compared to full SAM training. Based on your follow-up comment, we believe your concern may be about the practical applicability of our theoretical analysis to the case of switching from SAM to SGD.
>
> - **Clarification of the purpose of our theoretical analyses**: First, we would like to clarify that the primary goal of our theoretical analysis is to explain why SAM efficiently finds flatter minima compared to SGD when applied in the late stage of training. Our theoretical setup closely reflects the real-world settings, and in Fig. 4, our theoretical claims are well-supported by experiments conducted in practical scenarios. Therefore, our theoretical results are indeed applicable to real training scenarios and sufficiently explain our main discovery.
>
> - **Insights into the SAM-to-SGD**: Second, while our theoretical results do not directly explain the empirical observations when switching from SAM to SGD, they offer valuable insights into the underlying training behavior during this transition.
>
>   - In our theory (see Theorem 4.1 and Tab. 2), the global minima selected by SAM are inherently linearly stable for SGD. However, linear stability is a necessary but not sufficient condition for minima selection. It is possible that other conditions exist and drive SGD escapes the flatter minima found by SAM, resulting in relatively sharper minima and thus poorer generalization. Nevertheless, the relatively sharper minima are still required to be linear stable for SGD.
>   - In Sec. B.4.1, we explore how sharpness at the iterator evolves during training when we switches from SAM to GD. In Fig. 10, we observe that the sharpness rapidly increases once the switch occurs and then oscillates around $2/\eta$ (the linear stability condition for GD (Wu et al. 2018), also commonly known as the Edge of Stability phenomenon), regardless of how late the switch occurs. This implies that SGD immediately escapes the flat minima found by SAM and converges to relatively sharper minima, which aligns with the empirical observations in Sec. 5.
>
>   Together with our theory, we conjecture that other than linear stability, additional conditions govern the minima selection process and drive (S)GD towards relatively sharper solutions. We will leave the exploration of the conditions beyond linear stability as future directions.
>
>  **Q2: Misunderstandings regarding our main contribution. “In theory, the switch from SGD to SAM is expected to yield certain benefits; however, no experimental evidence is provided to support this.”**
>
> **A2**: We'd like to re-clarify the main contributions and logic progression of this paper:
>
> 1. **Sec. 3**: we empirically find that the solutions obtained by switching from SGD to SAM yield comparable generalization and sharpness than solutions found by full SAM training, even when the switch occurs a few epochs before the end of training.
> 2. **Sec. 4**: We theoretically establish a two-phase picture to explain the empirical findings. This picture is characterized by four key claims, all of which are well-supported by practical experiments.
>
> Together, the experimental evidence in Sec. 3 demonstrate the benefits of switching from SGD to SAM in terms of generalization and sharpness, and our theory in Sec. 4 provides a sufficient explanation for these benefits.

---

> ### Author Response · Authors · 2024-11-21
> **Response to Reviewer 6K1k (2/2)**
>
> **Q3: Suggestions for empirical evidence on transformer-based models. “The models discussed are primarily convolution-based, so it would be beneficial to include some transformer-based models as well.”**
>
> **A3**: Thank you for your suggestion. We have conducted **new experiments** on Vision Transformers (ViTs) to further verify that applying SAM in the last few epochs achieves generalization comparable to full SAM training. As Adam is a more common practice for training ViTs, we replace SGD with Adam before the switch. Then, consistently with the main paper, we measured how the generalization gap between $\boldsymbol{\theta}\_{{\rm Adam \to SAM}, t}^T$ and $\boldsymbol{\theta}\_{{\rm SAM}}^T$ changes as the SAM training proportion $\frac{T-t}{T}$ increases. As shown in [Figure R.1](https://anonymous.4open.science/api/repo/SAM-Stability-ICLR2025-rebuttal-6B51/file/reb_fig1.pdf?v=fc928695) (please click the link to check the figure), the generalization gap approaches zero once the SAM training proportion exceeds zero. This further strengthens our argument, confirming the efficiency of SAM during the late phase of training.
>
> **Q4: Question about the effect of perturbation radius when switching from SAM to SGD. “Could the authors explain how the neighborhood size affects the results when switching from SAM to SGD?”**
>
> **A4**: An interesting question. In our theory (see Theorem 4.1 and Tab. 2), the perturbation radius $\rho$  influences the sharpness of the global minima selected by SAM. Specifically, a larger perturbation radius leads to the selection of flatter global minima. In contrast, SGD's behavior is independent of the perturbation radius. However, as discussed in our conjecture in Sec. 5, the properties of the final solutions are primarily shaped by the optimization method chosen in the late training phase. Consequently, when switching from SAM to SGD, the generalization properties of the final solution are expected to be independent of the perturbation radius, as SGD dominate in the final stage of training.
>
> **Additional Experiments:** To further explore this, we have conducted new experiments to investigate the effect of perturbation radius on generalization when switching from SAM to SGD. Specifically, we varied the perturbation radius $\rho \in \\{0.02, 0.05, 0.1, 0.15, 0.2\\}$ for SAM and measured how the test loss and error of $\boldsymbol{\theta}\_{{\rm SAM \to SGD}, t}^T$ change as the SAM training proportion $\frac{t}{T}$ increases. In [Figure R.2](https://anonymous.4open.science/api/repo/SAM-Stability-ICLR2025-rebuttal-6B51/file/reb_fig2.pdf?v=af36cade), we observe no significant differences in test loss or error across different $\rho$ values, even when training with SAM for most of the time (e.g., $\frac{t}{T} = 0.8$). Notably, only full SAM training (i.e., $\frac{t}{T}=1$) shows a pronounced dependence on $\rho$, where a larger perturbation radius leads to better generalization. These findings align with our expectation that the perturbation radius has minimal impact on generalization when transitioning from SAM to SGD, further validating our theoretical results  in Sec. 4 and conjecture in Sec. 5.

---

> > ### Comment · Reviewer_6K1k · 2024-12-01
> >
> > Thank you for your detailed response, which addresses most of my concerns. I believe this is a strong paper that provides valuable insights into the SAM method, so I will maintain my score.

---

### Public Comment · ~Zhiwei_Jia1 · 2024-11-18
**An interesting work regarding applying SAM in the late training stage**

Hi authors,

Thanks for the great and insightful work! Please consider citing [1] which applies SAM-based techniques to encourage flatter minima. In its preliminary study, it also applies SAM only in the later stage of training for efficiency.

[1] Information-Theoretic Local Minima Characterization and Regularization, ICML 2020

---

> ### Author Response · Authors · 2024-11-18
>
> Hi Zhiwei,
>
> Thank you for your appreciation of the novelty and significance of our study!
>
> We reviewed the paper you mentioned, which introduces a sharpness-regularized training algorithm with a formulation similar to SAM. Notably, its ablation study demonstrates the algorithm's effectiveness when applied in the mid or late training stages. We will include a citation to this paper in our revision.
>
> Best regards,
>
> Authors of submission 1483

---

### Author Response · Authors · 2024-11-21
**Global Response**

Dear AC and reviewers,

We appreciate the great efforts of each reviewer in the reviewing process. We are encouraged that all reviewers acknowledged the **soundness** (4, 3, 3, 3, 3) and **contribution** (4, 3, 3, 3, 3) of our empirical discoveries and theoretical analyses. Specifically, the novelty and significance of our theoretical analyses regarding the two-phase picture are highlighted by Reviewer **WFBm**, **bzn5** and **S3eC**. The practical usage and implications of our empirical discovery—that the late training phase plays a more critical role for SAM in flat minima selection—are well-recognized by Reviewers **WFBm**, **cw28**, and **bzn5**. The originality of our contributions is particularly appreciated by Reviewer **6K1k**.

We have individually responded to each reviewer, trying our best to address their questions. Here, we provide a concise summary of our responses to some important/common concerns.

**Discrepancy between theoretical and experimental setups (WFBm, S3eC).** The discrepancy mainly arises from the choice of the squared loss function and the simplified SAM update rule in Eq. (4) for our theoretical analysis.

- **Clarification of theoretical choices**: First, we’d like to clarify that the use of the squared loss function is a common practice in theoretical analyses of neural networks, and the update rule in Eq. (4) has also been widely adopted in recent theoretical advances on SAM. These setups facilitate mathematical tractability while capturing the essential behavior of SAM in terms of training dynamics and generalization.
- **Bridging the gap**: Second, to further address this concern, we have conducted additional experiments to reproduce our main discovery in Sec. 3 using the squared loss and the simplified update rule of SAM. These results are consistent with our primary discovery, thus further validating the relevance of our theoretical framework to the observed phenomena.

**Concerns about Proposition 4.1. (WFBm, cw28, bzn5, S3eC).** The major concern about Proposition 4.1 come from its additional simplified setting, where we assume $p=1$. However, extending Proposition 4.1 to higher dimensions would require a more comprehensive definition of sub-quadratic properties in high-dimensional settings. While it is an interesting direction, this lies beyond the scope of our paper and is deferred to future work.

In addition, we have also revised Proposition 4.1 and include further commentary to address the specific concerns raised by Reviewer cw28 and bzn5.

**Writing problems (WFBm, cw28, S3eC).** We are happy to receive the feedback from each reviewer regarding the writing problems, such as improper notation usage, unclear definitions of important concepts, overstated claims, and other writing mistakes. We have carefully considered their suggestions and revised our paper accordingly.

**New experiments**: To enrich our analysis and explore our discovery in various settings, as suggested by the reviewers, we have conducted 6 new experiments during the discussion phase:

1. [Figure R.1](https://anonymous.4open.science/api/repo/SAM-Stability-ICLR2025-rebuttal-6B51/file/reb_fig1.pdf?v=fc928695), [Figure R.5](https://anonymous.4open.science/api/repo/SAM-Stability-ICLR2025-rebuttal-6B51/file/reb_fig5.pdf?v=6f25d852), [Figure R.6](https://anonymous.4open.science/api/repo/SAM-Stability-ICLR2025-rebuttal-6B51/file/reb_fig6.pdf?v=448aef37): Extends our main discovery in Sec. 3 to more settings: transformer-based models; simplified SAM; squared loss.
2. [Figure R.3](https://anonymous.4open.science/api/repo/SAM-Stability-ICLR2025-rebuttal-6B51/file/reb_fig3.pdf?v=a190c5f9): Provides a more general visualization of the loss landscape around SGD and SGD-to-SAM solutions.
3. [Figure R.2](https://anonymous.4open.science/api/repo/SAM-Stability-ICLR2025-rebuttal-6B51/file/reb_fig2.pdf?v=af36cade): Investigates how varying the perturbation radius affects generalization when switching from SAM to SGD.
4. [Figure R.4](https://anonymous.4open.science/api/repo/SAM-Stability-ICLR2025-rebuttal-6B51/file/reb_fig4.pdf?v=9ca10dc6): Re-run Fig. 4(b) and extends the loss curve to include more update steps.

**We have included all additional experimental results in our revised paper**.

Best regards,

Authors of submission 1483

---

### Author Response · Authors · 2024-11-21
**Paper Revision**

Dear AC and Reviewers,

Thank you for your patience.
We are currently finalizing the revisions to our paper and will complete them within **the next 24 hours**.

Best regards,

Authors of Submission 1483

---

### Author Response · Authors · 2024-11-22
**Paper Revision Complete!**

Dear AC and reviewers,

Thank you for your patience!

We have finalized the paper revision and updated to OpenReview system. All changes are highlighted in **orange** in the revised manuscript. Here, we provide a summary of our revisions:

- **Improved notations.** We have improved our notations for better consistency and readability. For example, we have unified the notation for datasets, as suggested by Reviewer WFBm.
- **Clarification of concepts and definitions.** We have added additional commentary to further clarify important concepts and definitions, such as the definition of linear stability and Proposition 4.1. We have also highlighted the difference between our theoretical and experimental setups regarding the update rule of SAM, as recommended by Reviewer S3eC.
- **New experiments.** We have included all of the additional experiments in our revised paper.
- **Other issues.** We have fixed some minor issues, such as typos and missing references.

We sincerely appreciate the reviewers’ valuable feedback and constructive suggestions, which helped improve the quality and presentation of our work. We are happy to provide further clarifications or address additional concerns.

Best regards,

Authors of submission 1483

---

> ### Author Response · Authors · 2024-12-01
> **Further Paper Revision**
>
> Dear AC and reviewers,
>
> We sincerely thank you for your valuable feedback and constructive suggestions. Following the recommendations from AC and Reviewer S3eC, we have further revised our paper. The additional changes are highlighted in **blue**. Below is a summary of the updates:
>
> - **Additional acknowledgement of Andriushchenko et al. (2022)**: As suggested by AC, we have included an additional commentary in the introduction section to explicitly acknowledge that Andriushchenko et al. (2022) made a similar finding.
> - **Clarifications in the proof of Theorem 4.1**: As suggested by Reviewer S3eC, we have
>   - Added **Remark C.1** in Sec. C.1 to clarify the use of $\mathbb{E}$.
>   - Added **Remark C.2** to explain that $\boldsymbol{\xi}^B(\boldsymbol{\theta}^{t+1/2})$ and $\boldsymbol{\theta}^t-\eta \nabla \mathcal{L}(\boldsymbol{\theta}^{t+1/2})$ are uncorrelated.
>
> Since authors cannot upload the revised paper directly to OpenReview, you may access the updated version [here](https://anonymous.4open.science/api/repo/ICLR-2025-SAM-Stability/file/_ICLR_2025__SAM_stability.pdf?v=10d2b9eb).
>
> We hope these updates address the concerns raised.
>
> Best regards,
>
> Authors of submission 1483

---

### Comment · Area_Chair_5gE3 · 2024-11-28
**Similarities with prior work**

Dear Authors,

Could you please clarify the similarities between your findings and the experiments presented in Andriushchenko et al. (2022),  Figure 9, and the paragraph titled The importance of the implicit bias of SAM at the end of training? It appears that this work may already cover similar findings.

I look forward to your response.

Thank you

Best regards,
The AC

---

> ### Author Response · Authors · 2024-11-29
>
> Dear AC,
>
> Thank you for your question. We indeed acknowledge in a separate paragraph of our paper (lines 221-224) that Andriushchenko et al. (2022) made a similar observation, *but* they did not give it significant attention. Our work isolates and extends this finding by showing that **even fewer epochs** of SAM (e.g., around 5 epochs for WideResNet-16-8 on CIFAR-10) applied at the end of training could achieve comparable test error and loss to full SAM training (see Fig. 3). More importantly, as noted by Reviewer 6K1k, our study provides an *in-depth* examination of SAM’s late phase behavior, both empirically and theoretically, an area that has been *understudied* in prior research. Here, we provide a concise summary of our extra contributions:
>
> - **More empirical investigations.**
>   - **New experiments regarding the sharpness.** Other than generalization ability, we investigate the effect of late-phase SAM on the sharpness of the final solution. This experiment *directly* verifies the implicit bias of SAM towards flatter minima when applied in the late phase of training.
>   - **Extensive experiments under various setups.** We conduct extensive experiments under various settings to justify the efficiency of late-phase SAM in selecting flatter minima. Our experiments cover various variants of SAM (e.g. vanilla SAM, ASAM and USAM), loss function (e.g., squared loss and cross entropy loss), architectures (e.g. convolution-based and transformer-based models), and datasets.
> - **Rigorous theoretical justifications.** Our theoretical analysis provably uncovers a two-phase picture in the training dynamics after switching to SAM in the late phase, which highlights the *distinct* value of our work, as acknowledge by Reviewer WFBm, bzn5 and S3eC.
>   - This two-phase picture is characterized by four key claims in Tab. 1, each rigorously supported by corresponding theorems or propositions. For example, our Corollary 4.2 explicitly prove the escaping behavior of SAM after the switch, which is also suggested by Andriushchenko et al. (2022). However, they did not provide an explicit proof or direct empirical evidence for this claim.
>   - We also design careful experiments in real-world setups to verify each claim (see Fig. 4, 14, and 16).
>
> - **In-depth discussions and extensions.** Beyond our central findings, we also investigate the necessity of using SAM in the early phase. We find that early-phase SAM may offer only limited improvements over SGD in terms of generalization and sharpness (see Fig. 5 and 15). Particularly, in Fig. 10, we observe that the sharpness rapidly increases once switching from SAM to GD and then oscillates around $2/\eta$​ (commonly known as the Edge of Stability phenomenon).
>   - **Conjecture on late training phase.** Together, contrary to the conventional view that early training is critical (Achille et al. 2019; Frankle et al. 2020), we conjecture that the optimization algorithm chosen at the end of training is more critical in shaping the final solutions’ properties.
>   - **Extension from SAM to AT.** We also validate our conjecture by extending our finding on SAM to Adversarial Training (AT). Specifically, we observe that AT efficiently enhances the robustness of neural networks when applied late in training.
>
> In summary, we believe our work provides distinct and significant contributions over Andriushchenko et al. (2022).
>
> We hope this clarification can address your question.
>
> Best,
>
> Authors of submission 1483

---

> > ### Comment · Area_Chair_5gE3 · 2024-11-29
> >
> > Dear Authors,
> >
> > Thank you for clarifying your contributions, which I do not question. However, if the main idea of your paper originates from prior work, it would be more appropriate to acknowledge this earlier in the introduction rather than on page 4. I trust that you will address this concern if the paper is accepted.
> >
> > Best regards,
> >
> > The AC

---

> > > ### Author Response · Authors · 2024-11-30
> > >
> > > Dear AC,
> > >
> > > Thank you for your suggestion. In our next revision, we will include additional commentary in the introduction to explicitly acknowledge that Andriushchenko et al. (2022) made a similar finding.
> > >
> > > Best regards,
> > >
> > > Authors of submission 1483

---

### Meta-Review · Area_Chair_5gE3 · 2024-12-19

**Metareview:**

The paper studies the SAM algorithm and analyzes its behavior when applied late in training. It offers valuable insights into the two-phase process where SAM first escapes sharp minima and subsequently converges to flatter ones. The reviewers appreciated the theoretical analysis and empirical results showing that a few epochs of SAM at the end of training can achieve comparable performance to full SAM.

The consensus is that the paper makes a clear contribution towards understanding SAM’s effectiveness. I recommend acceptance, given that the authors implement the changes discussed during the rebuttal and discussion period.

**Additional Comments On Reviewer Discussion:**

During the discussion, reviewers raised questions about the clarity of theoretical explanations and the scope of experiments supporting SAM's behavior. The authors addressed these concerns by providing clarifications on the two-phase training dynamics and offering additional insights into the experimental setup. Reviewers were satisfied with these responses and found the explanations sufficient. The overall consensus was that the theoretical and empirical contributions meaningfully advance understanding of SAM’s behavior. These points were carefully weighed in my final decision to recommend acceptance.

---

### Decision · Program_Chairs · 2025-01-22

Accept (Spotlight)